# Understanding Double Descent Requires a Fine-Grained Bias-Variance Decomposition

**Ben Adlam**[*][†]     **Jeffrey Pennington**[*]
Google Brain
{adlam, jpennin}@google.com

## Abstract

Classical learning theory suggests that the optimal generalization performance of a machine learning model should occur at an intermediate model complexity, with simpler models exhibiting high bias and more complex models exhibiting high variance of the predictive function. However, such a simple trade-off does not adequately describe deep learning models that simultaneously attain low bias and variance in the heavily overparameterized regime. A primary obstacle in explaining this behavior is that deep learning algorithms typically involve multiple sources of randomness whose individual contributions are not visible in the total variance. To enable fine-grained analysis, we describe an interpretable, symmetric decomposition of the variance into terms associated with the randomness from sampling, initialization, and the labels. Moreover, we compute the high-dimensional asymptotic behavior of this decomposition for random feature kernel regression, and analyze the strikingly rich phenomenology that arises. We find that the bias decreases monotonically with the network width, but the variance terms exhibit non-monotonic behavior and can diverge at the interpolation boundary, even in the absence of label noise. The divergence is caused by the *interaction* between sampling and initialization and can therefore be eliminated by marginalizing over samples (*i.e.* bagging) *or* over the initial parameters (*i.e.* ensemble learning).

## 1 Introduction

It is undeniable that modern neural networks (NNs) are becoming larger and more complex, with many state-of-the-art models now employing billions of trainable parameters [1–3]. While parameter count may be a crude way of quantifying complexity, there is little doubt that these models have enormous capacity, often far more than is needed to perfectly fit the training data, even if the labels are pure noise [4]. Surprisingly, these same high-capacity models generalize well when trained on real data.

These observations conflict with classical generalization theory, which contends that models of intermediate complexity should generalize best, striking a balance between the bias and the variance of their predictive functions. A paradigm for understanding the observed generalization behavior of modern methods is known as *double descent* [5], in which the test error behaves as predicted by classical theory and follows the standard U-shaped curve until the point where the training set can be fit exactly, but after this point it begins to descend again, eventually finding its global minimum in the overparameterized regime.

While double descent has been the focus of significant research, a concrete and interpretable theoretical explanation for the phenomenon has thus far been lacking. One of the challenges in developing

---

[*]Both authors contributed equally to this work. [†]Work done as a member of the Google AI Residency program (https://g.co/airesidency).

such an explanation is that the full phenomenology of double descent is not evident in linear models that are easy to analyze. Indeed, for linear models the number of parameters is tied to the number of features and there is no natural way to adjust the capacity of the model without simultaneously adjusting the data distribution. In this work, we overcome this challenge by providing a precise asymptotic analysis of random feature kernel regression, which is a model rich enough to exhibit all the interesting features of double descent.

Another challenge in understanding double descent is that the classical bias-variance decomposition is *itself* insufficiently nuanced to reveal all the underlying explanatory factors. Indeed, modern learning algorithms typically involve multiple sources of randomness and isolating the variation caused by each of these sources of randomness is key to building an effective interpretation. As we will see, it is not possible to fully understand the spike in test error near the interpolation threshold without performing a truly multivariate variance decomposition.

While decomposing the variance has been proposed before, prior work has naively relied on the law of total variance, which requires specifying an ordering of conditioning that leads to some arbitrariness. Instead, we present a principled symmetric decomposition which leads to unambiguous interpretations and clear credit assignment. Decomposing the variance of a random variable in this way is related to ANOVA [6], which has been used previously in a machine learning context to find the best approximating functions (in terms of mean squared error) to a random variable with limited dependence on the inputs [7, 8] and to study quasi Monte Carlo methods for integration [9].

Finally, we remark that an improved understanding of the bias and variance of machine learning models might naturally suggest ways to improve their performance. Specifically, any prior knowledge about what sources of variance may be dominant could help inform decisions about which types of ensemble or bagging techniques to utilize.

## 1.1 Related Work

The idea of a trade-off between bias and variance has a long history, with theoretical and experimental support having been well established in a variety of contexts over the years. The seminal paper of Geman et al. [10] examines a number of models, ranging from kernel regression to $k$-nearest neighbor to neural networks, and concludes that the trade-off exists in all cases[2]. The resulting U-shaped test error curve was verified theoretically in a variety of classical settings, see *e.g.* [11].

In recent years, these conclusions have been called into question by the intriguing experimental results of [4, 12], which were later replicated in a number of settings, see *e.g.* [13], which showed that deep neural networks and kernel methods can generalize well even in the interpolation regime, implying that both the bias and the variance can decrease as the model complexity increases. A number of theoretical results have since established this behavior in certain settings, such as interpolating nearest neighbor schemes [14] and kernel regression [15, 16]. These observations have given rise to the double descent paradigm for understanding how test error depends on model complexity [5]. The influential work [17] (which actually predates [5]) established initial theoretical insights for linear networks and found empirical evidence of double descent for nonlinear networks; more evidence has followed recently in [13, 18]. Precise theoretical predictions soon confirmed this picture for linear regression in various scenarios [19–22], and recently even for kernel regression [23, 24] with random features related to neural networks.

The primary focus of these recent works has been on double descent in the total test error, or perhaps the standard bias-variance decomposition with respect to label noise [23]. A multivariate philosophy similar to ours is advanced in [25], which revisited the empirical study of the bias-variance tradeoff in neural networks from [10] and showed the variance can decrease in the overparameterized regime. However, in that work the variance is simply decomposed using the law of total variance, which, while mathematically sound, can lead to ambiguous conclusions, as we discuss in Sec. 4.

The main mathematical tools we utilize come from random matrix theory and build on the results of [26–30] for studying random matrices with nonlinear dependencies. We also rely on techniques from operator-valued free probability for computing traces of large block matrices [31]. One advantage of these tools is that they facilitate the extension of our analysis to more general settings,

including the case of kernel regression with respect to the Neural Tangent Kernel (NTK) [32]. To ease the exposition we have deferred the discussion of the NTK and all proofs to the Supplementary Material (SM).

While finalizing this manuscript, we became aware of several concurrent works that examine similar questions. Yang et al. [33] define the total bias and variance similarly to [25], but they do not attempt a decomposition of the variance. Their results can be derived as a special case of our fine-grained decomposition by summing the variance terms in Thm. 1. Jacot et al. [34] study the relationship between the random feature model and the nonparametric Gaussian process which it approximates. The bias-variance decomposition considered in that paper is again univariate and is with respect to the randomness in the random features (the expressions are subsequently averaged over the training data). Closest to our work is [35], which also studies a multivariate decomposition of the random feature model in the high-dimensional limit. Unlike our approach, their decomposition is not symmetric with respect to the underlying random variables, and the results depend on the chosen order of conditioning. Their particular choice, and indeed all possible choices, arise as special cases of our general result. See Sec. S8 for a detailed discussion.

### 1.2 Our Contributions

1. We develop a symmetric, interpretable variance decomposition suitable for modern deep learning algorithms
2. We compute this decomposition analytically for random feature kernel regression in the high-dimensional asymptotic regime
3. We prove that the bias is monotonically decreasing as the width increases and that it is finite at the interpolation threshold
4. We clarify the relationship between label noise and double descent: while the test loss can diverge at the interpolation threshold without label noise, the divergence is exacerbated by it
5. We provide a quantitative description of how both ensemble and bagging methods can eliminate double descent, since the divergence is caused by variance terms due to the interactions between sampling and initialization

## 2 Bias-Variance Decomposition

In this section, we trace through the evolution of several ways to analyze the bias-variance trade-off. By analyzing their shortcomings, we motivate our fine-grained analysis that follows.

### 2.1 Classical Bias-Variance Decomposition

The bias-variance trade-off has long served as a useful paradigm for understanding the generalization of machine learning algorithms. For a given test point $\mathbf{x}$, it decomposes the expected error as

$$\mathbb{E}\left[\hat{y}(\mathbf{x}) - y(\mathbf{x})\right]^2 = \left(\mathbb{E}\hat{y}(\mathbf{x}) - \mathbb{E}y(\mathbf{x})\right)^2 + \mathbb{V}\left[\hat{y}(\mathbf{x})\right] + \mathbb{V}[y(\mathbf{x})], \quad (1)$$

and subsequently averages over the test point to obtain a decomposition of the test error in which the first term is the bias, the second term is the variance, and the third term is the irreducible noise. In classical settings, the randomness of the predictive function is usually regarded as coming from randomness in the training data, *i.e.* sampling noise. This leads to two common conventions, where the expectations in eqn. (1) are over both $X$ and $\mathbf{y}$ or are conditional on $X$ and only over the label noise in $\mathbf{y}$. For concreteness and to simplify the exposition, in this subsection we adopt the latter convention and make the common modelling assumption that the sampling noise is an additive term $\varepsilon$ on the training labels but is zero on the test labels $y(\mathbf{x})$. Using $\mathbb{E}_{\mathbf{x}}$ to denote expectation over the test point, we have

$$E_{\text{test}} := \mathbb{E}_{\mathbf{x}}\mathbb{E}_{\varepsilon}\left[\hat{y}(\mathbf{x}) - y(\mathbf{x})\right]^2 = \underbrace{\mathbb{E}_{\mathbf{x}}\left(\mathbb{E}_{\varepsilon}[\hat{y}(\mathbf{x})] - y(\mathbf{x})\right)^2}_{\text{Bias}} + \underbrace{\mathbb{E}_{\mathbf{x}}\mathbb{V}_{\varepsilon}\left[\hat{y}(\mathbf{x})\right]}_{\text{Variance}}. \quad (2)$$

We refer to eqn. (2) as the *classical bias-variance decomposition*.

### 2.2 Bias-Variance Decompositions for Modern Learning Methods

Modern methods for training neural networks often utilize additional sources of randomness, such as the initial parameter values, minibatch selection, *etc.*, which we collectively denote by $\theta$. One is

therefore left with a choice regarding whether or not to include $\theta$ in the expectations in eqn. (1), or to simply average over $\theta$ when computing the test loss. We explore the ramifications of these different choices below.

**Semi-classical Approach.** In what we call the *semi-classical* approach, the additional random variables $\theta$ coming from initialization or optimization are not included in the expectations in eqn. (1); we instead average over these quantities to define

$$E_{\text{test}} := \mathbb{E}_{\mathbf{x}}\mathbb{E}_{\theta}\mathbb{E}_{\varepsilon}[(\hat{y}(\mathbf{x}) - y(\mathbf{x}))^2|\theta] = \underbrace{\mathbb{E}_{\mathbf{x}}\mathbb{E}_{\theta}\left(\mathbb{E}_{\varepsilon}[\hat{y}(\mathbf{x})|\theta] - y(\mathbf{x})\right)^2}_{B_{SC}} + \underbrace{\mathbb{E}_{\mathbf{x}}\mathbb{E}_{\theta}\mathbb{V}_{\varepsilon}[\hat{y}(\mathbf{x})|\theta]}_{V_{SC}}. \quad (3)$$

In some scenarios, such as the high-dimensional setup analyzed in [23], the additional averaging over $\theta$ is unnecessary as the distributions concentrate around their mean. In those situations, the semi-classical decomposition is identical to the classical one, thus motivating this particular approach.

**Multivariate Approach.** In what we call the *multivariate approach*, the additional random variables $\theta$ are included in the expectations in eqn. (1), so that all random variables are on the same footing. We can then drop explicit references to $\theta$ and $\varepsilon$ and simply write,

$$E_{\text{test}} := \mathbb{E}_{\mathbf{x}}\mathbb{E}(\hat{y}(\mathbf{x}) - y(\mathbf{x}))^2 = \underbrace{\mathbb{E}_{\mathbf{x}}\left(\mathbb{E}[\hat{y}(\mathbf{x})] - y(\mathbf{x})\right)^2}_{B} + \underbrace{\mathbb{E}_{\mathbf{x}}\mathbb{V}[\hat{y}(\mathbf{x})]}_{V}. \quad (4)$$

One advantage of this perspective is that its form is completely symmetric with respect to the underlying random variables. Another is that the predictive function $\hat{y}(\mathbf{x})$ appearing in the bias $B$ is not conditional on any random variables. As we discuss in Sec. 4, this facilitates its interpretation as a measure of erroneous assumptions in the model.

The downside of this perspective is that the variance $V$ no longer admits a simple interpretation since it contains contributions from multiple random variables. This problem can be remedied by further decomposing the variance.

### 2.2.1 Symmetric Decomposition of the Variance

To gain further insight into the structure of the total variance $V$ and how individual random variables contribute to it, it can be useful to write $V$ as a sum of individual terms, each with an unambiguous meaning.

One path forward is to rely on the law of total variance: $\mathbb{V}[\mathcal{Y}] = \mathbb{E}\mathbb{V}[\mathcal{Y}|\mathcal{X}] + \mathbb{V}\mathbb{E}[\mathcal{Y}|\mathcal{X}]$, where the terms represent the variance of $\mathcal{Y}$ *unexplained* and *explained* by $\mathcal{X}$ respectively. However, one is immediately confronted by the question of which source of randomness to condition on. As we discuss in Sec. 4.2, different choices yield different terms and can lead to ambiguous interpretations.

To avoid this ambiguity, we introduce a fully-symmetric decomposition, which turns out to be unique if we additionally require self-consistency under marginalization with respect to all variables.

**Proposition 1.** *Let $X_1, \ldots, X_K$, and $Y$ be random variables and $\mathcal{X} := \{X_1, \ldots, X_K\}$. We define a* variance decomposition *of $Y$ to be a multiset $\{V_1, \ldots, V_N\}$ of nonnegative real numbers such that $\mathbb{V}[Y] = \sum_i V_i$. Then there exists a unique variance decomposition $\mathcal{V} := \{V_s : s \subseteq \mathcal{X}\}$ such that $\mathcal{V}$ is invariant under permutations of $\mathcal{X}$, and such that for all $S \subseteq \mathcal{X}$ the marginal variances satisfy the subset-sum relation,*

$$\mathbb{V}\mathbb{E}[Y|X_j \text{ for } j \in S] = \sum_{s \subseteq S} V_s. \quad (5)$$

**Example 1.** *Consider the case of two random variables, the parameters $P$ and the data $D$. Then $\mathcal{X} = \{P, D\}$ and the decomposition satisfying Prop. 1 is given by*

$$V_P := \mathbb{E}_{\mathbf{x}}\mathbb{V}\mathbb{E}[\hat{y}|P] \quad (6)$$

$$V_D := \mathbb{E}_{\mathbf{x}}\mathbb{V}\mathbb{E}[\hat{y}|D] \quad (7)$$

$$V_{PD} := \mathbb{E}_{\mathbf{x}}\mathbb{V}\mathbb{E}[\hat{y}|P, D] - \mathbb{E}_{\mathbf{x}}\mathbb{V}\mathbb{E}[\hat{y}|P] - \mathbb{E}_{\mathbf{x}}\mathbb{V}\mathbb{E}[\hat{y}|D]. \quad (8)$$

*We can interpret $V_{PD}$ as the variance explained by the parameters and data together beyond what they explain individually.*

**Example 2.** *Further decomposing $D$ into randomness from sampling the inputs $X$ and label noise $\varepsilon$, we can write $\mathcal{X} = \{P, X, \varepsilon\}$ and the decomposition satisfying Prop. 1 is given by,*

$$V_X := \mathbb{E}_{\mathbf{x}}\mathbb{VE}[\hat{y}|X], \tag{9}$$

$$V_{\varepsilon} := \mathbb{E}_{\mathbf{x}}\mathbb{VE}[\hat{y}|\varepsilon], \tag{10}$$

$$V_P := \mathbb{E}_{\mathbf{x}}\mathbb{VE}[\hat{y}|P], \tag{11}$$

$$V_{X\varepsilon} := \mathbb{E}_{\mathbf{x}}\mathbb{VE}[\hat{y}|X, \varepsilon] - \mathbb{E}_{\mathbf{x}}\mathbb{VE}[\hat{y}|X] - \mathbb{E}_{\mathbf{x}}\mathbb{VE}[\hat{y}|\varepsilon], \tag{12}$$

$$V_{PX} := \mathbb{E}_{\mathbf{x}}\mathbb{VE}[\hat{y}|P, X] - \mathbb{E}_{\mathbf{x}}\mathbb{VE}[\hat{y}|X] - \mathbb{E}_{\mathbf{x}}\mathbb{VE}[\hat{y}|P], \tag{13}$$

$$V_{P\varepsilon} := \mathbb{E}_{\mathbf{x}}\mathbb{VE}[\hat{y}|X, \varepsilon] - \mathbb{E}_{\mathbf{x}}\mathbb{VE}[\hat{y}|\varepsilon] - \mathbb{E}_{\mathbf{x}}\mathbb{VE}[\hat{y}|P], \tag{14}$$

$$V_{PX\varepsilon} := \mathbb{E}_{\mathbf{x}}\mathbb{VE}[\hat{y}|P, X, \varepsilon] - \mathbb{E}_{\mathbf{x}}\mathbb{VE}[\hat{y}|X, \varepsilon] - \mathbb{E}_{\mathbf{x}}\mathbb{VE}[\hat{y}|P, X] - \mathbb{E}_{\mathbf{x}}\mathbb{VE}[\hat{y}|X, \varepsilon]$$
$$+ \mathbb{E}_{\mathbf{x}}\mathbb{VE}[\hat{y}|X] + \mathbb{E}_{\mathbf{x}}\mathbb{VE}[\hat{y}|\varepsilon] + \mathbb{E}_{\mathbf{x}}\mathbb{VE}[\hat{y}|P]. \tag{15}$$

**Remark 1.** *Because $V_s \geq 0$ and $V = \mathbb{V}[\hat{y}] = \sum_s V_s$, the subset-sum relation (5) yields an interpretation of $V$ as the union of disjoint areas, forming a Venn diagram. See Fig. 1(d,e). The reader may also recognize the quantities above as those that are estimated in a three-way ANOVA.*

## 3 Asymptotic Variance Decomposition for Random Feature Regression

**Problem setup and notation.** Following prior work modeling double descent [21, 23, 24], we perform our analysis in the high-dimensional asymptotic scaling limit in which the dataset size $m$, feature dimensionality $n_0$, and hidden layer size $n_1$ all tend to infinity at the same rate, with $\phi := n_0/m$ and $\psi := n_0/n_1$ held constant.

We consider the task of learning an unknown function from $m$ independent samples $(\mathbf{x}_i, y_i) \in \mathbb{R}^{n_0} \times \mathbb{R}$, $i = 1, \ldots, m$, where the datapoints are standard Gaussian, $\mathbf{x}_i \sim \mathcal{N}(0, I_{n_0})$, and the labels are generated by a linear function parameterized by $\beta \in \mathbb{R}^{n_0}$, whose entries are drawn independently from $\mathcal{N}(0, 1)$. Concretely, we let

$$y(\mathbf{x}_i) = \beta^\top \mathbf{x}_i / \sqrt{n_0} + \varepsilon_i, \tag{16}$$

where $\varepsilon_i \sim \mathcal{N}(0, \sigma_{\varepsilon}^2)$ is additive label noise on the training points, yielding a signal-to-noise ratio SNR $= \sigma_{\varepsilon}^{-2}$. Although this may seem like a simple data distribution, it turns out that, in these high-dimensional asymptotics, the much more general setting in which the labels are produced by a non-linear teacher neural network can be exactly modeled with a linear teacher of this form (see Sec. S2.1).

We consider predictive functions $\hat{y}$ defined by approximate kernel ridge regression using the random feature model[3] of [36, 37], for which the random features are given by a single-layer neural network with random weights. Specifically, we define the random features on the training set $X = [\mathbf{x}_1, \ldots, \mathbf{x}_m]$ and test point $\mathbf{x}$ to be

$$F := \sigma(W_1 X / \sqrt{n_0}) \quad \text{and} \quad f := \sigma(W_1 \mathbf{x} / \sqrt{n_0}), \tag{17}$$

for a weight matrix $W_1 \in \mathbb{R}^{n_1 \times n_0}$ with iid entries $[W_1]_{ij} \sim \mathcal{N}(0, 1)$[4]. The kernel induced by these random features is

$$K(\mathbf{x}_1, \mathbf{x}_2) := \frac{1}{n_1}\sigma(W_1 \mathbf{x}_1 / \sqrt{n_0})^\top \sigma(W_1 \mathbf{x}_2 / \sqrt{n_0}), \tag{18}$$

and the model's predictions are given by

$$\hat{y}(\mathbf{x}) = Y K^{-1} K_{\mathbf{x}}, \tag{19}$$

where $Y := [y(\mathbf{x}_1), \ldots, y(\mathbf{x}_m)]$, $K := K(X, X) + \gamma I_m$, $K_{\mathbf{x}} := K(X, \mathbf{x})$, and $\gamma$ is a ridge regularization constant. For this model, $W_1$ plays the role of $\theta$ from Sec. 2.2.

Altogether, the test loss can be written as

$$E_{\text{test}} = \mathbb{E}_{\beta}\mathbb{E}_{\mathbf{x}}(y(\mathbf{x}) - \hat{y}(\mathbf{x}))^2 = \mathbb{E}_{\mathbf{x}}(\beta^\top \mathbf{x}/\sqrt{n_0} - Y K^{-1} K_{\mathbf{x}})^2, \tag{20}$$

where we dropped the outer expectation over $\beta$ because the distribution concentrates around its mean (see the SM).

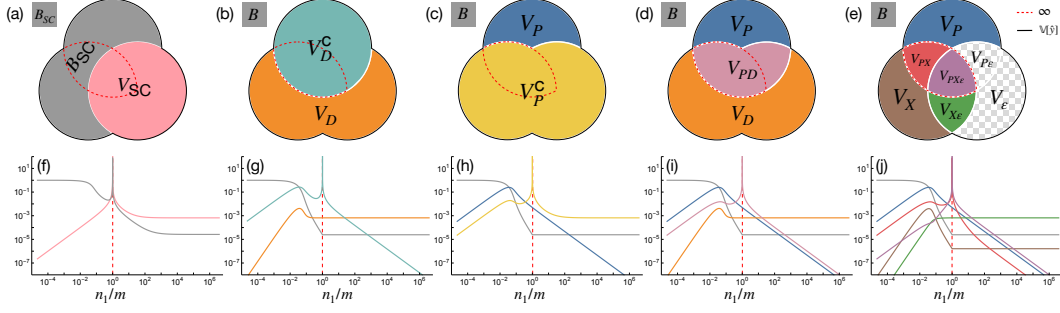

Figure 1: (a-e) The different bias-variance decompositions described in Sec. 4. (f-j) Corresponding theoretical predictions of Thm. 1 for $\gamma = 0$, $\phi = 1/16$ and $\sigma = \tanh$ with SNR = 100 as the model capacity varies across the interpolation threshold (dashed red). (a,f) The semi-classical decomposition of [21, 23] has a nonmonotonic and divergent bias term, conflicting with standard definitions of the bias. (b,g) The decomposition of [25] utilizing the law of total variance interprets the diverging term $V_D^C$ as "variance due to optimization". (c,h) An alternative application of the law of total variance suggests the opposite, *i.e.* the diverging term $V_P^C$ comes from "variance due to sampling". (d,i) A bivariate symmetric decomposition of the variance resolves this ambiguity and shows that the diverging term is actually $V_{PD}$, *i.e.* "the variance explained by the parameters and data together beyond what they explain individually." (e,j) A trivariate symmetric decomposition reveals that the divergence comes from two terms, $V_{PX}$ and $V_{PX\varepsilon}$ (outlined in dashed red), and shows that label noise exacerbates but does not cause double descent. Since $V_\varepsilon = V_{P\varepsilon} = 0$, they are not shown in (j).

## 3.1 Main Result: Exact Asymptotics for the Fine-Grained Variance Decomposition

**Lemma 1.** *Let $\eta := \mathbb{E}[\sigma(g)^2]$ and $\zeta := (\mathbb{E}[g\sigma(g)])^2$ for $g \sim \mathcal{N}(0,1)$. Then, in the high-dimensional asymptotics defined above, the traces $\tau_1(\gamma) := \frac{1}{m}\mathbb{E}\operatorname{tr}(K^{-1})$ and $\tau_2(\gamma) := \frac{1}{m}\mathbb{E}\operatorname{tr}(\frac{1}{n_0}X^\top X K^{-1})$ are given by the unique solutions to the coupled polynomial equations,*

$$\zeta\tau_1\tau_2(1 - \gamma\tau_1) = \phi/\psi\left(\zeta\tau_1\tau_2 + \phi(\tau_2 - \tau_1)\right) = (\tau_1 - \tau_2)\phi\left((\eta - \zeta)\tau_1 + \zeta\tau_2\right), \qquad (21)$$

*such that $\tau_1, \tau_2 \in \mathbb{C}^+$ for $\gamma \in \mathbb{C}^-$.*

**Theorem 1.** *Let $\tau_1$ and $\tau_2$ be defined as in Lemma 1, and use the prime symbol to denote their derivatives with respect to $\gamma$. Then, as $\Im(\gamma) \to 0^-$, the asymptotic bias and variance terms of eqns. (9)-(15) are given by*

$$
\begin{aligned}
B &= \tau_2^2/\tau_1^2 & V_{PX} &= -\tau_2'/\tau_1^2 - B - V_P - V_X \\
V_P &= \tau_2'/\tau_1' - B & V_{P\varepsilon} &= 0 \\
V_X &= \phi B(\tau_1 - \tau_2)^2/(\tau_1^2 - \phi(\tau_1 - \tau_2)^2) & V_{X\varepsilon} &= \sigma_\varepsilon^2 V_X/B \\
V_\varepsilon &= 0 & V_{PX\varepsilon} &= \sigma_\varepsilon^2(-\tau_1'/\tau_1^2 - 1) - V_{X\varepsilon}.
\end{aligned}
\qquad (22)
$$

**Corollary 1.** *In the ridgeless setting, the bias $B$ is a non-increasing function of the overparameterization ratio $n_1/m = \phi/\psi$. Furthermore, at the interpolation boundary $\psi = \phi$, $V_{PX}$ and $V_{PX\varepsilon}$ are divergent while the remaining terms are bounded.*

## 4 Fine-Grained Analysis of Double Descent

The fine-grained variance decomposition given in Thm. 1 provides a powerful tool for understanding the origins of double descent. In this section, we use this tool to reinterpret several counterintuitive observations made in prior work and to provide a clear and unambiguous characterization of the source of double descent.

### 4.1 Semi-classical Approach: The Bias Diverges

In [21, 23], double descent in random feature kernel regression was analyzed through the lens of the semi-classical bias-variance decomposition introduced in eqn. (3). In our setting,

$$E_{\text{test}} = B_{SC} + V_{SC}, \qquad (23)$$

where,

$$B_{SC} = \mathbb{E}_{\mathbf{x}} \mathbb{E}_{PX} \left( \mathbb{E}_{\boldsymbol{\epsilon}}[\hat{y}(\mathbf{x})|P, X] - y(\mathbf{x}) \right)^2 , \quad \text{and} \quad V_{SC} = \mathbb{E}_{\mathbf{x}} \mathbb{E}_{PX} [\mathbb{V}_{\boldsymbol{\epsilon}}[\hat{y}|P, X]|\mathbf{x}] . \tag{24}$$

To gain further insight into this decomposition, we can express $B_{SC}$ and $V_{SC}$ in terms of the variables in Thm. 1:

$$B_{SC} = B + V_P + V_X + V_{PX} , \quad \text{and} \quad V_{SC} = V_{\boldsymbol{\epsilon}} + V_{P\boldsymbol{\epsilon}} + V_{X\boldsymbol{\epsilon}} + V_{PX\boldsymbol{\epsilon}} . \tag{25}$$

Using the correspondence between the variance terms and areas mentioned in Remark 1, we illustrate this decomposition in Fig. 1(a). The figure shows that $B_{SC}$ is partially comprised of variance terms. Thm. 1 allows us to exactly characterize how $B_{SC}$ and $V_{SC}$ depend on the capacity of the model, with results shown in Fig. 1(f). As in [23], we observe that the bias $B_{SC}$ and variance $V_{SC}$ exhibit nonmonotonic behavior with respect to the model size and both diverge at the interpolation threshold.

Because $V_{\boldsymbol{\epsilon}} = V_{P\boldsymbol{\epsilon}} = 0$ and $V_{X\boldsymbol{\epsilon}}$ and $V_{PX\boldsymbol{\epsilon}}$ both vanish in the noiseless setting, the semi-classical decomposition has the nice property that $V_{SC} = 0$ when there is no label noise. However, it is hard to reconcile the nonmonotonicity of the bias with its desired interpretation as a measure of the erroneous assumptions in the model as the latter are expected to decrease as the model increases in capacity. For this reason, we believe the multivariate approach outlined in Sec. 2.2 provides a more interpretable basis for understanding double descent.

## 4.2 Multivariate Approach

**The Law of Total Variance: Ambiguous Conclusions.** Neal *et al.* [25] adopt the multivariate approach of Sec. 2.2 and decompose the test loss in terms of two sources of randomness, the optimization/initial parameters $P$ and data sampling $D$. The total variance is additionally decomposed according to the law of total variance:

$$V = \underbrace{\mathbb{E}_{\mathbf{x}} \mathbb{V}_D[\mathbb{E}_P[\hat{y}|D]|\mathbf{x}]}_{V_D} + \underbrace{\mathbb{E}_{\mathbf{x}} \mathbb{E}_D[\mathbb{V}_P[\hat{y}|D]|\mathbf{x}]}_{V_D^{\mathsf{c}}} , \tag{26}$$

where Neal *et al.* [25] suggests an interpretation for the two terms as "variance due to sampling" and "variance due to optimization," respectively. While the expressions in eqn. (26) are themselves unambiguous, we will see that attributing such an interpretation to them can be somewhat misleading.

Some simple algebra allows us to express $V_D^{\mathsf{c}}$ in terms of the terms in Thm. 1 as

$$V_D^{\mathsf{c}} = V_P + V_{PX} + V_{P\boldsymbol{\epsilon}} + V_{PX\boldsymbol{\epsilon}} . \tag{27}$$

Because eqn. (27) contains $V_{PX}$ and $V_{PX\boldsymbol{\epsilon}}$, Corollary 1 implies that $V_D^{\mathsf{c}}$ diverges at the interpolation threshold, and indeed we observe that in Fig. 1(g). From the above interpretation of the meaning of $V_D^{\mathsf{c}}$, we might therefore conclude that the "variance due to optimization" is the source of double descent.

On the other hand, we could have equally well decided to decompose the variance by conditioning on $P$ instead of $D$, yielding,

$$V = \underbrace{\mathbb{E}_{\mathbf{x}} \mathbb{V}_P[\mathbb{E}_D[\hat{y}|P]|\mathbf{x}]}_{V_P} + \underbrace{\mathbb{E}_{\mathbf{x}} \mathbb{E}_P[\mathbb{V}_D[\hat{y}|P]|\mathbf{x}]}_{V_P^{\mathsf{c}}} . \tag{28}$$

The corresponding interpretations of these terms would then be "variance due to optimization" and "variance due to sampling," respectively. As above, it is straightforward to express $V_P^{\mathsf{c}}$ as,

$$V_P^{\mathsf{c}} = V_X + V_{PX} + V_{X\boldsymbol{\epsilon}} + V_{PX\boldsymbol{\epsilon}} . \tag{29}$$

In this case, Corollary 1 implies that $V_P^{\mathsf{c}}$ diverges at the interpolation threshold, as Fig. 1(h) confirms. In this case, we might therefore conclude that the "variance due to sampling" is the source of double descent.

The above analysis reveals conflicting explanations for the source double descent, depending on which source of randomness is conditioned on when applying the law of total variance. We believe this ambiguity is undesirable and provides further motivation for the symmetric variance decomposition in Prop. 1.

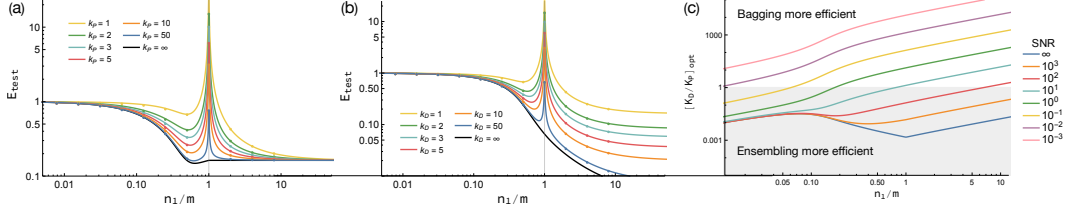

Figure 2: Comparison of (a) ensembles and (b) bagging. Solid lines are theoretical predictions and dots are simulation results. In (a,b) we set $\gamma = 10^{-6}$, $n_0 = 2^{13}$, $m = 2^{14}$, $\sigma = \tanh$, and SNR = 5. Note that as either $k_P$ or $k_D$ increase, the peak around the interpolation threshold decreases. In (c), we plot the optimal ratio $[k_D/k_P]_{\text{optimal}}$ (35) as a function of $n_1/m$ for different SNRs. The shaded area, $[k_D/k_P]_{\text{optimal}} < 1$, is where averaging over the parameters reduces variance more efficiently. As expected, for large width, bagging is much more efficient.

**Bivariate Symmetric Decomposition: $V_{PD}$ is the Source of Divergence.** In the previous two-variable setting, the symmetric decomposition can be written as (see Example 1),

$$V = V_P + V_D + V_{PD}. \tag{30}$$

See Fig. 1(d) for an illustration of this decomposition. This figure shows that $V_{PD}$ inhabits the ambiguous overlap region that was responsible for the inconsistent interpretations arising from a naive application of the law of total variance. From the theoretical results shown in Fig. 1(i), it is clear that neither the variance explained by the parameters, $V_P$, nor the variance explained by the data, $V_D$, can be responsible for double descent; instead it must be $V_{PD}$ that is causing the divergence. Recalling the definition of $V_{PD}$ in Ex. 1, we conclude that the divergence at the interpolation boundary is caused by "the variance explained by the parameters and training data together beyond what they explain individually."

One implication of this interpretation is that if we had a way of removing *either* the variance from the parameters *or* the variance from the data, then the divergence would be eliminated. We examine this phenomenon from the perspective of ensemble and bagging methods in Sec. 5 and confirm empirically that this is indeed the case. See Fig. 2.

**Trivariate Symmetric Decomposition: Divergence Persists in Absence of Label Noise.** Returning to the full model from Sec. 3 with three sources of randomness, we know from Thm. 1 that

$$V = V_P + V_X + V_{PX} + V_{X\varepsilon} + V_{PX\varepsilon}, \tag{31}$$

while the other two variance terms $V_\varepsilon$ and $V_{P\varepsilon}$ vanish. The seven variance terms are illustrated in Fig. 1(e). The dependence of the five non-zero terms on the model's capacity is plotted in Fig. 1(j). We find that $V_{PX}$ and $V_{PX\varepsilon}$ both diverge at the interpolation threshold while the other terms remain finite. This result helps explain recent empirical results that have found that label noise amplifies the double descent phenomena [13]: because $V_{PX}$ itself diverges, there is double descent even without label noise, but because $V_{PX\varepsilon}$ also diverges, label noise can exacerbate the effect.

## 5 Ensemble Learning

The understanding we have developed for the sources of variance enables explicit prediction of the effectiveness of ensemble and bagging techniques. We consider averaging the predictive functions of several independently initialized base learners as well as bagging the predictions from models with independent samples of training data. Specifically, we consider $k_P$ independent samples of the parameters, $P_i$, and $k_D$ independent samples of the training data, $X_j$ and $\varepsilon_j$. Then our predictive function on a test point $\mathbf{x}$ is

$$\hat{y}^*(\mathbf{x}) := \frac{1}{k_P k_D} \sum_{i,j} \hat{y}_{ij}(\mathbf{x}), \tag{32}$$

where the indicies of $\hat{y}$ indicate the specific sample of parameters and training data used to construct the predictor. A simple calculation gives the variance decomposition of $\hat{y}^*$ as

$$V_P^* = \frac{V_P}{k_P}, \quad V_X^* = \frac{V_X}{k_D}, \quad V_\varepsilon^* = \frac{V_\varepsilon}{k_D}, \quad V_{X\varepsilon}^* = \frac{V_{X\varepsilon}}{k_D}, \tag{33}$$

$$V_{P\varepsilon}^* = \frac{V_{P\varepsilon}}{k_P k_D}, \ V_{PX}^* = \frac{V_{PX}}{k_P k_D}, \text{ and } V_{PX\varepsilon}^* = \frac{V_{PX\varepsilon}}{k_P k_D}, \tag{34}$$

while the bias remains the same. We illustrate these results empirically in Fig. 2 and show that ensembles of base learners and bagging are both able to independently reduce the divergence around the interpolation threshold, as they reduce the divergent terms $V_{PX}$ and $V_{PX\varepsilon}$.

As the computation of eqn. (32) requires evaluating $k_P k_D$ base learners, it is natural to try to characterize the optimal combination of ensembles and bagging given a fixed computational budget. We find the optimal ratio is given as

$$[k_D/k_P]_{\text{optimal}} = (V_X + V_\varepsilon + V_{X\varepsilon})/V_P . \tag{35}$$

See Fig. 2, which shows that, for the kernel regression problem studied here, ensembles are typically more efficient at small width and bagging is more efficient at large width.

## 6  Conclusion

We analyzed the bias and variance trade-off in the modern setting, where the difference to the classical picture of under- and overfitting is marked. We argued that understanding the behavior of the bias and variance in learning algorithms that depend on large sources of randomness requires rethinking the classical definitions to encompass these sources.

We presented a bias-variance decomposition that is suitable for these settings, and showed how it can help attribute components of the loss to their causes, while avoiding counterintuitive or ambiguous conclusions. For random feature kernel regression, we gave exact predictions for all of the terms in the decomposition and proved that the bias is monotonically decreasing and identified the source of divergence at the interpolation threshold to be the interaction between the noise from sampling and initialization. We showed that while label noise does not cause the divergence, it can exacerbate the effect. Finally, we made exact predictions for ensemble learning and bagging and provided the computationally optimal strategy to combine them.

## Broader Impact

While it is hard to envision all future applications of this research, the authors do not believe this theoretical work will raise any ethical concerns or will generate any adverse future societal consequences.

## Acknowledgments and Disclosure of Funding

We are grateful to Boris Hanin, Jaehoon Lee, Mihai Nica, D. Sculley, Jasper Snoek, and Lechao Xiao for valuable feedback on an earlier version of the paper. We also thank the anonymous reviewers for pointing us to many related works, including the connection to ANOVA.

Funding in direct support of this work came from Google. No third party funding was used.

## Footnotes

[2]Interestingly, the variance of simple feed-forward neural networks was observed to eventually be a decreasing function of width, but the authors rationalized this early evidence of double descent as a quirk of the optimization.

[3]See the SM for an extension to the Neural Tangent Kernel of a single-hidden-layer neural network [32].

[4]Any non-zero variance $\sigma_{W_1}^2$ can be absorbed into a redefinition of $\sigma$.

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
