[Supplementary Material]

# S1 Symmetric variance decomposition

The purpose of this section is to prove Prop. 1 and derive eqns. (33)-(35). The strategy is to use the subset-sum relationship, eqn. (5), as a definition, derive explicit formulae for the variance terms, then prove all terms are nonnegative. In the statistics literature this approach is referred to as functional ANOVA [6–9], but we present a derivation here as it may be unfamiliar to members of the machine learning community.

**Motivation.** The law of total variance for two random variables $X$ and $Y$ is

$$\mathbb{V}[Y] = \mathbb{E}\mathbb{V}[Y|X] + \mathbb{V}\mathbb{E}[Y|X], \tag{S1}$$

where the two terms represents the variance of $Y$ that is unexplained and explained by $X$ respectively. Since the variance must be nonnegative it is possible to interpret it as an area. In Fig. S1a the total variance is represented by the square, which is in turn broken up into the explained variance (red circle) and unexplained variance (area outside of the circle).

It is possible to extend this idea to several variables. An observation that is key to the interpretation is

$$\mathbb{V}\mathbb{E}[Y|X_1] + \mathbb{V}\mathbb{E}[Y|X_2] \leq \mathbb{V}\mathbb{E}[Y|X_1, X_2], \tag{S2}$$

*i.e.* the "variance explained" is a superadditive function. So the decomposition for two variables could be written as

$$\mathbb{V}[Y] = \mathbb{V}\mathbb{E}[Y|X_1] + \mathbb{V}\mathbb{E}[Y|X_2] + (\mathbb{V}\mathbb{E}[Y|X_1, X_2] - \mathbb{V}\mathbb{E}[Y|X_1] - \mathbb{V}\mathbb{E}[Y|X_2]) + \mathbb{E}\mathbb{V}[Y|X_1, X_2], \tag{S3}$$

with the terms interpreted as the variance explained by $X_1$, the variance explained by $X_2$, the additional variance explained by $X_1$ and $X_2$ together, and the variance left unexplained by $X_1$ and $X_2$. Note that the terms are all guaranteed to be positive by eqn. (S2). See Fig S1b

**Several variables.** Generalizing, let $\mathbf{X} := (X_1, \ldots, X_k)$ be a collection of random variables. Consider a Venn diagram of $k$ circles, and denote the disjoint areas using $V_\mathbf{i}$ for a vector $\mathbf{i} \in \{0, 1\}^k$, where $i_j$ indicates whether the area is inside the $j$th circle (see Fig S1). We make use of the natural partial ordering on $\{0, 1\}^k$, *i.e.* $\mathbf{i} \leq \mathbf{j}$ if and only if $i_l \leq j_l$ for all $l$. Note the ordering indicates the subset relation if the vectors are thought of as indicator vectors. We also use the notation $\mathbf{e}_j$ for the standard basis vectors, and define the vectors $\mathbf{X}_\mathbf{i} := (X_j : i_j = 1)$.

For simplicity, assume $Y \in \sigma(\mathbf{X})$, so that $\mathbb{E}[f(Y)|\mathbf{X}] = f(Y)$ for any measurable function $f$ and all the variance of $Y$ is explained by $\mathbf{X}$, *i.e.*

$$\mathbb{V}\mathbb{E}[Y|\mathbf{X}] = \mathbb{V}[Y] \tag{S4}$$

or $V_\mathbf{0} = 0$. In fact, let us write $Y = h(\mathbf{X})$. We make this assumption without loss of generality as one can otherwise consider $X_{k+1} := Y - \mathbb{E}(Y|\mathbf{X})$, *i.e.* the orthogonal complement of $Y$ under projection onto the sigma algebra generated by $\mathbf{X}$.

Consistent with the $k = 1$ case, we define

$$V_{\mathbf{e}_j} = \mathbb{V}\mathbb{E}[Y|X_j], \tag{S5}$$

or more generally

$$\sum_{\mathbf{i}:\mathbf{i} \leq \mathbf{j}} V_\mathbf{i} = \mathbb{V}\mathbb{E}[Y|\mathbf{X}_\mathbf{j}]. \tag{S6}$$

Eqn. (S6) is exactly the subset-sum relationship in (5).

**Lemma S1.** *Eqn. (S6) is sufficient to define $V_\mathbf{i}$ for all $\mathbf{i}$.*

*Proof.* This lemma follows directly from the fact that (S6) defines $2^k$ equations in terms of $2^k$ unknowns, $V_\mathbf{i}$. However, we may get a more explicit solution for each $V_\mathbf{i}$: We proceed by induction on $|\mathbf{i}| := \sum_j i_j$. The special case of eqn. (S6), eqn. (S5), proves the base case, $|\mathbf{i}| = 1$. Assume we have defined $V_\mathbf{i}$ for all $|\mathbf{i}| \leq m$. Then for $\mathbf{i}$ such that $|\mathbf{i}| = m + 1$, using eqn. (S6) we may write

$$V_\mathbf{i} + \sum_{\mathbf{j}:\mathbf{j} \leq \mathbf{i}, \mathbf{j} \neq \mathbf{i}} V_\mathbf{j} = \mathbb{V}\mathbb{E}[Y|\mathbf{X}_\mathbf{i}]. \tag{S7}$$

Noting that $|\mathbf{j}| \leq m$ if $\mathbf{j} \leq \mathbf{i}$ and $\mathbf{j} \neq \mathbf{i}$ completes the proof. $\qquad\square$

**Calculating Variances** To calculate the variances terms used above for our model, we use a coupling: we introduce a copy of the underlying random variables $\mathbf{X}$ and take expectations under different independence assumptions on $\mathbf{X}$ and its copy. The simplest illustration of this idea is to express the variance of a random variable $Y$ using an iid copy of $Y$, denoted $Y'$, then define $\tilde{Y} := BY + (1 - B)Y'$ for $B \sim \text{Bern}(1/2)$ independent of $Y$ and $Y'$. We have

$$\mathbb{V}[Y] = \mathbb{E}[Y^2] - \mathbb{E}[Y]^2 = \mathbb{E}[YY] - \mathbb{E}[YY'] = \mathbb{E}[Y\tilde{Y}|B = 1] - \mathbb{E}[Y\tilde{Y}|B = 0]. \tag{S8}$$

This idea extends naturally to our setting. Recall $Y = h(\mathbf{X})$, then let $\tilde{\mathbf{X}} := (B_1 X_1 + (1 - B_1)X_1', \ldots, B_k X_k + (1 - B_k)X_k')$ for $\mathbf{X}'$ an iid copy of $\mathbf{X}$ and $B_i \sim \text{Bern}(1/2)$ iid. Define

$$H_{\mathbf{i}} := \mathbb{E}\left[h(\mathbf{X})h(\tilde{\mathbf{X}})|\mathbf{B} = \mathbf{i}\right]. \tag{S9}$$

Thus, $\mathbb{V}\mathbb{E}[Y|\mathbf{X_i}] = H_{\mathbf{i}} - H_{\mathbf{0}}$ and $\mathbb{E}\mathbb{V}[Y|\mathbf{X_i}] = H_{\mathbf{1}} - H_{\mathbf{i}}$.

**Theorem S1.** *Using $H$, we have the following formula for the areas. Let $|\mathbf{i}| > 0$, then*

$$V_{\mathbf{i}} := \sum_{l=0}^{|\mathbf{i}|} \sum_{\mathbf{j}:\mathbf{j}\leq\mathbf{i},|\mathbf{j}|=l} (-1)^{|\mathbf{i}|-l} H_{\mathbf{j}}. \tag{S10}$$

*Proof.* We again use induction on $|\mathbf{i}|$. The formula clearly holds for $|\mathbf{i}| = 1$. Then using the induction hypothesis and eqn. (S7), we see

$$V_{\mathbf{i}} = H_{\mathbf{i}} - H_{\mathbf{0}} - \sum_{\mathbf{j}:\mathbf{j}\leq\mathbf{i},\mathbf{j}\neq\mathbf{i}} \sum_{l=0}^{|\mathbf{j}|} \sum_{\mathbf{k}:\mathbf{k}\leq\mathbf{j},|\mathbf{k}|=l} (-1)^{|\mathbf{j}|-l} H_{\mathbf{k}} \tag{S11}$$

$$= \sum_{l=0}^{|\mathbf{i}|} \sum_{\mathbf{j}:\mathbf{j}\leq\mathbf{i},|\mathbf{j}|=l} (-1)^{|\mathbf{i}|-l} H_{\mathbf{j}}.$$

$\square$

**Lemma S2.** *The function $H$ is partially ordered, that is*

$$H_{\mathbf{i}} \leq H_{\mathbf{j}}, \tag{S12}$$

*if and only if $i_k \leq j_k$ for all $k \in \{1, \ldots, k\}$.*

*Proof.* Define $Z := \mathbb{E}[Y|X_{\mathbf{i}}]$, then

$$H_{\mathbf{i}} - H_{\mathbf{j}} = \mathbb{E}\left[Z^2 - \mathbb{E}[Z|X_{\mathbf{j}}]^2\right] \geq \mathbb{E}\left[Z^2 - \mathbb{E}[Z^2|X_{\mathbf{j}}]\right] = 0. \tag{S13}$$

$\square$

**Theorem S2.** *The areas $V_{\mathbf{i}}$ are nonnegative.*

*Proof.* The idea is similar to the proof of Lemma S2, and indeed this generalize the result. First we prove eqn. (S2) to illustrate the idea with simple notation. We see

$$\mathbb{V}\mathbb{E}[Y|X_1, X_2] - \mathbb{V}\mathbb{E}[Y|X_1] - \mathbb{V}\mathbb{E}[Y|X_2] \tag{S14}$$

$$= H_{11} + H_{00} - H_{01} - H_{10}$$

$$= \frac{1}{4}\mathbb{E}\left(h(X_1, X_2) + h(\tilde{X}_1, \tilde{X}_2) - h(\tilde{X}_1, X_2) - h(X_1, \tilde{X}_2)\right)^2$$

$$\geq 0.$$

Figure S1: In (a) the two disjoint areas represent $\mathbb{VE}[Y|X]$, the variance of $Y$ explained by $X$, and $\mathbb{EV}[Y|X]$, the variance of $Y$ unexplained by $X$. For simplicity, we assume that in (b) and (c) there is no variance that is not explained by $\mathbf{X}$, so that the area outside of the circles is zero.

For the general case, fix $\mathbf{i}$ and define

$$\bar{h}(\mathbf{X_i}) := \mathbb{E}_{\mathbf{X_{1-i}}}[h(\mathbf{X})|\mathbf{X_i}], \tag{S15}$$

that is, marginalize over all $X_j$ such that $i_j = 0$. Then note for $\mathbf{j} \leq \mathbf{i}$ that

$$H_{\mathbf{j}} = \mathbb{E}h(\mathbf{X_j}, \mathbf{X_{i-j}}, \mathbf{X_{1-i}})h(\mathbf{X_j}, \tilde{\mathbf{X}}_{\mathbf{i-j}}, \tilde{\mathbf{X}}_{\mathbf{1-i}}) = \mathbb{E}\bar{h}(\mathbf{X_j}, \mathbf{X_{i-j}})\bar{h}(\mathbf{X_j}, \tilde{\mathbf{X}}_{\mathbf{i-j}}). \tag{S16}$$

Now using Theorem S1, we see

$$\begin{aligned}
V_{\mathbf{i}} &= \sum_{l=0}^{|\mathbf{i}|} \sum_{\mathbf{j}:\mathbf{j}\leq\mathbf{i}, |\mathbf{j}|=l} (-1)^{|\mathbf{i}|-l} \mathbb{E}\bar{h}(\mathbf{X_j}, \mathbf{X_{i-j}})\bar{h}(\mathbf{X_j}, \tilde{\mathbf{X}}_{\mathbf{i-j}}) \\
&= \frac{1}{2^{|\mathbf{i}|}} \mathbb{E}\left( \sum_{\mathbf{j}:\mathbf{j}\leq\mathbf{i}} \bar{h}(\mathbf{X_j}, \tilde{\mathbf{X}}_{\mathbf{i-j}}) \right)^2 \\
&\geq 0.
\end{aligned} \tag{S17}$$

$\square$

**Examples for $k = 2$ and $k = 3$ used in the main text.** See Fig S1b. For $k = 2$, we have:

$$\begin{aligned}
V_{01} &= H_{10} - H_{00} \\
V_{10} &= H_{01} - H_{00} \\
V_{11} &= H_{11} - H_{01} - H_{10} + H_{00}.
\end{aligned}$$

See Fig S1c. For $k = 3$, we have:

$$
\begin{aligned}
V_{001} &= H_{001} - H_{000} \\
V_{010} &= H_{010} - H_{000} \\
V_{100} &= H_{100} - H_{000} \\
V_{011} &= H_{011} - H_{001} - H_{010} + H_{000} \\
V_{101} &= H_{101} - H_{001} - H_{100} + H_{000} \\
V_{110} &= H_{110} - H_{010} - H_{100} + H_{000} \\
V_{111} &= H_{111} - H_{011} - H_{101} - H_{110} + H_{001} + H_{010} + H_{100} - H_{000}.
\end{aligned}
\tag{S18}
$$

**Ensemble and bagging formulas.** To obtain these results, we first calculate the $H_{\mathbf{i}}$ terms associated with the averaged predictor. Specifically, define $P := \{P_1, \ldots, P_{k_P}\}$, $X := \{X_1, \ldots, X_{k_D}\}$, $\varepsilon := \{\varepsilon_1, \ldots, \varepsilon_{k_D}\}$, and

$$
Y := \frac{1}{k_P k_D} \sum_{i,j} \hat{y}_{ij}(\mathbf{x}),
\tag{S19}
$$

where the indices denote iid samples. We consider the variance decomposition of $Y$ with respect to $P$, $X$, and $\varepsilon$. Note, we could instead use the notation

$$
Y = \hat{y}(P, X, \varepsilon) = \frac{1}{k_P k_D} \sum_{i=1}^{k_P} \sum_{j=1}^{k_D} \hat{y}(P_i, X_j, \varepsilon_j),
\tag{S20}
$$

to make explicit the dependence on each of the random variables.

Clearly, $\mathbb{E}\hat{y}(P, X, \varepsilon) = \hat{y}(P_1, X_1, \varepsilon_1)$, so the predictors have the same bias. Now, we calculate the $H$s using superscripts to denote the ensemble and bagging sizes. First,

$$
H_{000}^{k_P k_D} = \mathbb{E}\hat{y}(P, X, \varepsilon)\hat{y}(\tilde{P}, \tilde{X}, \tilde{\varepsilon}) = H_{000}^{11}.
\tag{S21}
$$

Next, we see

$$
\begin{aligned}
H_{100}^{k_P k_D} &= \mathbb{E}\hat{y}(P, X, \varepsilon)\hat{y}(P, \tilde{X}, \tilde{\varepsilon}) \\
&= \frac{1}{k_P^2 k_D^2} \sum_{i=1}^{k_P} \sum_{j=1}^{k_D} \sum_{i'=1}^{k_P} \sum_{j'=1}^{k_D} \mathbb{E}\hat{y}(P_i, X_j, \varepsilon_j)\hat{y}(P_{i'}, \tilde{X}_{j'}, \tilde{\varepsilon}_{j'}) \\
&= \frac{1}{k_P^2 k_D^2} \sum_{i,j,j'} \mathbb{E}\hat{y}(P_i, X_j, \varepsilon_j)\hat{y}(P_i, \tilde{X}_{j'}, \tilde{\varepsilon}_{j'}) + \frac{1}{k_P^2 k_D^2} \sum_{i \neq i'} \sum_{j,j'} \mathbb{E}\hat{y}(P_i, X_j, \varepsilon_j)\mathbb{E}\hat{y}(P_{i'}, \tilde{X}_{j'}, \tilde{\varepsilon}_{j'}) \\
&= \frac{H_{100}^{11} - H_{000}^{11}}{k_P} + H_{000}^{11}.
\end{aligned}
\tag{S22}
$$

Similarly, to above we find

$$
H_{010}^{k_P k_D} = \frac{H_{010}^{11} - H_{000}^{11}}{k_D} + H_{000}^{11},
\tag{S23}
$$

$$
H_{001}^{k_P k_D} = \frac{H_{001}^{11} - H_{000}^{11}}{k_D} + H_{000}^{11},
\tag{S24}
$$

and

$$
H_{011}^{k_P k_D} = \frac{H_{011}^{11} - H_{000}^{11}}{k_D} + H_{000}^{11}.
\tag{S25}
$$

The other terms are more complex, but the idea is the same. We find

$$H_{110}^{k_P k_D} = \frac{1}{k_P^2 k_D^2} \sum_{i=1}^{k_P} \sum_{j=1}^{k_D} \sum_{i'=1}^{k_P} \sum_{j'=1}^{k_D} \mathbb{E}\hat{y}(P_i, X_j, \varepsilon_j)\hat{y}(P_{i'}, X_{j'}, \tilde{\varepsilon}_{j'})$$

$$= \frac{1}{k_P^2 k_D^2} \sum_i \sum_j \mathbb{E}\hat{y}(P_i, X_j, \varepsilon_j)\hat{y}(P_i, X_j, \tilde{\varepsilon}_j) + \frac{1}{k_P^2 k_D^2} \sum_{i \neq i'} \sum_j \mathbb{E}\hat{y}(P_i, X_j, \varepsilon_j)\hat{y}(P_{i'}, X_j, \tilde{\varepsilon}_j) \tag{S26}$$

$$+ \frac{1}{k_P^2 k_D^2} \sum_i \sum_{j \neq j'} \mathbb{E}\hat{y}(P_i, X_j, \varepsilon_j)\hat{y}(P_i, X_{j'}, \tilde{\varepsilon}_{j'}) + \frac{1}{k_P^2 k_D^2} \sum_{i \neq i'} \sum_{j \neq j'} \mathbb{E}\hat{y}(P_i, X_j, \varepsilon_j)\hat{y}(P_{i'}, X_{j'}, \tilde{\varepsilon}_{j'}) \tag{S27}$$

$$= \frac{H_{110}^{11} - H_{010}^{11} - H_{100}^{11} + H_{000}^{11}}{k_D k_P} + \frac{H_{010}^{11} - H_{000}^{11}}{k_P} + \frac{H_{100}^{11} - H_{000}^{11}}{k_D} + H_{000}^{11}. \tag{S28}$$

Similarly, we have

$$H_{101}^{k_P k_D} = \frac{H_{101}^{11} - H_{001}^{11} - H_{100}^{11} + H_{000}^{11}}{k_D k_P} + \frac{H_{001}^{11} - H_{000}^{11}}{k_P} + \frac{H_{100}^{11} - H_{000}^{11}}{k_D} + H_{000}^{11} \tag{S29}$$

and

$$H_{111}^{k_P k_D} = \frac{H_{111}^{11} - H_{011}^{11} - H_{100}^{11} + H_{000}^{11}}{k_D k_P} + \frac{H_{011}^{11} - H_{000}^{11}}{k_P} + \frac{H_{100}^{11} - H_{000}^{11}}{k_D} + H_{000}^{11}. \tag{S30}$$

Finally, substituting the expressions for the $H$s into eqn. (S18) and simplifying completes the derivation.

To find the optimal ratio, we write the test error as

$$B + \frac{V_P}{k_P} + \frac{V_X}{k_D} + \frac{V_\varepsilon}{k_D} + \frac{V_{X\varepsilon}}{k_D} + \frac{V_{PX}}{k_P k_D} + \frac{V_{P\varepsilon}}{k_P k_D} + \frac{V_{PX\varepsilon}}{k_P k_D} \tag{S31}$$

and substitute $k_D = K/k_P$, where $K$ is a fixed constant. Then differentiating eqn. (S31) with respect to $k_P$ and solving for the stationary point yields eqn. (35).

## S2 Model Definitions for the Full Neural Tangent Kernel

For clarity of presentation, in the main text we focused on a linear teacher and a simple unstructured random feature model. This model can also be viewed as a degeneration of the Neural Tangent Kernel (NTK) of a single-hidden-layer neural network under which the first-layer weights are held at their randomly-initialized values and only the second-layer weights are optimized. Our analysis and results actually extend to the full NTK, where all weights are optimized, and to a wide nonlinear teacher neural network. The results in the main text are special cases of the more general results we present here.

### S2.1 Data distribution

Following [24], we consider the task of learning an unknown function from $m$ independent samples $(\mathbf{x}_i, y_i) \in \mathbb{R}^{n_0} \times \mathbb{R}$, $i \leq m$, where the datapoints are standard Gaussian, $\mathbf{x}_i \sim \mathcal{N}(0, I_{n_0})$, and the labels are generated by a wide[5] single-hidden-layer neural network:

$$y_i | \mathbf{x}_i, \Omega, \omega \sim \omega \sigma_{\mathrm{T}}(\Omega \mathbf{x}_i / \sqrt{n_0})/\sqrt{n_{\mathrm{T}}} + \varepsilon_i. \tag{S32}$$

The teacher's activation function $\sigma_{\mathrm{T}}$ is applied coordinate-wise, and its parameters $\Omega \in \mathbb{R}^{n_{\mathrm{T}} \times n_0}$ and $\omega \in \mathbb{R}^{1 \times n_{\mathrm{T}}}$ are matrices whose entries are independently sampled once for all data from $\mathcal{N}(0, 1)$. We also allow for independent label noise, $\varepsilon_i \sim \mathcal{N}(0, \sigma_\varepsilon^2)$. In this case, the test loss for a predictive function $\hat{y}$ becomes,

$$\mathbb{E}(\omega \sigma_{\mathrm{T}}(\Omega \mathbf{x} / \sqrt{n_0})/\sqrt{n_{\mathrm{T}}} + \varepsilon - \hat{y}(\mathbf{x}))^2. \tag{S33}$$

Recall that in our high-dimensional asymptotics the limiting ratios $n_0/m \to \phi$ and $n_0/n_1 \to \psi$ are constant. As we will discuss in Sec. S3, in this regime only linear functions of the data can be learned, a finding that is consistent with observations made in [38, 23]. When the teacher width $n_\mathrm{T} \to \infty$, a precise decomposition of the teacher emerges that neatly captures its learning and unlearnable components. Specifically, if we define,

$$\zeta_\mathrm{T} := \left( \mathbb{E}\sigma'_\mathrm{T}(g) \right)^2, \quad \text{and} \quad \eta_\mathrm{T} := \mathbb{E}\sigma_\mathrm{T}(g)^2, \tag{S34}$$

then there is an equivalent linear teacher plus noise with signal-to-noise ratio given by,

$$\mathrm{SNR} = \zeta_\mathrm{T} / \left( \eta_\mathrm{T} - \zeta_\mathrm{T} + \sigma_\varepsilon^2 \right). \tag{S35}$$

We often make this equivalence to a linear teacher explicit by setting $\sigma_\mathrm{T}(x) = x$ (which implies $\eta_\mathrm{T} = \zeta_\mathrm{T} = 1$) and explicitly adding label noise $\sigma_\varepsilon^2 = 1/\mathrm{SNR}$. This procedure also removes the noise from the test label, but since this noise merely contributes an additive shift to the test loss, removing it does not change any of our conclusions.

## S2.2  NTK Regression

We consider predictive functions $\hat{y}$ defined by approximate (*i.e.* random feature) kernel ridge regression using the NTK of a single-hidden-layer neural network of width $n_1$ with entry-wise activation function $\sigma$, defined by,

$$N_0(\mathbf{x}) = W_2 \sigma(W_1 \mathbf{x}/\sqrt{n_0})/\sqrt{n_1}, \tag{S36}$$

for initial $n_1 \times n_0$ and $1 \times n_1$ weight matrices with iid entries $[W_1]_{ij} \sim \mathcal{N}(0,1)$[6] and $[W_2]_i \sim \mathcal{N}(0,\sigma_{W_2}^2)$.

The NTK can be considered a kernel $K$ that is approximated by random features corresponding to the Jacobian $J$ of the network's output with respect to its parameters, *i.e.* $K(\mathbf{x}_1, \mathbf{x}_2) = J(\mathbf{x}_1)J(\mathbf{x}_2)^\top$. The Jacobian itself naturally decomposes into the Jacobian with respect to $W_1$ and $W_2$, *i.e.* $J(\mathbf{x}) = [\partial N_0(\mathbf{x})/\partial W_1, \partial N_0(\mathbf{x})/\partial W_2] = [J_1(\mathbf{x}), J_2(\mathbf{x})]$. Therefore the kernel $K$ also decomposes this way, and we can write.

$$K(\mathbf{x}_1, \mathbf{x}_2) = J_1(\mathbf{x}_1)J_1(\mathbf{x}_2)^\top + J_2(\mathbf{x}_1)J_2(\mathbf{x}_2)^\top =: K_1(\mathbf{x}_1, \mathbf{x}_2) + K_2(\mathbf{x}_1, \mathbf{x}_2). \tag{S37}$$

As the width of the network becomes very large (compared to all other relevant scales in the system), the approximate NTK converges to a constant kernel determined by the network's initial parameters and describes the trajectory of the network's output under gradient descent.[7] In this work, we focus on the predictive function defined by the solution to this kernel regression problem,

$$\hat{y}(\mathbf{x}) := N_0(\mathbf{x}) + (Y - N_0(X))K^{-1}K_\mathbf{x} \tag{S38}$$

for $K := K(X, X) + \gamma I_m$, $K_\mathbf{x} := K(X, \mathbf{x})$, and $\gamma$ is a ridge regularization constant. A simple calculation yields the per-layer constituent kernels,

$$K_1(\mathbf{x}_1, \mathbf{x}_2) = \frac{X^\top X}{n_0} \odot \frac{(F')^\top \operatorname{diag}(W_2)^2 F'}{n_1} \quad \text{and} \tag{S39}$$

$$K_2(\mathbf{x}_1, \mathbf{x}_2) = \frac{1}{n_1} F^\top F, \tag{S40}$$

where we have introduced the abbreviations $F := \sigma(W_1 X/\sqrt{n_0})$ and $F' := \sigma'(W_1 X/\sqrt{n_0})$. Notice that when $\sigma_{W_2}^2 \to 0$, $K = K_2$, *i.e.* the NTK degenerates into the standard random features kernel of the main text.

**Centering** The predictive function (S38) contains an offset $N_0(\mathbf{x})$ which would typically be set to zero in standard random feature kernel regression because it simply increases the variance of test predictions. Removing this variance component has an analogous operation in neural network training: either the function value at initialization can be subtracted throughout training, or a symmetrization trick can be used in which two copies of the neural network are initialized identically, and their normalized difference $N \equiv \left(N^{(a)} - N^{(b)}\right)/\sqrt{2}$ is trained with gradient descent. Either method preserves the kernel $K$ while enforcing $N_0 \equiv 0$. We call this procedure *centering*, and present results with and without it.

Finally, we note that ridge regularization in the kernel perspective corresponds to using L2 regularization of the neural network's weights toward their initial values.

## S2.3    Exact Asymptotics for the Fine-Grained Variance Decomposition of the NTK

Here we state a generalization of the results from Sec. 3.1 to the NTK. As discussed above, the results for random feature kernel regression follow by setting $\sigma_{W_2} = 0$. The proofs are presented in the subsequent sections.

**High-dimensional asymptotics.**   We consider the limiting behavior of tracial expressions as the dimensions in our model diverge to infinity as their ratios are held fixed according to $\phi$ and $\psi$. The tracial expressions are random variables that converge in probability to deterministic constants, which are specified as the solution to a coupled equation defined below.

**Lemma S3.** *Let $g \sim \mathcal{N}(0, 1)$ and define,*

$$\zeta := (\mathbb{E}\sigma'(g))^2, \quad \eta := \mathbb{E}\sigma(g)^2, \quad \text{and} \quad \eta' := \mathbb{E}\sigma'(g)^2. \tag{S41}$$

*Then, in the high-dimensional asymptotics defined above, the limits of the traces $\tau_1(\gamma) = \frac{1}{m}\mathbb{E}\operatorname{tr}(K^{-1})$ and $\tau_2(\gamma) = \frac{1}{m}\mathbb{E}\operatorname{tr}(\frac{1}{n_0}X^\top X K^{-1})$ converge in probability to the unique solutions to the coupled polynomial equations,*

$$0 = \phi\left(\zeta\tau_2\tau_1 + \phi(\tau_2 - \tau_1)\right) + \zeta\tau_1\tau_2\psi\left(\gamma\tau_1 - 1\right) + \zeta\tau_1\tau_2\sigma_{W_2}^2\left(\zeta\left(\tau_2 - \tau_1\right)\psi + \tau_1\psi\eta' + \phi\right) \tag{S42}$$

$$0 = \zeta\tau_1^2\tau_2\left(\eta' - \eta\right)\sigma_{W_2}^2 + \zeta\tau_1\tau_2\left(\gamma\tau_1 - 1\right) - (\tau_2 - \tau_1)\phi\left(\zeta\left(\tau_2 - \tau_1\right) + \eta\tau_1\right). \tag{S43}$$

*such that $\tau_1, \tau_2 \in \mathbb{C}^+$ for $\gamma \in \mathbb{C}^+$.*

**Corollary S1.** *Lemma 1 follows from Lemma S3 by setting $\sigma_{W_2} = 0$.*

**Theorem S3.** *Let $\tau_1$ and $\tau_2$ be defined as in Lemma S3. Then the asymptotic bias and variance terms of eqns. (9)-(15) for the NTK are given by,*

$$
\begin{aligned}
B &= \tau_2^2/\tau_1^2 & V_{PX} &= -\tau_2'/\tau_1^2 - B - V_P - V_X + \nu T_2/(\gamma\tau_1)^2 \\
V_P &= \tau_2'/\tau_1' - B - \nu T_2/\tau_1' & V_{P\varepsilon} &= 0 \\
V_X &= \phi B(\tau_1 - \tau_2)^2/(\tau_1^2 - \phi(\tau_1 - \tau_2)^2) & V_{X\varepsilon} &= \sigma_\varepsilon^2 V_X/B \\
V_\varepsilon &= 0 & V_{PX\varepsilon} &= \sigma_\varepsilon^2(-\tau_1'/\tau_1^2 - 1) - V_{X\varepsilon},
\end{aligned}
\tag{S44}
$$

*where*

$$T_2 := \sigma_{W_2}^2\gamma^2\left(\tau_1 + (\sigma_{W_2}^2(\eta' - \zeta) + \gamma)\tau_1' + \sigma_{W_2}^2\zeta\tau_2'\right), \tag{S45}$$

*$\tau_i'$ is the derivative of $\tau_i$ with respect to $\gamma$, and $\nu = 0$ with centering and $\nu = 1$ without it.*

**Corollary S2.** *Theorem 1 follows from Theorem S3 by setting $\sigma_{W_2} = 0$.*

## S2.4    Discussion of Results for the NTK

We briefly highlight some results for the full version of Theorem S3 that are distinct from the special case Theorem 1. Note that since the model in eqn. (S38) corresponds to the full NTK, the model has $n_1(n_0 + 1)$ parameters. Thus $n_1 = m$ does not occur at the interpolation threshold but instead represents a significantly overparameterized model. Previous work has found nonmonotonic behavior in the test loss for the model in eqn. (S38) at both the interpolation threshold and when $n_1 = m$ [24].

Figure S2: We replicate Fig. 1 from the main text but using the NTK. As before we set $\gamma = 0$, $\phi = 1/16$ and $\sigma = \tanh$ with SNR $= 100$, and we use centering. Recall that the number of trainable parameters for the NTK is $n_1(n_0 + 1)$, so $n_1 = m$ no longer corresponds to the interpolation threshold but represents very overparameterized models. Despite this we still find nonmonotonic behavior in many of the variance terms. Specifically, $V_P$, $V_{PX}$, and $V_{PX\varepsilon}$ are all nonmonotonic and have a peak slightly before $n_1 = m$. In the semi-classical decomposition (a), these nonmonotonicities would again cause the bias to be nonmonotonic. Similar ambiguities to the random feature case occur for the NTK in (b) and (c). (d) shows the two variable decomposition and (e) the three variable decomposition. As in Fig. 1 the terms $V_\varepsilon$ and $V_{P\varepsilon}$ are zero.

Since $n_1 = m$ is far beyond the interpolation threshold, this second occurrence of nonmonotonicity is qualitatively different than double descent behavior. Our variance decomposition sheds light on the source of this second occurrence of nonmonotonic behavior (see Fig. S2).

We find that none of the variance terms are divergent, but the sources of the nonmonotonicity are $V_P$, $V_{PX}$, and $V_{PX\varepsilon}$. Curiously, bagging this predictive function for a large number of dataset samples would remove all other sources of variance except $V_P$. This would have the effect of highlighting the nonmonotonicity in the total variance.

## S3    Gaussian Equivalents

Here we review the analysis from [24] for computing the test loss in our high-dimensional asymptotic limit. In the next sections, we extend this procedure to compute the constituent bias and variance terms.

As a first step, we exploit some simplifications that happen in our asymptotic limit that allow us to make the following replacements without changing any of the variance terms or the bias:

$$K_1 \to \sigma_{W_2}^2(\eta' - \zeta)I_m + \frac{\sigma_{W_2}^2 \zeta}{n_0}X^\top X \tag{S46}$$

$$F \to \sqrt{\frac{\zeta}{n_0}}W_1 X + \sqrt{\eta - \zeta}\,\Theta_F \tag{S47}$$

$$Y \to \sqrt{\frac{\zeta_{\mathrm{T}}}{n_{\mathrm{T}} n_0}}\omega\Omega X + \sqrt{\frac{\eta_{\mathrm{T}} - \zeta_{\mathrm{T}}}{n_{\mathrm{T}}}}\omega\Theta_Y + \mathcal{E} \tag{S48}$$

$$f \to \sqrt{\frac{\zeta}{n_0}}W_1 \mathbf{x} + \sqrt{\eta - \zeta}\,\theta_f \tag{S49}$$

$$y \to \sqrt{\frac{\zeta_{\mathrm{T}}}{n_{\mathrm{T}} n_0}}\omega\Omega \mathbf{x} + \sqrt{\frac{\eta_{\mathrm{T}} - \zeta_{\mathrm{T}}}{n_{\mathrm{T}}}}\omega\theta_y \,. \tag{S50}$$

where $f := \sigma(W_1 \mathbf{x}/\sqrt{n_0})$ is the random feature representation of the test point $\mathbf{x}$ and $y := \omega \sigma_{\text{T}}(\Omega \mathbf{x}/\sqrt{n_0})/\sqrt{n_{\text{T}}}$ is its label. The new objects $\Theta_F$, $\Theta_Y$, $\theta_f$, and $\theta_y$ are matrices of the appropriate shapes with iid standard Gaussian entries. The constants $\eta'$, $\eta$, and $\zeta$ (see eq. (S41)), as well as $\eta_{\text{T}}$ and $\zeta_{\text{T}}$ (see eqn. (S34)) are chosen so that the mixed moments up to second order are the same for the original and linearized versions.

To give some intuition on these substitutions, many of the statistics of random matrices are universal, that is, their limiting behavior as the matrix gets larger is insensitive to the detailed properties of their entries' distributions. Considerable work has gone into demonstrating universality for an increasingly large class of random matrices and a growing number of detailed statistics. In our case, the test loss is a global measurement of several random matrices. This perspective gives some intuition for why we are able to replace many of the intractable terms in the expressions we analyze with tractable terms, which only need to match quite superficial properties of the distributions to ensure the limiting test loss is the same.

In Secs. S4 and S5, we use this replacement strategy in two distinct situations. The first is for terms of the form

$$\text{tr}(AB) = \sum_{ij} A_{ij} B_{ji}, \tag{S51}$$

for deterministic $A$ and random $B$. Under assumptions on $A$ and $B$, standard concentration inequalities can be used to describe the limiting behavior of sums like eqn. (S51). In our setting, one finds that this behavior only depends on the the low-order moments of $B$. By matching these low-order moments with Gaussian random variables, we can replace $B$ with a Gaussian random matrix with the same limiting behavior. Note, often $A$ is not actually deterministic, we are simply conditioning on it and only considering the randomness in $B$. The approach is suitable for determining the average behavior of eqn. (S51) when we have control over the (weak) correlations in the entries of $A$ and $B$. Linearizing the matrices $A$ and $B$ in this setting is just a convenient bookkeeping device for performing these computations.

When one of the matrices in eqn. (S51) is inverted, the situation is more complex, and indeed this is the case for the kernel matrix $K$ in expressions for the training and test loss. As in [24], to apply the linear pencil algorithm [39, 40], we must first replace the kernels in all expressions with linearized versions (using eqns. (S46)-(S50)), yielding a rational expression of the i.i.d. Gaussian matrices, $X, W_1$, *etc.*

It should be expected that a linearized version of $F$ will lead to the same asymptotic statistics due to some very general results on the limiting behavior of expressions of the form,

$$\text{tr}\left(A \frac{1}{B - zI}\right), \tag{S52}$$

where $A$ is symmetric and $z \in \mathbb{C}^+$. The resolvent matrix $(B - z)^{-1}$ is intimately related to the spectral properties of $B$. Recently, isotropic results for quite general $A$ have been developed for matrices with correlated entries, which show that under certain assumptions the limiting behavior of eqn. (S52) depends only on the low-order moments of $B$. Specifically, the limiting behavior of eqn. (S52) is described by the matrix Dyson equation in many cases. For a summary of these results and related topics see e.g. [41].

Finding Gaussian equivalents for $A$ and $B$ in expressions like eqns. (S51) and (S52) is relatively simple in our case. We encounter terms for which the matrix $B$ depends on some other random matrix $C$ through a coordinate-wise nonlinear function $f(C)$. For such cases, Taylor expanding the function $f$ is the key tool to finding these equivalents (see e.g. [28] for more details on this type of approach).

## S4 Exact asymptotics for the training loss

### S4.1 Decomposition of terms

The model's predictions on the training set, $\hat{y}(X)$, take a simple form,

$$\hat{y}(X) = N_0(X) + (Y - N_0(X))K^{-1}K(X, X) \tag{S53}$$

$$= Y - \gamma(Y - N_0(X))K^{-1}. \tag{S54}$$

The training loss can be written as,

$$E_{\text{train}} = \frac{1}{m}\mathbb{E}_{(X,Y)} \operatorname{tr}\left((Y - \hat{y}(X))(Y - \hat{y}(X))^\top\right) \tag{S55}$$

$$= \frac{\gamma^2}{m}\mathbb{E}_{(X,\varepsilon)} \operatorname{tr}\left((Y - N_0(X))^\top(Y - N_0(X))K^{-2}\right) \tag{S56}$$

$$= T_1 + \nu T_2 \tag{S57}$$

where $\nu = 0$ with centering and $\nu = 1$ without it and,

$$T_1 := \frac{\gamma^2}{m}\mathbb{E}_\varepsilon \operatorname{tr}(Y^\top Y K^{-2}) \tag{S58}$$

$$T_2 := \frac{\gamma^2}{m} \operatorname{tr}(N_0(X)^\top N_0(X)K^{-2}). \tag{S59}$$

We have suppressed the terms linear in $N_0$ since they vanish owing to the linear dependence on the symmetric random variable $W_2$. The Neural Tangent Kernel $K = K(X,X) + \gamma I_m$ and is given by,

$$K = \sigma_{W_2}^2\left[(\eta' - \zeta)I_m + \frac{\zeta X^\top X}{n_0}\right] + \frac{F^\top F}{n_1} + \gamma I_m. \tag{S60}$$

Note that $N_0(X)^\top N_0(X) = \sigma_{W_2}^2/n_1 F^T F$, so eqn. (S60) gives,

$$N_0(X)^\top N_0(X) = \sigma_{W_2}^2 K - \sigma_{W_2}^2\left[\sigma_{W_2}^2(\eta' - \zeta) + \gamma I_m\right] - \sigma_{W_2}^4 \frac{\zeta X^\top X}{n_0}. \tag{S61}$$

Next we recall the substitution (S48) (as mentioned above, without loss of generality we special to the case of a linear teacher),

$$Y \to \sqrt{\frac{1}{n_\mathrm{T} n_0}}\omega\Omega X + \mathcal{E}, \tag{S62}$$

and consider the leading order behavior with respect to the random variables $\omega$, $\Omega$, and $W_2$ using eqn. (S51) to find

$$Y^\top Y = \frac{1}{n_0}X^\top X + \sigma_\varepsilon^2 I_m. \tag{S63}$$

Putting these pieces together, we can write for $\tau_1 = \tau_1(\gamma)$ and $\tau_2 = \tau_2(\gamma)$,

$$T_1 = -\gamma^2(\sigma_\varepsilon^2 \tau_1' + \tau_2') \tag{S64}$$

$$T_2 = \sigma_{W_2}^2 \gamma^2 \left(\tau_1 + (\sigma_{W_2}^2(\eta' - \zeta) + \gamma)\tau_1' + \sigma_{W_2}^2 \zeta \tau_2'\right), \tag{S65}$$

where,

$$\tau_1 = \frac{1}{m}\operatorname{tr}(K^{-1}), \quad \text{and} \quad \tau_2 = \frac{1}{m}\operatorname{tr}(\frac{1}{n_0}X^\top X K^{-1}). \tag{S66}$$

Self-consistent equations for $\tau_1$ and $\tau_2$ can be computed using the resolvent method, as was done in [28] for the case of $\sigma_{W_2} = 0$. In order to pave the way for the analysis of the test error, we instead demonstrate how to compute these traces using operator-valued free probability.

**Remark 2.** *In the remainder of this section, and in Sec. S5, we assume at times that $\sigma$ is non-linear (so that $\eta' > \zeta$ and $\eta > \zeta$) and/or $\gamma > 0$ in order that certain denominator factors are non-zero. The linear and/or ridgeless cases can be obtained by limits of our general results, or through special cases of the pertinent intermediate formulas.*

## S4.2 Linear pencils

To begin, we construct linear pencils for $\tau_1$ and $\tau_2$. Specifically, straightforward block-matrix inversion confirms that

$$\tau_1 = \mathrm{tr}([Q_T^{-1}]_{1,1}) \quad \text{and} \quad \tau_2 = \mathrm{tr}([Q_T^{-1}]_{2,4}) \,, \tag{S67}$$

where,

$$Q_T = \begin{pmatrix} I_m \left( \gamma + \sigma_{W_2}^2 (\eta' - \zeta) \right) & \frac{\zeta X^\top \sigma_{W_2}^2}{n_0} & \frac{\sqrt{\eta - \zeta}\Theta_F^\top}{n_1} & \frac{\sqrt{\zeta} X^\top}{\sqrt{n_0 n_1}} \\ -X & I_{n_0} & 0 & 0 \\ -\sqrt{\eta - \zeta}\Theta_F & -\frac{\sqrt{\zeta} W_1}{\sqrt{n_0}} & I_{n_1} & 0 \\ 0 & 0 & \frac{\sqrt{\zeta}\psi W_1^\top}{\sqrt{n_0}\phi} & -\frac{\sqrt{\zeta}\psi I_{n_0}}{\sqrt{n_0}\phi} \end{pmatrix} . \tag{S68}$$

The matrix $Q_T$ is not self-adjoint, but a self-adjoint representation can be obtained from it by doubling the dimensionality. In particular, letting

$$\bar{Q}_T = \begin{pmatrix} 0 & Q_T^\top \\ Q_T & 0 \end{pmatrix} , \tag{S69}$$

we have,

$$\tau_1 = \mathrm{tr}([\bar{Q}_T^{-1}]_{1,5}) \,, \quad \text{and} \quad \mathrm{tr}([\bar{Q}_T^{-1}]_{2,8}) \,. \tag{S70}$$

Observe that $\bar{Q}_T$ is a self-adjoint matrix whose blocks are either constants or proportional to one of $\{X, X^\top, W_1, W_1^\top, \Theta_F, \Theta_F^\top\}$; let us denote the constant terms as $Z$. As such, we can directly utilize the results of [31, 40] to compute the necessary traces.

## S4.3 Operator-valued Stieltjes transform

The traces can be extracted from the operator-valued Stieltjes transform $G : M_d(\mathbb{C})^+ \to M_d(\mathbb{C})^+$, which is a solution of the equation,

$$ZG = I_d + \eta(G)G \,, \tag{S71}$$

where $d$ is the number of blocks, $\eta : M_d(\mathbb{C}) \to M_d(\mathbb{C})$ defined by

$$[\eta(D)]_{ij} = \sum_{kl} \sigma(i, k; l, j)\alpha_k D_{kl} \,, \tag{S72}$$

where $\alpha_k$ is dimensionality of the $k$th block and $\sigma(i, k; l, k)$ denotes the covariance between the entries of the blocks $ij$ block of $\bar{Q}$ and entries of the $kl$ block of $\bar{Q}$. Eqn. (S71) may admit many solutions, but there is a unique solution such that $\mathrm{Im}\,G \succ 0$ for $\mathrm{Im}\,Z \succ 0$.

The constants $Z$, the entries of $\sigma$, and therefore the equations (S72) are manifest by inspection of the block matrix representation for $\bar{Q}_T$. Although the matrix representation of the equations is too large to reproduce here, we can nevertheless extract the equations satisfied by each entry of $G$.

The equations satisfied by the operator-valued Stieltjes transform $G$ of $\bar{Q}_T$ induce the following structure on $G$,

$$G = \begin{pmatrix} 0 & G_{12} \\ G_{12}^\top & 0 \end{pmatrix} , \tag{S73}$$

where,

$$G_{12} = \begin{pmatrix} \tau_1 & 0 & 0 & 0 \\ 0 & g_3 & 0 & \tau_2 \\ 0 & 0 & g_4 & 0 \\ 0 & g_6 & 0 & g_5 \end{pmatrix} \tag{S74}$$

and the independent entry-wise component functions $g_i$, $\tau_1$ and $\tau_2$ satisfy the following system of polynomial equations,

$$0 = \sqrt{\zeta} g_6 \psi - \zeta g_3 g_4 \sqrt{n_0} \tag{S75}$$

$$0 = \sqrt{\zeta}\psi\big(\tau_2 - g_3\tau_1\big) \tag{S76}$$

$$0 = \sqrt{\zeta}\psi\big(g_5 - g_6\tau_1\big) + \sqrt{n_0}\phi \tag{S77}$$

$$0 = -\zeta g_4 g_5 - g_6\big(\zeta\tau_1\sigma_{W_2}^2 + \phi\big) \tag{S78}$$

$$0 = \sqrt{\zeta}g_5\psi + \sqrt{n_0}\big(\phi - \zeta g_4\tau_2\big) \tag{S79}$$

$$0 = \phi - g_4\big(\tau_1\psi(\eta - \zeta) + \zeta\tau_2\psi + \phi\big) \tag{S80}$$

$$0 = -\zeta g_4\tau_2 - g_3\big(\zeta\tau_1\sigma_{W_2}^2 + \phi\big) + \phi \tag{S81}$$

$$0 = -\sqrt{\zeta}g_5\tau_1\psi - \sqrt{n_0}\tau_2\big(\zeta\tau_1\sigma_{W_2}^2 + \phi\big) \tag{S82}$$

$$0 = \sqrt{n_0}\big(\phi - g_3\big(\zeta\tau_1\sigma_{W_2}^2 + \phi\big)\big) - \sqrt{\zeta}g_6\tau_1\psi \tag{S83}$$

$$0 = \sqrt{n_0}\big(1 - \tau_1\big(\gamma + g_4(\eta - \zeta) + \sigma_{W_2}^2\big(\eta' + \zeta(g_3 - 1)\big)\big)\big) - \sqrt{\zeta}g_6\tau_1\psi\,. \tag{S84}$$

It is straightforward algebra to eliminate $g_3, g_4, g_5$ and $g_6$ from the above equations. A simple set of equations for $\tau_1$ and $\tau_2$ follows,

$$0 = \phi\big(\zeta\tau_2\tau_1 + \phi(\tau_2 - \tau_1)\big) + \zeta\tau_1\tau_2\psi\big(\gamma\tau_1 - 1\big) + \zeta\tau_1\tau_2\sigma_{W_2}^2\big(\zeta\big(\tau_2 - \tau_1\big)\psi + \tau_1\psi\eta' + \phi\big) \tag{S85}$$

$$0 = \zeta\tau_1^2\tau_2\big(\eta' - \eta\big)\sigma_{W_2}^2 + \zeta\tau_1\tau_2\big(\gamma\tau_1 - 1\big) - \big(\tau_2 - \tau_1\big)\phi\big(\zeta\big(\tau_2 - \tau_1\big) + \eta\tau_1\big)\,. \tag{S86}$$

Although these equations admit multiple solutions, the general results of [31, 40] guarantee that the correct root is given by the unique solutions $\tau_1, \tau_2 : \mathbb{C}^+ \to \mathbb{C}^+$ which are analytic in the upper half-plane.

It will prove useful to obtain expressions for $\tau_1'(\gamma)$ and $\tau_2'(\gamma)$. By differentiating eqns. (S85) and (S86) with respect to $\gamma$, we find

$$\tau_1' = -\frac{\zeta^2\tau_2^2\big(\psi\tilde{\tau}_1^2 - \phi^2\big)}{\psi\tilde{\tau}_1^2\big(\zeta^2\big(\tilde{\tau}_2 + 1\big)^2 + \phi\big(\zeta\tilde{\tau}_2 + \eta\big)\big(\zeta\tilde{\tau}_2\big(2\tilde{\tau}_2 + 3\big) + \eta\big)\big) + \zeta^2\phi^2\big(\tilde{\tau}_2 + 1\big)^2\big(\phi\tilde{\tau}_2^2 - 1\big)} \tag{S87}$$

$$\tau_2' = -\frac{\zeta\tau_2^2\big(\psi\tilde{\tau}_1^2(\zeta - \eta) - \zeta\phi^2\big(\tilde{\tau}_2 + 1\big)^2\big)}{\psi\tilde{\tau}_1^2\big(\zeta^2\big(\tilde{\tau}_2 + 1\big)^2 + \phi\big(\zeta\tilde{\tau}_2 + \eta\big)\big(\zeta\tilde{\tau}_2\big(2\tilde{\tau}_2 + 3\big) + \eta\big)\big) + \zeta^2\phi^2\big(\tilde{\tau}_2 + 1\big)^2\big(\phi\tilde{\tau}_2^2 - 1\big)}\,, \tag{S88}$$

where we have introduced some auxiliary variables to ease the presentation,

$$\tilde{\tau}_1 = \sigma_{W_2}^2\zeta\tau_2 + \phi\tilde{\tau}_2 \quad \text{and} \quad \tilde{\tau}_2 = -1 + \tau_2/\tau_1\,. \tag{S89}$$

## S5  Exact asymptotics for the test loss

### S5.1  Decomposition of terms

The test loss can be written as,

$$E_{\text{test}} = \mathbb{E}_{(\mathbf{x},y)}(y - \hat{y}(\mathbf{x}))^2 = E_1 + E_2 + E_3 \tag{S90}$$

with

$$E_1 = \mathbb{E}_{(\mathbf{x},\varepsilon)}\operatorname{tr}(y(\mathbf{x})y(\mathbf{x})^\top) + \mathbb{E}_{(\mathbf{x},\varepsilon)}\operatorname{tr}(N_0(\mathbf{x})N_0(\mathbf{x})^\top) \tag{S91}$$

$$E_2 = -2\mathbb{E}_{(\mathbf{x},\varepsilon)}\operatorname{tr}(K_\mathbf{x}^\top K^{-1}Y^\top y(\mathbf{x})) - 2\mathbb{E}_{(\mathbf{x},\varepsilon)}\operatorname{tr}(K_\mathbf{x}^\top K^{-1}N_0(X)^\top N_0(\mathbf{x})) \tag{S92}$$

$$E_3 = \mathbb{E}_{(\mathbf{x},\varepsilon)}\operatorname{tr}(K_\mathbf{x}^\top K^{-1}Y^\top Y K^{-1}K_\mathbf{x}) + \mathbb{E}_{(\mathbf{x},\varepsilon)}\operatorname{tr}(K_\mathbf{x}^\top K^{-1}N_0(X)^\top N_0(X)K^{-1}K_\mathbf{x})\,, \tag{S93}$$

where we have suppressed the terms linear in $N_0$ since they vanish owing to the linear dependence on the symmetric random variable $W_2$. The Neural Tangent Kernels $K = K(X, X)$ and $K_\mathbf{x} = K(X, \mathbf{x})$ are given by,

$$K = \sigma_{W_2}^2\left[(\eta' - \zeta)I_m + \frac{\zeta X^\top X}{n_0}\right] + \frac{F^\top F}{n_1} + \gamma I_m \quad \text{and} \quad K_\mathbf{x} = \frac{\sigma_{W_2}^2\zeta}{n_0}X^\top\mathbf{x} + \frac{1}{n_1}F^\top f\,. \tag{S94}$$

**Remark 3.** *In eqn. (S46), we argued that the leading order behavior (all that is relevant for the test loss) of $K_1$ is relatively simple, leading to the expression for $K$ in eqn. (S94). Implicitly this requires that $\eta' \neq \zeta$, and similarly, in many of the expressions denominators are assumed to be nonzero. We handle degenerate expressions of this kind as special cases, but avoid details here to streamline the presentation.*

Using the cyclicity and linearity of the trace, the expectation over $\mathbf{x}$ requires the computation of

$$\mathbb{E}_{\mathbf{x}} K_{\mathbf{x}} K_{\mathbf{x}}^\top, \qquad \mathbb{E}_{\mathbf{x}} y(\mathbf{x}) K_{\mathbf{x}}^\top, \qquad \mathbb{E}_{\mathbf{x}} y(\mathbf{x}) y(\mathbf{x})^\top, \qquad \mathbb{E}_{\mathbf{x}} N_0(\mathbf{x}) K_{\mathbf{x}}^\top, \qquad \text{and} \quad \mathbb{E}_{\mathbf{x}} N_0(\mathbf{x}) N_0(\mathbf{x})^\top. \quad (S95)$$

As described in Sec. S3, without loss of generality we can consider the case of a linear teacher, so that $\eta_{\mathrm{T}} = \zeta_{\mathrm{T}} = 1$ and (S50) and (S49) become

$$y \to y^{\mathrm{lin}} = \frac{\sqrt{\zeta_{\mathrm{T}}}}{\sqrt{n_0 n_{\mathrm{T}}}} \omega \Omega \mathbf{x} + \sqrt{\eta_{\mathrm{T}} - \zeta_{\mathrm{T}}} \frac{1}{\sqrt{n_{\mathrm{T}}}} \omega \theta_y = \frac{1}{\sqrt{n_0 n_{\mathrm{T}}}} \omega \Omega \mathbf{x} \qquad \text{and} \qquad f \to f^{\mathrm{lin}} = \frac{\sqrt{\zeta}}{\sqrt{n_0}} W_1 \mathbf{x} + \sqrt{\eta - \zeta} \theta_f. \quad (S96)$$

Using these substitutions, the expectations over $\mathbf{x}$ are now trivial and we readily find,

$$\mathbb{E}_{\mathbf{x}} K_{\mathbf{x}} K_{\mathbf{x}}^\top = \frac{\sigma_{W_2}^4 \zeta^2}{n_0^2} X^\top X + \frac{\sigma_{W_2}^2 \zeta^{3/2}}{n_0^{3/2} n_1} (X^\top W_1^T F + F^\top W_1 X) + \frac{1}{n_1^2} F^\top \left( \frac{\zeta}{n_0} W_1 W_1^\top + (\eta - \zeta) I_{n_1} \right) F \quad (S97)$$

$$\mathbb{E}_{\mathbf{x}} y(\mathbf{x}) K_{\mathbf{x}}^\top = \frac{\sigma_{W_2}^2 \zeta}{n_0^{3/2} \sqrt{n_{\mathrm{T}}}} \omega \Omega X + \frac{\sqrt{\zeta}}{n_0 n_1 \sqrt{n_{\mathrm{T}}}} \omega \Omega W_1^\top F \quad (S98)$$

$$\mathbb{E}_{\mathbf{x}} y(\mathbf{x}) y(\mathbf{x})^\top = \frac{1}{n_0 n_{\mathrm{T}}} \omega \Omega \Omega^\top \omega^\top \quad (S99)$$

$$\mathbb{E}_{\mathbf{x}} N_0(\mathbf{x}) K_{\mathbf{x}}^\top = \frac{\sigma_{W_2}^2 \zeta^{3/2}}{n_0^{3/2} \sqrt{n_1}} W_2 W_1 X + \frac{1}{n_1^{3/2}} W_2 \left( \frac{\zeta}{n_0} W_1 W_1^\top + (\eta - \zeta) I_{n_1} \right) F \quad (S100)$$

$$\mathbb{E}_{\mathbf{x}} \operatorname{tr}(N_0(\mathbf{x}) N_0(\mathbf{x})^\top) = \sigma_{W_2}^2 \eta. \quad (S101)$$

One may interpret the substitutions in eqn. (S96) as a tool to calculate the expectations above to leading order as it leads to terms like eqn. (S51). Next we recall the substitution (S62),

$$Y \to \frac{1}{\sqrt{n_0 n_{\mathrm{T}}}} \omega \Omega X + \mathcal{E}. \quad (S102)$$

As above, we consider the leading order behavior with respect to the random variables $\omega$, $\Omega$, and $W_2$ using eqn. (S51) to find

$$\mathbb{E}_{\omega, \Omega, \mathcal{E}} \left[ Y^\top Y \right] = \frac{1}{n_0} X^\top X + \sigma_\varepsilon^2 I_m \quad (S103)$$

$$\mathbb{E}_{\omega, \Omega, \mathcal{E}, W_2} \left[ Y^\top \mathbb{E}_{\mathbf{x}} y(\mathbf{x}) K_{\mathbf{x}}^\top \right] = \frac{\sigma_{W_2}^2 \zeta}{n_0^2} X^\top X + \frac{\sqrt{\zeta}}{n_0^{3/2} n_1} X^\top W_1^\top F \quad (S104)$$

$$\mathbb{E}_{W_2} \left[ N_0(X)^\top N_0(X) \right] = \frac{\sigma_{W_2}^2}{n_1} F^\top F \quad (S105)$$

$$\mathbb{E}_{W_2} \left[ N_0(X)^\top \mathbb{E}_{\mathbf{x}} N_0(\mathbf{x}) K_{\mathbf{x}}^\top \right] = \frac{\sigma_{W_2}^4 \zeta^{3/2}}{n_0^{3/2} n_1} F^\top W_1 X + \frac{\sigma_{W_2}^2}{n_1^2} F^\top \left( \frac{\zeta}{n_0} W_1 W_1^\top + (\eta - \zeta) I_{n_1} \right) F. \quad (S106)$$

$$F \to F^{\mathrm{lin}} = \frac{\sqrt{\zeta}}{\sqrt{n_0}} W_1 X + \sqrt{\eta - \zeta} \Theta_F, \quad (S107)$$

we can write,

$$\frac{\sqrt{\zeta}}{\sqrt{n_0}} F^\top W_1 X + \frac{\sqrt{\zeta}}{\sqrt{n_0}} X^\top W_1^\top F = F^\top F + \frac{\zeta}{n_0} X^\top W_1^\top W_1 X - (\eta - \zeta) \Theta_F^\top \Theta_F. \quad (S108)$$

Putting these pieces together, we have

$$E_1 = 1 + \nu \sigma_{W_2}^2 \eta \tag{S109}$$

$$E_2 = E_{21} + \nu E_{22} \tag{S110}$$

$$E_3 = E_{31} + E_{32} + \nu E_{33} \,, \tag{S111}$$

where $\nu = 0$ with centering and $\nu = 1$ without it,

$$E_{21} = -\mathbb{E} \operatorname{tr} \left( 2 \frac{\sigma_{W_2}^2 \zeta}{n_0^2} X K^{-1} X^\top + \frac{1}{n_0 n_1} F K^{-1} F^\top + \frac{\zeta}{n_0^2 n_1} W_1 X K^{-1} X^\top W_1^\top - \frac{\eta - \zeta}{n_0 n_1} \Theta_F K^{-1} \Theta_F^\top \right) \tag{S112}$$

$$E_{22} = -\frac{2\sigma_{W_2}^2}{n_1} \mathbb{E} \operatorname{tr} \left( \frac{\sigma_{W_2}^2 \zeta^{3/2}}{n_0^{3/2}} K^{-1} F^\top W_1 X + \frac{\zeta}{n_0 n_1} K^{-1} F^\top W_1 W_1^\top F + \frac{\eta - \zeta}{n_1} K^{-1} F^\top F \right) \tag{S113}$$

$$E_{31} = \sigma_\varepsilon^2 \mathbb{E} \operatorname{tr} \left( K^{-1} \Sigma_3 K^{-1} \right) \tag{S114}$$

$$E_{32} = \frac{1}{n_0} \mathbb{E} \operatorname{tr} \left( X K^{-1} \Sigma_3 K^{-1} X^\top \right) \tag{S115}$$

$$E_{33} = \frac{\sigma_{W_2}^2}{n_1} \mathbb{E} \operatorname{tr} \left( F K^{-1} \Sigma_3 K^{-1} F^\top \right) \,, \tag{S116}$$

and,

$$\Sigma_3 = \frac{\sigma_{W_2}^4 \zeta^2}{n_0^2} X^\top X + \left( \frac{\sigma_{W_2}^2 \zeta}{n_0 n_1} + \frac{\eta - \zeta}{n_1^2} \right) F^\top F + \frac{\zeta}{n_0 n_1^2} F^\top W_1 W_1^\top F + \frac{\sigma_{W_2}^2 \zeta^2}{n_0^2 n_1} X^\top W_1^\top W_1 X - \frac{\sigma_{W_2}^2 \zeta (\eta - \zeta)}{n_0 n_1} \Theta_F^\top \Theta_F \,. \tag{S117}$$

## S5.2 Linear pencils

Repeated application of the Schur complement formula for block matrix inversion establishes the following representations for $E_{21}, E_{22}, E_{31}, E_{32}, E_{33}$.

### S5.2.1 $E_{21}$

A linear pencil for $E_{21}$ follows from the representation,

$$E_{21} = \operatorname{tr}(U_{21}^T Q_{21}^{-1} V_{21}) \,, \tag{S118}$$

where,

$$U_{21}^T = \begin{pmatrix} 0 & -\frac{2\zeta I_{n_0} \sigma_{W_2}^2}{n_0} & 0 & 0 & 0 & \frac{(\eta - \zeta) I_{n_1}}{n_0} & 0 & 0 & 0 & 0 & 0 & -\frac{I_{n_1}}{n_0} & 0 & 0 \end{pmatrix} \tag{S119}$$

$$V_{21}^T = \begin{pmatrix} 0 & 0 & 0 & -\frac{\sqrt{n_0} n_1 I_{n_0}}{\sqrt{\zeta}} & 0 & 0 & 0 & 0 & 0 & I_{n_1} & 0 & 0 & 0 & 0 \end{pmatrix} \tag{S120}$$

and,

$$Q_{21} = \begin{pmatrix} Q_{21}^{11} & 0 & 0 \\ 0 & Q_{21}^{22} & Q_{21}^{23} \\ 0 & 0 & Q_{21}^{33} \end{pmatrix} \tag{S121}$$

with,

$$Q_{21}^{11} = \begin{pmatrix} I_m \left( \gamma + \sigma_{W_2}^2 (\eta' - \zeta) \right) & \frac{\zeta X^\top \sigma_{W_2}^2}{n_0} & \frac{\sqrt{\eta - \zeta} \Theta_F^\top}{n_1} & \frac{\sqrt{\zeta} X^\top}{\sqrt{n_0 n_1}} \\ -X & I_{n_0} & 0 & 0 \\ -\sqrt{\eta - \zeta} \Theta_F & -\frac{\sqrt{\zeta} W_1}{\sqrt{n_0}} & I_{n_1} & 0 \\ 0 & 0 & -W_1^\top & I_{n_0} \end{pmatrix} \tag{S122}$$

$$Q_{21}^{22} = \begin{pmatrix} I_m\left(\gamma + \sigma_{W_2}^2\left(\eta' - \zeta\right)\right) & 0 & \frac{\zeta X^\top \sigma_{W_2}^2}{n_0} & \frac{\sqrt{\eta-\zeta}\Theta_F^\top}{n_1} & \frac{\sqrt{\zeta}X^\top}{\sqrt{n_0 n_1}} \\ -\Theta_F & I_{n_1} & -\frac{\sqrt{\zeta}W_1}{\sqrt{n_0}\sqrt{\eta-\zeta}} & 0 & 0 \\ -X & 0 & I_{n_0} & 0 & 0 \\ -\sqrt{\eta-\zeta}\Theta_F & 0 & -\frac{\sqrt{\zeta}W_1}{\sqrt{n_0}} & I_{n_1} & \\ 0 & 0 & 0 & -W_1^\top & I_{n_0} \end{pmatrix} \tag{S123}$$

$$Q_{21}^{23} = \begin{pmatrix} -\Theta_F^\top & 0 & 0 & 0 & 0 \\ 0 & 0 & 0 & \frac{\sqrt{\zeta}W_1}{\sqrt{n_0}(\eta-\zeta)} & 0 \\ 0 & 0 & 0 & 0 & 0 \\ 0 & 0 & 0 & 0 & 0 \\ 0 & 0 & 0 & 0 & 0 \\ I_{n_1} & 0 & 0 & 0 & 0 \end{pmatrix} \tag{S124}$$

$$Q_{21}^{33} = \begin{pmatrix} -\sqrt{\eta-\zeta}\Theta_F^\top & I_m\left(\gamma + \sigma_{W_2}^2\left(\eta' - \zeta\right)\right) & \frac{\sqrt{\eta-\zeta}\Theta_F^\top}{n_1} & \frac{\zeta X^\top \sigma_{W_2}^2}{n_0} & \frac{\sqrt{\zeta}X^\top}{\sqrt{n_0 n_1}} \\ 0 & -\sqrt{\eta-\zeta}\Theta_F & I_{n_1} & -\frac{\sqrt{\zeta}W_1}{\sqrt{n_0}} & 0 \\ 0 & -X & 0 & I_{n_0} & 0 \\ n_1 W_1^\top & 0 & -W_1^\top & 0 & I_{n_0} \end{pmatrix}. \tag{S125}$$

### S5.2.2 $E_{22}$

A linear pencil for $E_{22}$ follows from the representation,

$$E_{22} = \mathrm{tr}(U_{22}^T Q_{22}^{-1} V_{22}), \tag{S126}$$

where,

$$U_{22}^T = \begin{pmatrix} 0 & -\frac{2\sqrt{\zeta}I_{n_1}\sigma_{W_2}^2\left(n_0(\eta-\zeta)+\zeta n_1 \sigma_{W_2}^2\right)}{n_0^{3/2} n_1} & 0 & \frac{2(\zeta-\eta)I_{n_1}\sigma_{W_2}^2}{n_1} & 0 & 0 & 0 \end{pmatrix} \tag{S127}$$

$$V_{22}^T = \begin{pmatrix} 0 & 0 & 0 & 0 & 0 & -n_1 I_{n_1} & 0 \end{pmatrix} \tag{S128}$$

and,

$$Q_{22} = \begin{pmatrix} I_{n_0} & 0 & -X & 0 & 0 & 0 & 0 \\ -W_1 & I_{n_1} & 0 & 0 & -\frac{\sqrt{n_0}W_1}{\sqrt{\zeta}n_1\sigma_{W_2}^2} & 0 & 0 \\ \frac{\zeta X^\top \sigma_{W_2}^2}{n_0} & 0 & I_m\left(\gamma+\sigma_{W_2}^2(\eta'-\zeta)\right) & 0 & 0 & \frac{\sqrt{\eta-\zeta}\Theta_F^\top}{n_1} & \frac{\sqrt{\zeta}X^\top}{\sqrt{n_0 n_1}} \\ 0 & 0 & -\sqrt{\eta-\zeta}\Theta_F & I_{n_1} & \frac{W_1}{n_1\sigma_{W_2}^2} & 0 & 0 \\ 0 & -\frac{\sqrt{\zeta}W_1^\top}{\sqrt{n_0}} & 0 & -W_1^\top & I_{n_0} & 0 & 0 \\ -\frac{\sqrt{\zeta}W_1}{\sqrt{n_0}} & 0 & -\sqrt{\eta-\zeta}\Theta_F & 0 & 0 & I_{n_1} & 0 \\ 0 & 0 & 0 & 0 & 0 & -W_1^\top & I_{n_0} \end{pmatrix}. \tag{S129}$$

### S5.2.3 $E_{31}$

A linear pencil for $E_{31}$ follows from the representation,

$$E_{31} = \mathrm{tr}(U_{31}^T Q_{31}^{-1} V_{31}), \tag{S130}$$

where,

$$U_{31}^T = \begin{pmatrix} m\sigma_\varepsilon^2 I_m & 0 & 0 & 0 & 0 & 0 & 0 & 0 \end{pmatrix}, \quad V_{31}^T = \begin{pmatrix} 0 & 0 & 0 & 0 & 0 & I_m & 0 & 0 \end{pmatrix} \tag{S131}$$

and, for $\beta = \left(n_0(\zeta - \eta) - \zeta n_1 \sigma_{W_2}^2\right)$,

$$Q_{31} = \begin{pmatrix} I_m\left(\gamma+\sigma_{W_2}^2\left(\eta'-\zeta\right)\right) & \frac{\zeta X^\top \sigma_{W_2}^2}{n_0} & \frac{\sqrt{\eta-\zeta}\Theta_F^\top}{n_1} & \frac{\sqrt{\zeta}X^\top}{\sqrt{n_0}n_1} & -\frac{\zeta^2 X^\top \sigma_{W_2}^4}{n_0^2} & 0 & \frac{\sqrt{\eta-\zeta}\Theta_F^\top \beta}{n_0 n_1^2} & \frac{\sqrt{\zeta}X^\top \beta}{n_0^{3/2}n_1^2} \\ -X & I_{n_0} & 0 & 0 & 0 & 0 & 0 & 0 \\ -\sqrt{\eta-\zeta}\Theta_F & -\frac{\sqrt{\zeta}W_1}{\sqrt{n_0}} & I_{n_1} & 0 & 0 & -\frac{\zeta\sqrt{\eta-\zeta}\Theta_F \sigma_{W_2}^2}{n_0} & 0 & \frac{\zeta W_1}{n_0 n_1} \\ 0 & 0 & -W_1^\top & I_{n_0} & 0 & 0 & \frac{\zeta W_1^\top \sigma_{W_2}^2}{n_0} & 0 \\ 0 & 0 & 0 & 0 & I_{n_0} & -X & 0 & 0 \\ 0 & 0 & 0 & 0 & \frac{\zeta X^\top \sigma_{W_2}^2}{n_0} & I_m\left(\gamma+\sigma_{W_2}^2\left(\eta'-\zeta\right)\right) & \frac{\sqrt{\eta-\zeta}\Theta_F^\top}{n_1} & \frac{\sqrt{\zeta}X^\top}{\sqrt{n_0}n_1} \\ 0 & 0 & 0 & 0 & -\frac{\sqrt{\zeta}W_1}{\sqrt{n_0}} & -\sqrt{\eta-\zeta}\Theta_F & I_{n_1} & 0 \\ 0 & 0 & 0 & 0 & 0 & 0 & -W_1^\top & I_{n_0} \end{pmatrix}. \tag{S132}$$

## S5.2.4 $E_{32}$

A linear pencil for $E_{32}$ follows from the representation,

$$E_{32} = \mathrm{tr}(U_{32}^T Q_{32}^{-1} V_{32}), \tag{S133}$$

where,

$$U_{32}^T = \begin{pmatrix} 0 & I_{n_0} & 0 & 0 & 0 & 0 & 0 & 0 & 0 \end{pmatrix}, \quad V_{32}^T = \begin{pmatrix} 0 & 0 & 0 & 0 & 0 & 0 & 0 & 0 & -\frac{\sqrt{n_0}n_1 I_{n_0}}{\sqrt{\zeta}} \end{pmatrix} \tag{S134}$$

and, for $\beta = \left(n_0(\zeta - \eta) - \zeta n_1 \sigma_{W_2}^2\right)$

$$Q_{32} = \begin{pmatrix} I_m\left(\gamma+\sigma_{W_2}^2\left(\eta'-\zeta\right)\right) & 0 & \frac{\zeta X^\top \sigma_{W_2}^2}{n_0} & \frac{\sqrt{\eta-\zeta}\Theta_F^\top}{n_1} & \frac{\sqrt{\zeta}X^\top}{\sqrt{n_0}n_1} & -\frac{\zeta^2 X^\top \sigma_{W_2}^4}{n_0^2} & 0 & \frac{\sqrt{\eta-\zeta}\Theta_F^\top \beta}{n_0 n_1^2} & 0 \\ -X & I_{n_0} & 0 & 0 & 0 & 0 & 0 & 0 & 0 \\ -X & 0 & I_{n_0} & 0 & 0 & 0 & 0 & \frac{\sqrt{\zeta}W_1^\top}{\sqrt{n_0}n_1} & 0 \\ -\sqrt{\eta-\zeta}\Theta_F & 0 & -\frac{\sqrt{\zeta}W_1}{\sqrt{n_0}} & I_{n_1} & 0 & 0 & -\frac{\zeta\sqrt{\eta-\zeta}\Theta_F \sigma_{W_2}^2}{n_0} & 0 & 0 \\ 0 & 0 & 0 & -W_1^\top & I_{n_0} & 0 & 0 & W_1^\top\left(\frac{\eta-\zeta}{n_1}+\frac{\zeta\sigma_{W_2}^2}{n_0}\right) & 0 \\ 0 & 0 & 0 & 0 & 0 & I_{n_0} & -X & 0 & 0 \\ 0 & 0 & 0 & 0 & 0 & \frac{\zeta X^\top \sigma_{W_2}^2}{n_0} & I_m\left(\gamma+\sigma_{W_2}^2\left(\eta'-\zeta\right)\right) & \frac{\sqrt{\eta-\zeta}\Theta_F^\top}{n_1} & \frac{\sqrt{\zeta}X^\top}{\sqrt{n_0}n_1} \\ 0 & 0 & 0 & 0 & 0 & -\frac{\sqrt{\zeta}W_1}{\sqrt{n_0}} & -\sqrt{\eta-\zeta}\Theta_F & I_{n_1} & 0 \\ 0 & 0 & 0 & 0 & 0 & 0 & 0 & -W_1^\top & I_{n_0} \end{pmatrix}. \tag{S135}$$

## S5.2.5 $E_{33}$

A linear pencil for $E_{33}$ follows from the representation,

$$E_{33} = \mathrm{tr}(U_{33}^T Q_{33}^{-1} V_{33}), \tag{S136}$$

where,

$$U_{33}^T = \begin{pmatrix} 0 & I_{n_1}\sigma_{W_2}^2 & 0 & 0 & 0 & 0 & 0 & 0 & 0 & 0 \end{pmatrix} \tag{S137}$$

$$V_{33}^T = \begin{pmatrix} 0 & 0 & 0 & 0 & 0 & 0 & 0 & 0 & 0 & -n_1 I_{n_1} & 0 \end{pmatrix} \tag{S138}$$

and, for $\beta = \left(n_0(\zeta - \eta) - \zeta n_1 \sigma_{W_2}^2\right)$,

$$Q_{33} = \begin{pmatrix}
I_m\left(\gamma + \sigma_{W_2}^2(\eta' - \zeta)\right) & 0 & 0 & \frac{\zeta X^\top \sigma_{W_2}^2}{n_0} & \frac{\sqrt{\eta-\zeta}\Theta_F^\top}{n_1} & \frac{\sqrt{\zeta}X^\top}{\sqrt{n_0}n_1} & -\frac{\zeta^2 X^\top \sigma_{W_2}^4}{n_0^2} & 0 & \frac{\sqrt{\eta-\zeta}\Theta_F^\top \beta}{n_0 n_1^2} & 0 & 0 \\
-\sqrt{\eta-\zeta}\Theta_F & I_{n_1} & -\frac{\sqrt{\zeta}W_1}{\sqrt{n_0}} & 0 & 0 & 0 & 0 & 0 & 0 & 0 & 0 \\
-X & 0 & I_{n_0} & 0 & 0 & 0 & 0 & 0 & 0 & 0 & 0 \\
-X & 0 & 0 & I_{n_0} & 0 & 0 & 0 & 0 & \frac{\sqrt{\zeta}W_1^\top}{\sqrt{n_0}n_1} & 0 & 0 \\
-\sqrt{\eta-\zeta}\Theta_F & 0 & 0 & -\frac{\sqrt{\zeta}W_1}{\sqrt{n_0}} & I_{n_1} & 0 & 0 & -\frac{\zeta\sqrt{\eta-\zeta}\Theta_F\sigma_{W_2}^2}{n_0} & 0 & 0 & 0 \\
0 & 0 & 0 & 0 & -W_1^\top & I_{n_0} & 0 & 0 & W_1^\top\left(\frac{\eta-\zeta}{n_1}+\frac{\zeta\sigma_{W_2}^2}{n_0}\right) & 0 & 0 \\
0 & 0 & 0 & 0 & 0 & 0 & I_{n_0} & -X & 0 & 0 & 0 \\
0 & 0 & 0 & 0 & 0 & 0 & \frac{\zeta X^\top \sigma_{W_2}^2}{n_0} & I_m\left(\gamma + \sigma_{W_2}^2(\eta' - \zeta)\right) & 0 & \frac{\sqrt{\eta-\zeta}\Theta_F^\top}{n_1} & \frac{\sqrt{\zeta}X^\top}{\sqrt{n_0}n_1} \\
0 & 0 & 0 & 0 & 0 & 0 & -\frac{\sqrt{\zeta}W_1}{\sqrt{n_0}} & -\sqrt{\eta-\zeta}\Theta_F & I_{n_1} & 0 & 0 \\
0 & 0 & 0 & 0 & 0 & 0 & -\frac{\sqrt{\zeta}W_1}{\sqrt{n_0}} & -\sqrt{\eta-\zeta}\Theta_F & 0 & I_{n_1} & 0 \\
0 & 0 & 0 & 0 & 0 & 0 & 0 & 0 & 0 & -W_1^\top & I_{n_0}
\end{pmatrix}.$$

$$\text{(S139)}$$

## S5.3 Operator-valued Stieltjes transform

Even though the individual error terms $E_{21}, E_{22}, E_{31}, E_{32}, E_{33}$ can be written as the trace of self-adjoint matrices, the individual $Q$ matrices are not themselves self-adjoint. However, by enlarging the dimensionality by a factor of two, equivalent self-adjoint representations can easily be constructed. To do so, we simply utilize the identity,

$$U^T Q V = \bar{U}^\top \bar{Q} \bar{V} \equiv \left(\tfrac{1}{2}U^\top \quad V^\top\right)\begin{pmatrix} 0 & Q^\top \\ Q & 0 \end{pmatrix}\begin{pmatrix} \tfrac{1}{2}U \\ V \end{pmatrix}. \tag{S140}$$

Observe that $\bar{Q}_{21}, \bar{Q}_{22}, \bar{Q}_{31}, \bar{Q}_{32}$ and $\bar{Q}_{33}$ are all self-adjoint block matrices whose blocks are either constants or proportional to one of $\{X, X^\top, W_1, W_1^\top, \Theta_F, \Theta_F^\top\}$; let us denote the constant terms as $Z$. As such, we can directly utilize the results of [31, 40] to compute the error terms in question.

For each linear pencil, the corresponding error term can be extracted from the operator-valued Stieltjes transform $G : M_d(\mathbb{C})^+ \to M_d(\mathbb{C})^+$, which is a solution of the equation,

$$ZG = I_d + \eta(G)G, \tag{S141}$$

where $d$ is the number of blocks, $\eta : M_d(\mathbb{C}) \to M_d(\mathbb{C})$ defined by

$$[\eta(D)]_{ij} = \sum_{kl} \sigma(i,k;l,j)\alpha_k D_{kl}, \tag{S142}$$

where $\alpha_k$ is dimensionality of the $k$th block and $\sigma(i,k;l,k)$ denotes the covariance between the entries of the $ij$ block of $\bar{Q}$ and entries of the $kl$ block of $\bar{Q}$. Eqn. (S141) may admit many solutions, but there is a unique solution such that $\text{Im}G \succ 0$ for $\text{Im}Z \succ 0$.

The constants $Z$, the entries of $\sigma$, and therefore the equations (S142) are manifest by inspection of the block matrix representations for $Q$. Although the matrix representations are too large to reproduce here, we can nevertheless extract the equations satisfied by each entry of $G$, which we present in the subsequent sections.

### S5.3.1 $E_{21}$

The equations satisfied by the operator-valued Stieltjes transform $G$ of $\bar{Q}_{21}$ induce the following structure on $G$,

$$G = \begin{pmatrix} 0 & G_{12} \\ G_{12}^\top & 0 \end{pmatrix}, \tag{S143}$$

where,

$$
G_{12} = \begin{pmatrix}
g_8 & 0 & 0 & 0 & 0 & 0 & 0 & 0 & 0 & 0 & 0 & 0 & 0 & 0 \\
0 & g_9 & 0 & g_6 & 0 & 0 & 0 & 0 & 0 & 0 & 0 & 0 & 0 & 0 \\
0 & 0 & g_{11} & 0 & 0 & 0 & 0 & 0 & 0 & 0 & 0 & 0 & 0 & 0 \\
0 & g_{12} & 0 & g_{10} & 0 & 0 & 0 & 0 & 0 & 0 & 0 & 0 & 0 & 0 \\
0 & 0 & 0 & 0 & g_8 & 0 & 0 & 0 & 0 & 0 & 0 & 0 & 0 & 0 \\
0 & 0 & 0 & 0 & 0 & g_1 & 0 & g_5 & 0 & g_4 & 0 & g_7 & 0 & 0 \\
0 & 0 & 0 & 0 & 0 & 0 & g_9 & 0 & g_6 & 0 & 0 & 0 & 0 & 0 \\
0 & 0 & 0 & 0 & 0 & 0 & 0 & g_{11} & 0 & g_3 & 0 & 0 & 0 & 0 \\
0 & 0 & 0 & 0 & 0 & 0 & g_{12} & 0 & g_{10} & 0 & 0 & 0 & 0 & 0 \\
0 & 0 & 0 & 0 & 0 & 0 & 0 & 0 & 0 & g_1 & 0 & 0 & 0 & 0 \\
0 & 0 & 0 & 0 & 0 & 0 & 0 & 0 & 0 & 0 & g_8 & 0 & 0 & 0 \\
0 & 0 & 0 & 0 & 0 & 0 & 0 & 0 & 0 & g_2 & 0 & g_{11} & 0 & 0 \\
0 & 0 & 0 & 0 & 0 & 0 & 0 & 0 & 0 & 0 & 0 & 0 & g_9 & g_6 \\
0 & 0 & 0 & 0 & 0 & 0 & 0 & 0 & 0 & 0 & 0 & 0 & g_{12} & g_{10}
\end{pmatrix}, \tag{S144}
$$

and the independent entry-wise component functions $g_i$ combine to produce the error $E_{21}$ through the relation,

$$
E_{21} = \frac{g_4(\eta - \zeta)}{n_0} + \frac{2\sqrt{\zeta}\, g_6 \sqrt{n_0}\, \sigma_{W_2}^2}{\psi} - \frac{g_2}{n_0}, \tag{S145}
$$

and themselves satisfy the following system of polynomial equations,

$$0 = 1 - g_1 \tag{S146a}$$

$$0 = \sqrt{\zeta}\, g_9 g_{11} \sqrt{n_0} - g_{12}\psi \tag{S146b}$$

$$0 = \sqrt{\zeta}\, g_6 g_{11} \sqrt{n_0} - g_{10}\psi + \psi \tag{S146c}$$

$$0 = g_7(\eta - \zeta) + \sqrt{\zeta}\, g_6 g_{11} \sqrt{n_0} \tag{S146d}$$

$$0 = g_8 g_{11} n_0 \sqrt{\eta - \zeta} - g_3 \phi\big(\gamma + \sigma_{W_2}^2(\eta' - \zeta)\big) \tag{S146e}$$

$$0 = -\sqrt{\zeta}\, g_8 g_9 \psi - g_6 \sqrt{n_0}\, \phi\big(\gamma + \sigma_{W_2}^2(\eta' - \zeta)\big) \tag{S146f}$$

$$0 = -\sqrt{\zeta}\, g_8 g_{12} \psi - (g_{10} - 1)\sqrt{n_0}\, \phi\big(\gamma + \sigma_{W_2}^2(\eta' - \zeta)\big) \tag{S146g}$$

$$0 = g_6 \sqrt{n_0}\, \phi\big(\gamma + \sigma_{W_2}^2(\eta' - \zeta)\big) + g_8\big(\sqrt{\zeta}\, g_{10}\psi + \zeta g_6 \sqrt{n_0}\, \sigma_{W_2}^2\big) \tag{S146h}$$

$$0 = g_8 g_{11} \psi(\eta - \zeta) - \phi\big(g_5 \sqrt{\eta - \zeta} - \sqrt{\zeta}\, g_6 g_{11} \sqrt{n_0}\big)\big(\sigma_{W_2}^2(\zeta - \eta') - \gamma\big) \tag{S146i}$$

$$0 = (g_9 - 1)\sqrt{n_0}\, \phi\big(\gamma + \sigma_{W_2}^2(\eta' - \zeta)\big) + g_8\big(\sqrt{\zeta}\, g_{12}\psi + \zeta g_9 \sqrt{n_0}\, \sigma_{W_2}^2\big) \tag{S146j}$$

$$0 = g_1 g_8 n_0 \sqrt{\eta - \zeta} + g_3\big(g_8 \psi(\zeta - \eta) + \phi\big(\sqrt{\zeta}\, g_6 \sqrt{n_0} - 1\big)\big(\gamma + \sigma_{W_2}^2(\eta' - \zeta)\big)\big) \tag{S146k}$$

$$0 = \sqrt{\zeta}\, g_{10} g_{11} \sqrt{n_0}\, \phi\big(\sigma_{W_2}^2(\zeta - \eta') - \gamma\big) + g_{12}\psi\big(\gamma\phi + \sigma_{W_2}^2(-\zeta\phi + \phi\eta' + \zeta g_8)\big) \tag{S146l}$$

$$0 = g_{11}\big(g_8 \psi(\zeta - \eta) + \phi\big(\sqrt{\zeta}\, g_6 \sqrt{n_0} - 1\big)\big(\gamma + \sigma_{W_2}^2(\eta' - \zeta)\big)\big) + \phi\big(\gamma + \sigma_{W_2}^2(\eta' - \zeta)\big) \tag{S146m}$$

$$0 = g_{11} n_0\big(g_8 \psi(\eta - \zeta) + \sqrt{\zeta}\, g_6 \sqrt{n_0}\, \phi\big(\sigma_{W_2}^2(\zeta - \eta') - \gamma\big)\big) - g_2 \psi\phi\big(\gamma + \sigma_{W_2}^2(\eta' - \zeta)\big) \tag{S146n}$$

$$0 = g_9 \psi\big(\gamma\phi + \sigma_{W_2}^2\big(\phi(\eta' - \zeta) + \zeta g_8\big)\big) - \phi\big(\sqrt{\zeta}\, g_6 g_{11} \sqrt{n_0} + \psi\big)\big(\gamma + \sigma_{W_2}^2(\eta' - \zeta)\big) \tag{S146o}$$

$$0 = g_8\big(-\sqrt{\zeta}\, g_{12}\psi - \sqrt{n_0}\big(\gamma + g_{11}(\eta - \zeta) + \sigma_{W_2}^2\big(\eta' + \zeta(g_9 - 1)\big)\big)\big) + \sqrt{n_0}\big(\gamma + \sigma_{W_2}^2(\eta' - \zeta)\big) \tag{S146p}$$

$$0 = \sqrt{\zeta}\, g_1 g_6 \sqrt{n_0}\, \phi\big(\sigma_{W_2}^2(\zeta - \eta') - \gamma\big) - g_7(\zeta - \eta)\big(g_8 \psi(\zeta - \eta) + \phi\big(\sqrt{\zeta}\, g_6 \sqrt{n_0} - 1\big)\big(\gamma + \sigma_{W_2}^2(\eta' - \zeta)\big)\big) \tag{S146q}$$

$$
\begin{aligned}
0 = {}& g_1 n_0\big(g_8 \psi(\eta - \zeta) + \sqrt{\zeta}\, g_6 \sqrt{n_0}\, \phi\big(\sigma_{W_2}^2(\zeta - \eta') - \gamma\big)\big) + g_2 \psi\big(g_8 \psi(\zeta - \eta) \\
& + \phi\big(\sqrt{\zeta}\, g_6 \sqrt{n_0} - 1\big)\big(\gamma + \sigma_{W_2}^2(\eta' - \zeta)\big)\big)
\end{aligned}
\tag{S146r}
$$

$$
\begin{aligned}
0 = {}& g_1\big(g_8 \psi(\eta - \zeta) + \sqrt{\zeta}\, g_6 \sqrt{n_0}\, \phi\big(\sigma_{W_2}^2(\zeta - \eta') - \gamma\big)\big) + g_5 \sqrt{\eta - \zeta}\big(g_8 \psi(\eta - \zeta) \\
& - \phi\big(\sqrt{\zeta}\, g_6 \sqrt{n_0} - 1\big)\big(\gamma + \sigma_{W_2}^2(\eta' - \zeta)\big)\big)
\end{aligned}
\tag{S146s}
$$

$$0 = n_0\big(-\zeta g_5 g_8 \psi \sqrt{\eta-\zeta} + \eta g_5 g_8 \psi \sqrt{\eta-\zeta} + g_8 \psi(\zeta-\eta)\big(g_7(\zeta-\eta)-g_1\big)$$
$$+ \sqrt{\zeta}g_6\sqrt{n_0}\phi\big(g_7(\zeta-\eta)+g_1\big)\big(\gamma+\sigma_{W_2}^2(\eta'-\zeta)\big)\big) + g_4\psi\phi(\zeta-\eta)\big(\gamma+\sigma_{W_2}^2(\eta'-\zeta)\big) \tag{S146t}$$
$$0 = \sqrt{n_0}\sqrt{\eta-\zeta}\big(g_1 g_8 \sqrt{n_0}\psi(\eta-\zeta) + \sqrt{\zeta}g_1 g_6 n_0 \phi\big(\gamma+\sigma_{W_2}^2(\eta'-\zeta)\big) - \sqrt{\zeta}g_2 g_6 \psi\phi\big(\gamma+\sigma_{W_2}^2(\eta'-\zeta)\big)\big)$$
$$+ g_3\psi(\zeta-\eta)\big(g_8\psi(\eta-\zeta)+\sqrt{\zeta}g_6\sqrt{n_0}\phi\big(\sigma_{W_2}^2(\zeta-\eta')-\gamma\big)\big) + g_4\psi(-\phi)(\eta-\zeta)^{3/2}\big(\gamma+\sigma_{W_2}^2(\eta'-\zeta)\big). \tag{S146u}$$

After some straightforward algebra, one can eliminate all $g_i$ except for $g_6$ and $g_8$, which satisfy coupled polynomial equations. Those equations can be shown to be identical to eqn. (S66) by invoking the change of variables,

$$g_6 = -\frac{\sqrt{\zeta}\psi}{\sqrt{n_0}\phi}\tau_2, \quad \text{and} \quad g_8 = \big(\gamma+\sigma_{W_2}^2(\eta'-\zeta)\big)\tau_1. \tag{S147}$$

In terms of these variables, the error $E_{21}$ is given by,

$$E_{21} = 2(\tau_2/\tau_1 - 1). \tag{S148}$$

### S5.3.2 $E_{22}$

The equations satisfied by the operator-valued Stieltjes transform $G$ of $\bar{Q}_{22}$ induce the following structure on $G$,

$$G = \begin{pmatrix} 0 & G_{12} \\ G_{12}^\top & 0 \end{pmatrix}, \tag{S149}$$

where,

$$G_{12} = \begin{pmatrix} g_{11} & 0 & 0 & 0 & 0 & 0 & g_7 \\ 0 & g_5 & 0 & g_2 & 0 & g_9 & 0 \\ 0 & 0 & g_{10} & 0 & 0 & 0 & 0 \\ 0 & g_3 & 0 & g_4 & 0 & g_8 & 0 \\ g_{14} & 0 & 0 & 0 & g_1 & 0 & g_6 \\ 0 & 0 & 0 & 0 & 0 & g_{13} & 0 \\ g_{14} & 0 & 0 & 0 & 0 & 0 & g_{12} \end{pmatrix}, \tag{S150}$$

and the independent entry-wise component functions $g_i$ combine to produce the error $E_{22}$ through the relation,

$$E_{22} = \frac{2\sqrt{\zeta}g_9\sigma_{W_2}^2\big(\psi(\eta-\zeta)+\zeta\sigma_{W_2}^2\big)}{\sqrt{n_0}\psi} + 2g_8(\eta-\zeta)\sigma_{W_2}^2, \tag{S151}$$

and themselves satisfy the following system of polynomial equations,

$$0 = \sqrt{\zeta}g_{11}g_{13}\sqrt{n_0} - g_{14}\psi \tag{S152a}$$
$$0 = \sqrt{\zeta}g_7 g_{13}\sqrt{n_0} - g_{12}\psi + \psi \tag{S152b}$$
$$0 = g_1\psi\big(g_3\sqrt{n_0} - \sqrt{\zeta}g_4\big) - g_3\sqrt{n_0}\sigma_{W_2}^2 \tag{S152c}$$
$$0 = -g_1\psi\big(\sqrt{\zeta}g_5 + g_3\sqrt{n_0}\big) - g_3\sqrt{n_0}\sigma_{W_2}^2 \tag{S152d}$$
$$0 = g_1\psi\big(g_5\sqrt{n_0} - \sqrt{\zeta}g_2\big) - \sqrt{\zeta}g_2\sigma_{W_2}^2 \tag{S152e}$$
$$0 = g_1\psi\big(\sqrt{\zeta}g_2 + g_4\sqrt{n_0}\big) - \sqrt{\zeta}g_2\sigma_{W_2}^2 \tag{S152f}$$
$$0 = g_1\psi\big(g_5\sqrt{n_0} - \sqrt{\zeta}g_2\big) - (g_5-1)\sqrt{n_0}\sigma_{W_2}^2 \tag{S152g}$$
$$0 = -g_1\psi\big(\sqrt{\zeta}g_2 + g_4\sqrt{n_0}\big) - (g_4-1)\sqrt{n_0}\sigma_{W_2}^2 \tag{S152h}$$
$$0 = g_1\psi\big(g_3\sqrt{n_0} - \sqrt{\zeta}g_4\big) - \sqrt{\zeta}(g_4-1)\sigma_{W_2}^2 \tag{S152i}$$
$$0 = g_1\psi\big(\sqrt{\zeta}g_5 + g_3\sqrt{n_0}\big) - \sqrt{\zeta}(g_5-1)\sigma_{W_2}^2 \tag{S152j}$$
$$0 = -\sqrt{\zeta}g_{10}g_{11}\psi - g_7\sqrt{n_0}\phi\big(\gamma+\sigma_{W_2}^2(\eta'-\zeta)\big) \tag{S152k}$$

$$0 = -\sqrt{\zeta}g_{10}g_{14}\psi - g_6\sqrt{n_0}\phi\big(\gamma + \sigma_{W_2}^2(\eta' - \zeta)\big) \tag{S152l}$$

$$0 = -\sqrt{\zeta}g_{10}g_{14}\psi - (g_{12} - 1)\sqrt{n_0}\phi\big(\gamma + \sigma_{W_2}^2(\eta' - \zeta)\big) \tag{S152m}$$

$$0 = g_1\big(-\zeta g_2 + \sqrt{\zeta}(g_5 - g_4)\sqrt{n_0} + g_3 n_0\big) - \sqrt{\zeta}(g_1 - 1)\sqrt{n_0}\sigma_{W_2}^2 \tag{S152n}$$

$$0 = g_1\psi\big(\sqrt{\zeta}g_9 + g_8\sqrt{n_0}\big) + \sqrt{\zeta}(g_7 g_{13} n_0 - g_9)\sigma_{W_2}^2 + g_6 g_{13}\sqrt{n_0}\psi \tag{S152o}$$

$$0 = g_7\sqrt{n_0}\phi\big(\gamma + \sigma_{W_2}^2(\eta' - \zeta)\big) + g_{10}\big(\sqrt{\zeta}g_{12}\psi + \zeta g_7\sqrt{n_0}\sigma_{W_2}^2\big) \tag{S152p}$$

$$0 = (g_{11} - 1)\sqrt{n_0}\phi\big(\gamma + \sigma_{W_2}^2(\eta' - \zeta)\big) + g_{10}\big(\sqrt{\zeta}g_{14}\psi + \zeta g_{11}\sqrt{n_0}\sigma_{W_2}^2\big) \tag{S152q}$$

$$0 = \sqrt{\zeta}g_{12}g_{13}\sqrt{n_0}\phi\big(\sigma_{W_2}^2(\zeta - \eta') - \gamma\big) + g_{14}\psi\big(\gamma\phi + \sigma_{W_2}^2(-\zeta\phi + \phi\eta' + \zeta g_{10})\big) \tag{S152r}$$

$$0 = g_{13}\big(g_{10}\psi(\zeta - \eta) + \phi(\sqrt{\zeta}g_7\sqrt{n_0} - 1)(\gamma + \sigma_{W_2}^2(\eta' - \zeta))\big) + \phi\big(\gamma + \sigma_{W_2}^2(\eta' - \zeta)\big) \tag{S152s}$$

$$0 = g_6\psi\big(-\zeta g_2 + \sqrt{\zeta}(g_5 - g_4)\sqrt{n_0} + g_3 n_0\big) + \sqrt{\zeta}\sqrt{n_0}\sigma_{W_2}^2\big(g_7(\zeta g_9 + \sqrt{\zeta}(g_5 + g_8)\sqrt{n_0} + g_3 n_0) - g_6\psi\big) \tag{S152t}$$

$$0 = g_{11}\psi\big(\gamma\phi + \sigma_{W_2}^2(\phi(\eta' - \zeta) + \zeta g_{10})\big) - \phi\big(\sqrt{\zeta}g_7 g_{13}\sqrt{n_0} + \psi\big)\big(\gamma + \sigma_{W_2}^2(\eta' - \zeta)\big) \tag{S152u}$$

$$0 = g_{10}\big(-\sqrt{\zeta}g_{14}\psi - \sqrt{n_0}(\gamma + g_{13}(\eta - \zeta) + \sigma_{W_2}^2(\eta' + \zeta(g_{11} - 1)))\big) + \sqrt{n_0}\big(\gamma + \sigma_{W_2}^2(\eta' - \zeta)\big) \tag{S152v}$$

$$0 = g_{14}\psi\big(-\zeta g_2 + \sqrt{\zeta}(g_5 - g_4)\sqrt{n_0} + g_3 n_0\big) + \sqrt{\zeta}\sqrt{n_0}\sigma_{W_2}^2\big(g_{11}(\zeta g_9$$
$$+ \sqrt{\zeta}(g_5 + g_8)\sqrt{n_0} + g_3 n_0) - g_{14}\psi\big) \tag{S152w}$$

$$0 = \sqrt{\zeta}g_6 g_{13}\sqrt{n_0}\phi\big(\sigma_{W_2}^2(\zeta - \eta') - \gamma\big) - g_1\phi\big(\zeta g_9 + \sqrt{\zeta}(g_5 + g_8)\sqrt{n_0} + g_3 n_0\big)\big(\gamma + \sigma_{W_2}^2(\eta' - \zeta)\big)$$
$$+ g_{14}\psi\big(\gamma\phi + \sigma_{W_2}^2(-\zeta\phi + \phi\eta' + \zeta g_{10})\big) \tag{S152x}$$

$$0 = g_1\psi\phi\big(\sqrt{\zeta}g_9 + g_8\sqrt{n_0}\big)\big(\gamma + \sigma_{W_2}^2(\eta' - \zeta)\big) + \sqrt{n_0}\big(\sigma_{W_2}^2(g_{10}g_{13}\psi(\eta - \zeta) + g_8\phi(\gamma + \sigma_{W_2}^2(\eta' - \zeta)))$$
$$+ g_6 g_{13}\psi\phi(\gamma + \sigma_{W_2}^2(\eta' - \zeta))\big) \tag{S152y}$$

$$0 = \sqrt{\zeta}g_8\sigma_{W_2}^2\big(g_{10}\psi(\eta - \zeta) - \phi(\sqrt{\zeta}g_7\sqrt{n_0} - 1)(\gamma + \sigma_{W_2}^2(\eta' - \zeta))\big) - g_3\sqrt{n_0}\phi\big(\sqrt{\zeta}g_7\sqrt{n_0}\sigma_{W_2}^2 + g_6\psi\big)$$
$$\big(\gamma + \sigma_{W_2}^2(\eta' - \zeta)\big) + \sqrt{\zeta}g_4\psi\big(g_6\phi(\gamma + \sigma_{W_2}^2(\eta' - \zeta)) + g_{10}(\eta - \zeta)\sigma_{W_2}^2\big) \tag{S152z}$$

$$0 = \sqrt{\zeta}g_9\sigma_{W_2}^2\big(g_{10}\psi(\eta - \zeta) - \phi(\sqrt{\zeta}g_7\sqrt{n_0} - 1)(\gamma + \sigma_{W_2}^2(\eta' - \zeta))\big) - g_5\sqrt{n_0}\phi\big(\sqrt{\zeta}g_7\sqrt{n_0}\sigma_{W_2}^2 + g_6\psi\big)$$
$$\big(\gamma + \sigma_{W_2}^2(\eta' - \zeta)\big) + \sqrt{\zeta}g_2\psi\big(g_6\phi(\gamma + \sigma_{W_2}^2(\eta' - \zeta)) + g_{10}(\eta - \zeta)\sigma_{W_2}^2\big) \tag{S152aa}$$

After some straightforward algebra, one can eliminate all $g_i$ except for $g_7$ and $g_{10}$, which satisfy coupled polynomial equations. Those equations can be shown to be identical to eqn. (S66) by invoking the change of variables,

$$g_7 = -\frac{\sqrt{\zeta}\psi}{\sqrt{n_0}\phi}\tau_2, \quad \text{and} \quad g_{10} = \big(\gamma + \sigma_{W_2}^2(\eta' - \zeta)\big)\tau_1. \tag{S153}$$

The error $E_{22}$ is then given by,

$$E_{22} = 2\zeta\left(\frac{\tau_2}{\tau_1} - 1\right) + \frac{2\psi\big(\zeta(\tau_2 - \tau_1) + \eta\tau_1\big)^2\big((\tau_2 - \tau_1)\phi + \zeta\tau_1\tau_2\sigma_{W_2}^2\big)}{\zeta\tau_1^2\tau_2\phi}. \tag{S154}$$

### S5.3.3  $E_{31}$

The equations satisfied by the operator-valued Stieltjes transform $G$ of $\bar{Q}_{31}$ induce the following structure on $G$,

$$G = \begin{pmatrix} 0 & G_{12} \\ G_{12}^\top & 0 \end{pmatrix}, \tag{S155}$$

where,

$$G_{12} = \begin{pmatrix} g_5 & 0 & 0 & 0 & 0 & g_2 & 0 & 0 \\ 0 & g_6 & 0 & g_1 & g_3 & 0 & 0 & g_4 \\ 0 & 0 & g_8 & 0 & 0 & 0 & g_{12} & 0 \\ 0 & g_{11} & 0 & g_7 & g_{10} & 0 & 0 & g_9 \\ 0 & 0 & 0 & 0 & g_6 & 0 & 0 & g_1 \\ 0 & 0 & 0 & 0 & 0 & g_5 & 0 & 0 \\ 0 & 0 & 0 & 0 & 0 & 0 & g_8 & 0 \\ 0 & 0 & 0 & 0 & g_{11} & 0 & 0 & g_7 \end{pmatrix}, \tag{S156}$$

and the independent entry-wise component functions $g_i$ give the error $E_{31}$ through the relation,

$$E_{31} = \frac{g_2 n_0 \sigma_\varepsilon^2}{\phi\big(\gamma + \sigma_{W_2}^2(\eta' - \zeta)\big)}, \tag{S157}$$

and themselves satisfy the following system of polynomial equations,

$$0 = \sqrt{\zeta} g_6 g_8 \sqrt{n_0} - g_{11}\psi \tag{S158a}$$

$$0 = \sqrt{\zeta} g_1 g_8 \sqrt{n_0} - g_7\psi + \psi \tag{S158b}$$

$$0 = -\sqrt{\zeta} g_5 g_6 \psi - g_1 \sqrt{n_0}\phi\big(\gamma + \sigma_{W_2}^2(\eta' - \zeta)\big) \tag{S158c}$$

$$0 = -\sqrt{\zeta} g_5 g_{11}\psi - (g_7 - 1)\sqrt{n_0}\phi\big(\gamma + \sigma_{W_2}^2(\eta' - \zeta)\big) \tag{S158d}$$

$$0 = -\zeta g_7 g_8 \psi + \sqrt{\zeta}\sqrt{n_0}\big((g_4 g_8 + g_1 g_{12})n_0 - \zeta g_1 g_8 \sigma_{W_2}^2\big) - g_9 n_0 \psi \tag{S158e}$$

$$0 = g_1 \sqrt{n_0}\phi\big(\gamma + \sigma_{W_2}^2(\eta' - \zeta)\big) + g_5\big(\sqrt{\zeta} g_7 \psi + \zeta g_1 \sqrt{n_0}\sigma_{W_2}^2\big) \tag{S158f}$$

$$0 = \sqrt{\zeta} g_6 g_{12} n_0^{3/2} - g_8\big(\zeta g_{11}\psi + \sqrt{\zeta}\sqrt{n_0}(\zeta g_6 \sigma_{W_2}^2 - g_3 n_0)\big) - g_{10} n_0 \psi \tag{S158g}$$

$$0 = (g_6 - 1)\sqrt{n_0}\phi\big(\gamma + \sigma_{W_2}^2(\eta' - \zeta)\big) + g_5\big(\sqrt{\zeta} g_{11}\psi + \zeta g_6 \sqrt{n_0}\sigma_{W_2}^2\big) \tag{S158h}$$

$$0 = \sqrt{\zeta} g_7 g_8 \sqrt{n_0}\phi\big(\sigma_{W_2}^2(\zeta - \eta') - \gamma\big) + g_{11}\psi\big(\gamma\phi + \sigma_{W_2}^2(-\zeta\phi + \phi\eta' + \zeta g_5)\big) \tag{S158i}$$

$$0 = g_8\big(g_5\psi(\zeta - \eta) + \phi(\sqrt{\zeta} g_1 \sqrt{n_0} - 1)(\gamma + \sigma_{W_2}^2(\eta' - \zeta))\big) + \phi\big(\gamma + \sigma_{W_2}^2(\eta' - \zeta)\big) \tag{S158j}$$

$$0 = g_6\psi\big(\gamma\phi + \sigma_{W_2}^2(-\zeta\phi + \phi\eta' + \zeta g_5)\big) - \phi\big(\sqrt{\zeta} g_1 g_8 \sqrt{n_0} + \psi\big)\big(\gamma + \sigma_{W_2}^2(\eta' - \zeta)\big) \tag{S158k}$$

$$0 = g_5\big(\sqrt{\zeta} g_{11}\psi + \sqrt{n_0}(\gamma + g_8(\eta - \zeta) + \sigma_{W_2}^2(\eta' + \zeta(g_6 - 1)))\big) - \sqrt{n_0}\big(\gamma + \sigma_{W_2}^2(\eta' - \zeta)\big) \tag{S158l}$$

$$0 = \sqrt{\zeta} g_5\psi\big(g_6(\psi(\eta - \zeta) + \zeta\sigma_{W_2}^2) - g_3 n_0\big) - \sqrt{n_0}\big(\sqrt{\zeta} g_2 g_6 \sqrt{n_0}\psi + g_4 n_0 \phi(\gamma + \sigma_{W_2}^2(\eta' - \zeta))$$
$$+ \zeta g_1 g_8 \phi(\gamma + \sigma_{W_2}^2(\eta' - \zeta))\big) \tag{S158m}$$

$$0 = \sqrt{\zeta} g_5\psi\big(g_{11}(\psi(\eta - \zeta) + \zeta\sigma_{W_2}^2) - g_{10} n_0\big) - \sqrt{n_0}\big(\sqrt{\zeta} g_2 g_{11} \sqrt{n_0}\psi + g_9 n_0 \phi(\gamma + \sigma_{W_2}^2(\eta' - \zeta))$$
$$+ \zeta g_7 g_8 \phi(\gamma + \sigma_{W_2}^2(\eta' - \zeta))\big) \tag{S158n}$$

$$0 = g_5\big(-\sqrt{\zeta} g_9 n_0 \psi + \zeta\sqrt{n_0}\sigma_{W_2}^2(\zeta g_1 \sigma_{W_2}^2 - g_4 n_0) + \sqrt{\zeta} g_7\psi(\psi(\eta - \zeta) + \zeta\sigma_{W_2}^2)\big)$$
$$- n_0\big(g_4 \sqrt{n_0}\phi(\gamma + \sigma_{W_2}^2(\eta' - \zeta)) + g_2(\sqrt{\zeta} g_7\psi + \zeta g_1 \sqrt{n_0}\sigma_{W_2}^2)\big) \tag{S158o}$$

$$0 = g_5\big(-\sqrt{\zeta} g_{10} n_0 \psi + \zeta\sqrt{n_0}\sigma_{W_2}^2(\zeta g_6 \sigma_{W_2}^2 - g_3 n_0) + \sqrt{\zeta} g_{11}\psi(\psi(\eta - \zeta) + \zeta\sigma_{W_2}^2)\big)$$
$$- n_0\big(g_3 \sqrt{n_0}\phi(\gamma + \sigma_{W_2}^2(\eta' - \zeta)) + g_2(\sqrt{\zeta} g_{11}\psi + \zeta g_6 \sqrt{n_0}\sigma_{W_2}^2)\big) \tag{S158p}$$

$$0 = g_2 g_8 n_0 \psi(\eta - \zeta) - g_5\psi(\zeta - \eta)\big(g_8\psi(\zeta - \eta) + g_{12} n_0\big) - \sqrt{n_0}\phi\big(g_{12}(\sqrt{\zeta} g_1 n_0 - \sqrt{n_0})$$
$$+ \sqrt{\zeta} g_8(g_4 n_0 - \zeta g_1 \sigma_{W_2}^2)\big)\big(\gamma + \sigma_{W_2}^2(\eta' - \zeta)\big) + \zeta g_7 g_8 \psi\phi\big(\gamma + \sigma_{W_2}^2(\eta' - \zeta)\big) \tag{S158q}$$

$$0 = g_2 n_0\big(-\sqrt{\zeta} g_{11}\psi - \sqrt{n_0}(\gamma + g_8(\eta - \zeta) + \sigma_{W_2}^2(\eta' + \zeta(g_6 - 1)))\big) + g_5\big(\sqrt{n_0}(g_8\psi(\zeta - \eta)^2$$
$$+ g_{12} n_0(\zeta - \eta) - \sqrt{\zeta} g_{10}\sqrt{n_0}\psi - \zeta g_3 n_0 \sigma_{W_2}^2 + \zeta^2 g_6 \sigma_{W_2}^4\big) + \sqrt{\zeta} g_{11}\psi(\psi(\eta - \zeta) + \zeta\sigma_{W_2}^2)\big) \tag{S158r}$$

$$0 = g_3 n_0 \psi\big(\gamma\phi + \sigma_{W_2}^2(\phi(\eta' - \zeta) + \zeta g_5)\big) - \sqrt{\zeta}\big(g_4 g_8 n_0^{3/2}\phi(\gamma + \sigma_{W_2}^2(\eta' - \zeta))$$

$$+ g_1\sqrt{n_0}\phi\big(g_{12}n_0 - \zeta g_8\sigma_{W_2}^2\big)\big(\gamma + \sigma_{W_2}^2(\eta' - \zeta)\big) + \sqrt{\zeta}g_6\psi\sigma_{W_2}^2\big(\zeta g_5\sigma_{W_2}^2 - g_2 n_0\big)\big) \tag{S158s}$$

$$0 = g_{10}n_0\psi\big(\gamma\phi + \sigma_{W_2}^2(\phi(\eta' - \zeta) + \zeta g_5)\big) - \sqrt{\zeta}\big(g_7 g_{12}n_0^{3/2}\phi(\gamma + \sigma_{W_2}^2(\eta' - \zeta))$$
$$+ g_8\sqrt{n_0}\phi\big(g_9 n_0 - \zeta g_7\sigma_{W_2}^2\big)\big(\gamma + \sigma_{W_2}^2(\eta' - \zeta)\big) + \sqrt{\zeta}g_{11}\psi\sigma_{W_2}^2\big(\zeta g_5\sigma_{W_2}^2 - g_2 n_0\big)\big) \tag{S158t}$$

After some straightforward algebra, one can eliminate all $g_i$ except for $g_1$ and $g_5$, which satisfy coupled polynomial equations. Those equations can be shown to be identical to eqn. (S66) by invoking the change of variables,

$$g_1 = -\frac{\sqrt{\zeta}\psi}{\sqrt{n_0}\phi}\tau_2, \quad \text{and} \quad g_5 = \big(\gamma + \sigma_{W_2}^2(\eta' - \zeta)\big)\tau_1. \tag{S159}$$

The error $E_{31}$ can then be written in terms of $\tau_1$ and its derivative $\tau_1'$ (S87),

$$E_{31} = \sigma_\varepsilon^2\big(-\tau_1'/\tau_1^2 - 1\big). \tag{S160}$$

### S5.3.4  $E_{32}$

The equations satisfied by the operator-valued Stieltjes transform $G$ of $\bar{Q}_{32}$ induce the following structure on $G$,

$$G = \begin{pmatrix} 0 & G_{12} \\ G_{12}^\top & 0 \end{pmatrix}, \tag{S161}$$

where,

$$G_{12} = \begin{pmatrix}
g_9 & 0 & 0 & 0 & 0 & 0 & g_6 & 0 & 0 \\
0 & g_1 & g_3 & 0 & g_4 & g_7 & 0 & 0 & g_2 \\
0 & 0 & g_{10} & 0 & g_4 & g_{13} & 0 & 0 & g_5 \\
0 & 0 & 0 & g_{12} & 0 & 0 & 0 & g_{16} & 0 \\
0 & 0 & g_{15} & 0 & g_{11} & g_{14} & 0 & 0 & g_8 \\
0 & 0 & 0 & 0 & 0 & g_{10} & 0 & 0 & g_4 \\
0 & 0 & 0 & 0 & 0 & 0 & g_9 & 0 & 0 \\
0 & 0 & 0 & 0 & 0 & 0 & 0 & g_{12} & 0 \\
0 & 0 & 0 & 0 & 0 & g_{15} & 0 & 0 & g_{11}
\end{pmatrix}, \tag{S162}$$

and the independent entry-wise component functions $g_i$ give the error $E_{32}$ through the relation,

$$E_{32} = -g_2 n_0^{3/2}/(\sqrt{\zeta}\psi), \tag{S163}$$

and themselves satisfy the following system of polynomial equations,

$$0 = \sqrt{\zeta}g_{10}g_{12}\sqrt{n_0} - g_{15}\psi \tag{S164a}$$

$$0 = \sqrt{\zeta}g_4 g_{12}\sqrt{n_0} - g_{11}\psi + \psi \tag{S164b}$$

$$0 = -\sqrt{\zeta}g_9 g_{10}\psi - g_4\sqrt{n_0}\phi(\gamma + \sigma_{W_2}^2(\eta' - \zeta)) \tag{S164c}$$

$$0 = -\sqrt{\zeta}g_9 g_{15}\psi - (g_{11} - 1)\sqrt{n_0}\phi(\gamma + \sigma_{W_2}^2(\eta' - \zeta)) \tag{S164d}$$

$$0 = -\sqrt{\zeta}g_9\psi - \sqrt{\zeta}g_3 g_9\psi - g_4\sqrt{n_0}\phi(\gamma + \sigma_{W_2}^2(\eta' - \zeta)) \tag{S164e}$$

$$0 = -\sqrt{\zeta}g_6 g_{10}\psi - \sqrt{\zeta}g_9 g_{13}\psi - g_5\sqrt{n_0}\phi(\gamma + \sigma_{W_2}^2(\eta' - \zeta)) \tag{S164f}$$

$$0 = -\sqrt{\zeta}g_9 g_{14}\psi - \sqrt{\zeta}g_6 g_{15}\psi - g_8\sqrt{n_0}\phi(\gamma + \sigma_{W_2}^2(\eta' - \zeta)) \tag{S164g}$$

$$0 = \sqrt{\zeta}g_5 g_{12}n_0 + \sqrt{\zeta}g_4\big(g_{16}n_0 + g_{12}(\zeta\psi - \eta\psi - \zeta\sigma_{W_2}^2)\big) + g_8\sqrt{n_0}(-\psi) \tag{S164h}$$

$$0 = g_4\sqrt{n_0}\phi(\gamma + \sigma_{W_2}^2(\eta' - \zeta)) + g_9\big(\sqrt{\zeta}g_{11}\psi + \zeta g_4\sqrt{n_0}\sigma_{W_2}^2\big) \tag{S164i}$$

$$0 = g_3\sqrt{n_0}\phi(\gamma + \sigma_{W_2}^2(\eta' - \zeta)) + g_9\big(\sqrt{\zeta}g_{15}\psi + \zeta g_{10}\sqrt{n_0}\sigma_{W_2}^2\big) \tag{S164j}$$

$$0 = \sqrt{\zeta}g_{12}g_{13}n_0 + \sqrt{\zeta}g_{10}\big(g_{16}n_0 + g_{12}(\zeta\psi - \eta\psi - \zeta\sigma_{W_2}^2)\big) + g_{14}\sqrt{n_0}(-\psi) \tag{S164k}$$

$$0 = (g_{10} - 1)\sqrt{n_0}\phi\big(\gamma + \sigma_{W_2}^2(\eta' - \zeta)\big) + g_9\big(\sqrt{\zeta}g_{15}\psi + \zeta g_{10}\sqrt{n_0}\sigma_{W_2}^2\big) \tag{S164l}$$

$$0 = -\sqrt{\zeta}\big((g_1 + g_3)g_6 + g_7 g_9\big)\psi - \gamma g_2\sqrt{n_0}\phi + \zeta g_2\sqrt{n_0}\phi\sigma_{W_2}^2 + g_2\sqrt{n_0}(-\phi)\eta'\sigma_{W_2}^2 \tag{S164m}$$

$$0 = \sqrt{\zeta}g_{11}g_{12}\sqrt{n_0}\phi\big(\sigma_{W_2}^2(\zeta - \eta') - \gamma\big) + g_{15}\psi\big(\gamma\phi + \sigma_{W_2}^2\big(-\zeta\phi + \phi\eta' + \zeta g_9\big)\big) \tag{S164n}$$

$$0 = g_{12}\big(g_9\psi(\zeta - \eta) + \phi\big(\sqrt{\zeta}g_4\sqrt{n_0} - 1\big)\big(\gamma + \sigma_{W_2}^2(\eta' - \zeta)\big)\big) + \phi\big(\gamma + \sigma_{W_2}^2(\eta' - \zeta)\big) \tag{S164o}$$

$$0 = g_{10}\psi\big(\gamma\phi + \sigma_{W_2}^2\big(-\zeta\phi + \phi\eta' + \zeta g_9\big)\big) - \phi\big(\sqrt{\zeta}g_4 g_{12}\sqrt{n_0} + \psi\big)\big(\gamma + \sigma_{W_2}^2(\eta' - \zeta)\big) \tag{S164p}$$

$$0 = g_9\big(\sqrt{\zeta}g_{15}\psi + \sqrt{n_0}\big(\gamma + g_{12}(\eta - \zeta) + \sigma_{W_2}^2\big(\eta' + \zeta(g_{10} - 1)\big)\big)\big) - \sqrt{n_0}\big(\gamma + \sigma_{W_2}^2(\eta' - \zeta)\big) \tag{S164q}$$

$$0 = -\sqrt{\zeta}g_4 g_{12}\sqrt{n_0}\phi\big(\gamma + \sigma_{W_2}^2(\eta' - \zeta)\big) + g_3\psi\big(\gamma\phi + \sigma_{W_2}^2\big(-\zeta\phi + \phi\eta' + \zeta g_9\big)\big) + \zeta g_9\psi\sigma_{W_2}^2 \tag{S164r}$$

$$0 = g_7 n_0\phi\big(\gamma + \sigma_{W_2}^2(\eta' - \zeta)\big) + g_6\big(\sqrt{\zeta}g_{15}\sqrt{n_0}\psi + \zeta g_{10}n_0\sigma_{W_2}^2\big) + g_9\big(\sqrt{\zeta}g_{14}\sqrt{n_0}\psi + \zeta\sigma_{W_2}^2\big(g_{13}n_0 - \zeta g_{10}\sigma_{W_2}^2\big)\big) \tag{S164s}$$

$$\begin{aligned}
0 = {}& \gamma g_2 n_0\phi + \sqrt{\zeta}g_8 g_9\sqrt{n_0}\psi + g_6\big(\sqrt{\zeta}g_{11}\sqrt{n_0}\psi + \zeta g_4 n_0\sigma_{W_2}^2\big) - \zeta g_2 n_0\phi\sigma_{W_2}^2 + \zeta g_5 g_9 n_0\sigma_{W_2}^2 \\
& + g_2 n_0\phi\eta'\sigma_{W_2}^2 - \zeta^2 g_4 g_9\sigma_{W_2}^4
\end{aligned} \tag{S164t}$$

$$\begin{aligned}
0 = {}& g_6\big(-\sqrt{\zeta}g_{15}\sqrt{n_0}\psi - n_0\big(\gamma + g_{12}(\eta - \zeta) + \sigma_{W_2}^2\big(\eta' + \zeta(g_{10} - 1)\big)\big)\big) + g_9\big(g_{12}\psi(\zeta - \eta)^2 \\
& + g_{16}n_0(\zeta - \eta) - \sqrt{\zeta}g_{14}\sqrt{n_0}\psi - \zeta g_{13}n_0\sigma_{W_2}^2 + \zeta^2 g_{10}\sigma_{W_2}^4\big)
\end{aligned} \tag{S164u}$$

$$\begin{aligned}
0 = {}& \gamma g_5 n_0\phi + \sqrt{\zeta}g_8 g_9\sqrt{n_0}\psi + \sqrt{\zeta}g_6 g_{11}\sqrt{n_0}\psi + \zeta g_4\big(g_6 n_0\sigma_{W_2}^2 + g_{12}\phi\big(\gamma + \sigma_{W_2}^2(\eta' - \zeta)\big) - \zeta g_9\sigma_{W_2}^4\big) \\
& - \zeta g_5 n_0\phi\sigma_{W_2}^2 + \zeta g_5 g_9 n_0\sigma_{W_2}^2 + g_5 n_0\phi\eta'\sigma_{W_2}^2
\end{aligned} \tag{S164v}$$

$$\begin{aligned}
0 = {}& \gamma g_{13}n_0\phi + \sqrt{\zeta}g_6 g_{15}\sqrt{n_0}\psi + \zeta g_{10}\big(g_6 n_0\sigma_{W_2}^2 + g_{12}\phi\big(\gamma + \sigma_{W_2}^2(\eta' - \zeta)\big) - \zeta g_9\sigma_{W_2}^4\big) \\
& + g_9\big(\sqrt{\zeta}g_{14}\sqrt{n_0}\psi + \zeta g_{13}n_0\sigma_{W_2}^2\big) - \zeta g_{13}n_0\phi\sigma_{W_2}^2 + g_{13}n_0\phi\eta'\sigma_{W_2}^2
\end{aligned} \tag{S164w}$$

$$\begin{aligned}
0 = {}& -\sqrt{\zeta}g_{12}\phi\big(\gamma + \sigma_{W_2}^2(\eta' - \zeta)\big)\big(\sqrt{n_0}\big(g_8 n_0 + g_{11}\big(\zeta\psi - \eta\psi - \zeta\sigma_{W_2}^2\big)\big) - \sqrt{\zeta}g_{15}\psi\big) \\
& + g_{14}n_0\psi\big(\gamma\phi + \sigma_{W_2}^2\big(-\zeta\phi + \phi\eta' + \zeta g_9\big)\big) - \sqrt{\zeta}g_{11}g_{16}n_0^{3/2}\phi\big(\gamma + \sigma_{W_2}^2(\eta' - \zeta)\big) + \zeta g_{15}\psi\sigma_{W_2}^2\big(g_6 n_0 - \zeta g_9\sigma_{W_2}^2\big)
\end{aligned} \tag{S164x}$$

$$\begin{aligned}
0 = {}& g_9\psi(-(\zeta - \eta))\big(g_{12}\psi(\zeta - \eta) + g_{16}n_0\big) - \sqrt{\zeta}g_4\sqrt{n_0}\phi\big(\gamma + \sigma_{W_2}^2(\eta' - \zeta)\big)\big(g_{16}n_0 + g_{12}\big(\zeta\psi - \eta\psi - \zeta\sigma_{W_2}^2\big)\big) \\
& + n_0\big(g_6 g_{12}\psi(\eta - \zeta) + \phi\big(g_{16} - \sqrt{\zeta}g_5 g_{12}\sqrt{n_0}\big)\big(\gamma + \sigma_{W_2}^2(\eta' - \zeta)\big)\big) + \zeta g_{10}g_{12}\psi\phi\big(\gamma + \sigma_{W_2}^2(\eta' - \zeta)\big)
\end{aligned} \tag{S164y}$$

$$\begin{aligned}
0 = {}& g_{13}n_0\psi\big(\gamma\phi + \sigma_{W_2}^2\big(-\zeta\phi + \phi\eta' + \zeta g_9\big)\big) - \sqrt{\zeta}g_4\sqrt{n_0}\phi\big(\gamma + \sigma_{W_2}^2(\eta' - \zeta)\big) + \sqrt{\zeta}g_5 g_{12}n_0^{3/2}\phi\big(\sigma_{W_2}^2(\zeta - \eta') - \gamma\big) \\
& \big(g_{16}n_0 + g_{12}\big(\zeta\psi - \eta\psi - \zeta\sigma_{W_2}^2\big)\big) + \zeta g_{10}\psi\big(g_6 n_0\sigma_{W_2}^2 + g_{12}\phi\big(\gamma + \sigma_{W_2}^2(\eta' - \zeta)\big) - \zeta g_9\sigma_{W_2}^4\big)
\end{aligned} \tag{S164z}$$

$$\begin{aligned}
0 = {}& -\gamma\sqrt{\zeta}g_2 g_{12}n_0^{3/2}\phi + \gamma g_7 n_0\psi\phi - \sqrt{\zeta}g_4\sqrt{n_0}\phi\big(\gamma + \sigma_{W_2}^2(\eta' - \zeta)\big)\big(g_{16}n_0 + g_{12}\big(\zeta\psi - \eta\psi - \zeta\sigma_{W_2}^2\big)\big) \\
& + \zeta g_3\psi\big(g_6 n_0\sigma_{W_2}^2 + g_{12}\phi\big(\gamma + \sigma_{W_2}^2(\eta' - \zeta)\big) - \zeta g_9\sigma_{W_2}^4\big) + \zeta^{3/2}g_2 g_{12}n_0^{3/2}\phi\sigma_{W_2}^2 - \zeta^2 g_9\psi\sigma_{W_2}^4 \\
& + n_0\phi\eta'\sigma_{W_2}^2\big(g_7\psi - \sqrt{\zeta}g_2 g_{12}\sqrt{n_0}\big) + \zeta g_6 n_0\psi\sigma_{W_2}^2 + \zeta g_7 g_9 n_0\psi\sigma_{W_2}^2 - \zeta g_7 n_0\psi\phi\sigma_{W_2}^2
\end{aligned} \tag{S164aa}$$

After some straightforward algebra, one can eliminate all $g_i$ except for $g_4$ and $g_9$, which satisfy coupled polynomial equations. Those equations can be shown to be identical to eqn. (S66) by invoking the change of variables,

$$g_4 = -\frac{\sqrt{\zeta}\psi}{\sqrt{n_0}\phi}\tau_2, \quad \text{and} \quad g_9 = \big(\gamma + \sigma_{W_2}^2(\eta' - \zeta)\big)\tau_1. \tag{S165}$$

In terms of $\tau_1$, $\tau_2$, and $\tau_2'$ (S88), the error $E_{32}$ is given by,

$$E_{32} = 1 - 2\tau_2/\tau_1 - \tau_2'/\tau_1^2. \tag{S166}$$

### S5.3.5  $E_{33}$

The equations satisfied by the operator-valued Stieltjes transform $G$ of $\bar{Q}_{32}$ induce the following structure on $G$,

$$G = \begin{pmatrix} 0 & G_{12} \\ G_{12}^\top & 0 \end{pmatrix}, \tag{S167}$$

where,

$$
G_{12} = \begin{pmatrix}
g_{13} & 0 & 0 & 0 & 0 & 0 & 0 & g_8 & 0 & 0 & 0 \\
0 & g_1 & 0 & 0 & g_5 & 0 & 0 & 0 & g_{11} & g_3 & 0 \\
0 & 0 & g_1 & g_4 & 0 & g_6 & g_9 & 0 & 0 & 0 & g_2 \\
0 & 0 & 0 & g_{14} & 0 & g_6 & g_{17} & 0 & 0 & 0 & g_7 \\
0 & 0 & 0 & 0 & g_{16} & 0 & 0 & 0 & g_{20} & g_{12} & 0 \\
0 & 0 & 0 & g_{19} & 0 & g_{15} & g_{18} & 0 & 0 & 0 & g_{10} \\
0 & 0 & 0 & 0 & 0 & 0 & g_{14} & 0 & 0 & 0 & g_6 \\
0 & 0 & 0 & 0 & 0 & 0 & 0 & g_{13} & 0 & 0 & 0 \\
0 & 0 & 0 & 0 & 0 & 0 & 0 & 0 & g_1 & g_5 & 0 \\
0 & 0 & 0 & 0 & 0 & 0 & 0 & 0 & 0 & g_{16} & 0 \\
0 & 0 & 0 & 0 & 0 & 0 & g_{19} & 0 & 0 & 0 & g_{15}
\end{pmatrix} , \tag{S168}
$$

and the independent entry-wise component functions $g_i$ give the error $E_{32}$ through the relation,

$$
E_{33} = -g_3 n_0 \sigma_{W_2}^2 / \psi , \tag{S169}
$$

and themselves satisfy the following system of polynomial equations,

$$
0 = \sqrt{\zeta} g_{14} g_{16} \sqrt{n_0} - g_{19}\psi \tag{S170a}
$$

$$
0 = \sqrt{\zeta} g_6 g_{16} \sqrt{n_0} - g_{15}\psi + \psi \tag{S170b}
$$

$$
0 = -\sqrt{\zeta} g_{13} g_{14}\psi - g_6 \sqrt{n_0}\phi\big(\gamma + \sigma_{W_2}^2(\eta' - \zeta)\big) \tag{S170c}
$$

$$
0 = -\sqrt{\zeta} g_{13} g_{19}\psi - (g_{15} - 1)\sqrt{n_0}\phi\big(\gamma + \sigma_{W_2}^2(\eta' - \zeta)\big) \tag{S170d}
$$

$$
0 = -\sqrt{\zeta} g_{13}\psi - \sqrt{\zeta} g_4 g_{13}\psi - g_6 \sqrt{n_0}\phi\big(\gamma + \sigma_{W_2}^2(\eta' - \zeta)\big) \tag{S170e}
$$

$$
0 = -\sqrt{\zeta} g_8 g_{14}\psi - \sqrt{\zeta} g_{13} g_{17}\psi - g_7 \sqrt{n_0}\phi\big(\gamma + \sigma_{W_2}^2(\eta' - \zeta)\big) \tag{S170f}
$$

$$
0 = -\sqrt{\zeta} g_{13} g_{18}\psi - \sqrt{\zeta} g_8 g_{19}\psi - g_{10} \sqrt{n_0}\phi\big(\gamma + \sigma_{W_2}^2(\eta' - \zeta)\big) \tag{S170g}
$$

$$
0 = g_{13} g_{16}\psi(\zeta - \eta) - \phi\big(g_5 - \sqrt{\zeta} g_6 g_{16}\sqrt{n_0}\big)\big(\gamma + \sigma_{W_2}^2(\eta' - \zeta)\big) \tag{S170h}
$$

$$
0 = g_6 \sqrt{n_0}\phi\big(\gamma + \sigma_{W_2}^2(\eta' - \zeta)\big) + g_{13}\big(\sqrt{\zeta} g_{15}\psi + \zeta g_6 \sqrt{n_0}\sigma_{W_2}^2\big) \tag{S170i}
$$

$$
0 = g_4 \sqrt{n_0}\phi\big(\gamma + \sigma_{W_2}^2(\eta' - \zeta)\big) + g_{13}\big(\sqrt{\zeta} g_{19}\psi + \zeta g_{14} \sqrt{n_0}\sigma_{W_2}^2\big) \tag{S170j}
$$

$$
0 = (g_{14} - 1)\sqrt{n_0}\phi\big(\gamma + \sigma_{W_2}^2(\eta' - \zeta)\big) + g_{13}\big(\sqrt{\zeta} g_{19}\psi + \zeta g_{14} \sqrt{n_0}\sigma_{W_2}^2\big) \tag{S170k}
$$

$$
0 = -\sqrt{\zeta}\big((g_4 + 1)g_8 + g_9 g_{13}\big)\psi - \gamma g_2 \sqrt{n_0}\phi + \zeta g_2 \sqrt{n_0}\phi\sigma_{W_2}^2 + g_2 \sqrt{n_0}(-\phi)\eta'\sigma_{W_2}^2 \tag{S170l}
$$

$$
0 = \sqrt{\zeta} g_{15} g_{16} \sqrt{n_0}\phi\big(\sigma_{W_2}^2(\zeta - \eta') - \gamma\big) + g_{19}\psi\big(\gamma\phi + \sigma_{W_2}^2(-\zeta\phi + \phi\eta' + \zeta g_{13})\big) \tag{S170m}
$$

$$
0 = g_{16}\big(g_{13}\psi(\zeta - \eta) + \phi\big(\sqrt{\zeta} g_6 \sqrt{n_0} - 1\big)\big(\gamma + \sigma_{W_2}^2(\eta' - \zeta)\big)\big) + \phi\big(\gamma + \sigma_{W_2}^2(\eta' - \zeta)\big) \tag{S170n}
$$

$$
0 = g_{13}\big(\sqrt{\zeta} g_{19}\psi + \sqrt{n_0}\big(\gamma + g_{16}(\eta - \zeta) + \sigma_{W_2}^2(\eta' + \zeta(g_{14} - 1))\big)\big) - \sqrt{n_0}\big(\gamma + \sigma_{W_2}^2(\eta' - \zeta)\big) \tag{S170o}
$$

$$
0 = g_{14}\psi\big(\gamma\phi + \sigma_{W_2}^2(\phi(\eta' - \zeta) + \zeta g_{13})\big) - \phi\big(\sqrt{\zeta} g_6 g_{16}\sqrt{n_0} + \psi\big)\big(\gamma + \sigma_{W_2}^2(\eta' - \zeta)\big) \tag{S170p}
$$

$$
0 = -\sqrt{\zeta} g_6 g_{16}\sqrt{n_0}\phi\big(\gamma + \sigma_{W_2}^2(\eta' - \zeta)\big) + g_4\psi\big(\gamma\phi + \sigma_{W_2}^2(-\zeta\phi + \phi\eta' + \zeta g_{13})\big) + \zeta g_{13}\psi\sigma_{W_2}^2 \tag{S170q}
$$

$$
0 = \sqrt{\zeta}\big(g_7 g_{16} + g_6(g_{12} + g_{20})\big)n_0 + g_{10}\sqrt{n_0}(-\psi) + \sqrt{\zeta} g_6\big(\psi(\zeta - \eta) - \zeta\sigma_{W_2}^2\big) + \sqrt{\zeta} g_5 g_6\big(\zeta\psi - \eta\psi - \zeta\sigma_{W_2}^2\big) \tag{S170r}
$$

$$
0 = \sqrt{\zeta}\big(g_{16} g_{17} + g_{14}(g_{12} + g_{20})\big)n_0 + g_{18}\sqrt{n_0}(-\psi) + \sqrt{\zeta} g_{14}\big(\psi(\zeta - \eta) - \zeta\sigma_{W_2}^2\big) + \sqrt{\zeta} g_5 g_{14}\big(\zeta\psi - \eta\psi - \zeta\sigma_{W_2}^2\big) \tag{S170s}
$$

$$
0 = g_{13}\psi(\zeta - \eta) + g_5\big(g_{13}\psi(\zeta - \eta) + \phi\big(\sqrt{\zeta} g_6 \sqrt{n_0} - 1\big)\big(\gamma + \sigma_{W_2}^2(\eta' - \zeta)\big)\big) + \sqrt{\zeta} g_6 \sqrt{n_0}\phi\big(\gamma + \sigma_{W_2}^2(\eta' - \zeta)\big) \tag{S170t}
$$

$$
0 = g_9 n_0 \phi\big(\gamma + \sigma_{W_2}^2(\eta' - \zeta)\big) + g_8\big(\sqrt{\zeta} g_{19}\sqrt{n_0}\psi + \zeta g_{14} n_0 \sigma_{W_2}^2\big) + g_{13}\big(\sqrt{\zeta} g_{18}\sqrt{n_0}\psi + \zeta\sigma_{W_2}^2(g_{17} n_0 - \zeta g_{14}\sigma_{W_2}^2)\big) \tag{S170u}
$$

$$
0 = \gamma g_2 n_0 \phi + \sqrt{\zeta} g_{10} g_{13}\sqrt{n_0}\psi + g_8\big(\sqrt{\zeta} g_{15}\sqrt{n_0}\psi + \zeta g_6 n_0 \sigma_{W_2}^2\big) - \zeta g_2 n_0 \phi\sigma_{W_2}^2
$$
$$
+ \zeta g_7 g_{13} n_0 \sigma_{W_2}^2 + g_2 n_0 \phi\eta'\sigma_{W_2}^2 - \zeta^2 g_6 g_{13}\sigma_{W_2}^4 \tag{S170v}
$$

$$0 = g_{13}g_{16}\psi(-(\zeta - \eta))\big(\psi(\zeta - \eta) - \zeta\sigma_{W_2}^2\big)$$
$$- \phi\big(\gamma + \sigma_{W_2}^2(\eta' - \zeta)\big)\big(-\zeta g_{14}g_{16}\psi + \sqrt{\zeta}g_6 g_{16}\sqrt{n_0}(\zeta\psi - \eta\psi - \zeta\sigma_{W_2}^2) - g_{20}n_0\big) \tag{S170w}$$

$$0 = -\sqrt{\zeta}\phi\big(\gamma + \sigma_{W_2}^2(\eta' - \zeta)\big)\big(g_6\sqrt{n_0}(\zeta\psi - \eta\psi - \zeta\sigma_{W_2}^2) - \sqrt{\zeta}g_{14}\psi\big) + g_{20}n_0\big(g_{13}\psi(\eta - \zeta)$$
$$- \phi(\sqrt{\zeta}g_6\sqrt{n_0} - 1)(\gamma + \sigma_{W_2}^2(\eta' - \zeta))\big) + g_{13}\psi(-(\zeta - \eta))\big(\psi(\zeta - \eta) - \zeta\sigma_{W_2}^2\big) \tag{S170x}$$

$$0 = \big(\psi(\zeta - \eta) - \zeta\sigma_{W_2}^2\big)\big(g_{13}\psi(\eta - \zeta) + \sqrt{\zeta}g_6\sqrt{n_0}\phi(\sigma_{W_2}^2(\zeta - \eta') - \gamma)\big) + n_0\big(g_{13}g_{20}\psi(\eta - \zeta)$$
$$+ \phi(g_{11} - \sqrt{\zeta}g_6 g_{20}\sqrt{n_0})(\gamma + \sigma_{W_2}^2(\eta' - \zeta))\big) + \zeta g_4\psi\phi\big(\gamma + \sigma_{W_2}^2(\eta' - \zeta)\big) \tag{S170y}$$

$$0 = \gamma g_7 n_0\phi + \sqrt{\zeta}g_{10}g_{13}\sqrt{n_0}\psi + \sqrt{\zeta}g_8 g_{15}\sqrt{n_0}\psi - \zeta g_7 n_0\phi\sigma_{W_2}^2 + \zeta g_6 g_8 n_0\sigma_{W_2}^2 + \zeta g_7 g_{13}n_0\sigma_{W_2}^2 + g_7 n_0\phi\eta'\sigma_{W_2}^2$$
$$+ \zeta g_6\phi\big(\gamma + \sigma_{W_2}^2(\eta' - \zeta)\big) + \zeta g_5 g_6\phi\big(\gamma + \sigma_{W_2}^2(\eta' - \zeta)\big) - \zeta^2 g_6 g_{13}\sigma_{W_2}^4 \tag{S170z}$$

$$0 = \gamma g_{17}n_0\phi + \sqrt{\zeta}g_{13}g_{18}\sqrt{n_0}\psi + \sqrt{\zeta}g_8 g_{19}\sqrt{n_0}\psi - \zeta g_{17}n_0\phi\sigma_{W_2}^2 + \zeta g_8 g_{14}n_0\sigma_{W_2}^2 + \zeta g_{13}g_{17}n_0\sigma_{W_2}^2 + g_{17}n_0\phi\eta'\sigma_{W_2}^2$$
$$+ \zeta g_{14}\phi\big(\gamma + \sigma_{W_2}^2(\eta' - \zeta)\big) + \zeta g_5 g_{14}\phi\big(\gamma + \sigma_{W_2}^2(\eta' - \zeta)\big) - \zeta^2 g_{13}g_{14}\sigma_{W_2}^4 \tag{S170aa}$$

$$0 = g_5\big(\psi(\zeta - \eta) - \zeta\sigma_{W_2}^2\big)\big(g_{13}\psi(\eta - \zeta) + \sqrt{\zeta}g_6\sqrt{n_0}\phi(\sigma_{W_2}^2(\zeta - \eta') - \gamma)\big) + n_0\big(\phi(g_3 - \sqrt{\zeta}(g_6 g_{12} + g_2 g_{16})\sqrt{n_0})$$
$$\big(\gamma + \sigma_{W_2}^2(\eta' - \zeta)\big) - (g_{12}g_{13} + g_8 g_{16})\psi(\zeta - \eta)\big) + \zeta g_4 g_5\psi\phi\big(\gamma + \sigma_{W_2}^2(\eta' - \zeta)\big) \tag{S170ab}$$

$$0 = \big(\psi(\zeta - \eta) - \zeta\sigma_{W_2}^2\big)\big(g_{13}\psi(\eta - \zeta) + \sqrt{\zeta}g_6\sqrt{n_0}\phi(\sigma_{W_2}^2(\zeta - \eta') - \gamma)\big) + g_5\big(g_{13}\psi(-(\zeta - \eta))(\psi(\zeta - \eta) - \zeta\sigma_{W_2}^2)$$
$$- \sqrt{\zeta}\phi(\gamma + \sigma_{W_2}^2(\eta' - \zeta))(g_6\sqrt{n_0}(\zeta\psi - \eta\psi - \zeta\sigma_{W_2}^2) - \sqrt{\zeta}g_{14}\psi)\big) + g_{11}n_0\phi\big(\gamma + \sigma_{W_2}^2(\eta' - \zeta)\big)$$
$$+ \zeta g_4\psi\phi\big(\gamma + \sigma_{W_2}^2(\eta' - \zeta)\big) \tag{S170ac}$$

$$0 = g_{12}n_0\big(g_{13}\psi(\eta - \zeta) - \phi(\sqrt{\zeta}g_6\sqrt{n_0} - 1)(\gamma + \sigma_{W_2}^2(\eta' - \zeta))\big) - g_{16}\big(g_8 n_0\psi(\zeta - \eta) + \sqrt{\zeta}g_7 n_0^{3/2}\phi(\gamma + \sigma_{W_2}^2(\eta' - \zeta))$$
$$+ \zeta g_{13}\psi(\zeta - \eta)\sigma_{W_2}^2\big) + g_5\big(g_{13}\psi(-(\zeta - \eta))(\psi(\zeta - \eta) - \zeta\sigma_{W_2}^2)$$
$$- \sqrt{\zeta}\phi(\gamma + \sigma_{W_2}^2(\eta' - \zeta))(g_6\sqrt{n_0}(\zeta\psi - \eta\psi - \zeta\sigma_{W_2}^2) - \sqrt{\zeta}g_{14}\psi)\big) \tag{S170ad}$$

$$0 = \gamma\sqrt{\zeta}g_7 g_{16}n_0^{3/2}\phi + \gamma\sqrt{\zeta}g_6 g_{20}n_0^{3/2}\phi + g_8 g_{16}n_0\psi(\zeta - \eta) + \zeta g_{13}g_{20}n_0\psi - \eta g_{13}g_{20}n_0\psi$$
$$+ g_{12}n_0\big(g_{13}\psi(\zeta - \eta) + \phi(\sqrt{\zeta}g_6\sqrt{n_0} - 1)(\gamma + \sigma_{W_2}^2(\eta' - \zeta))\big) - \zeta^{3/2}g_7 g_{16}n_0^{3/2}\phi\sigma_{W_2}^2 - \zeta^{3/2}g_6 g_{20}n_0^{3/2}\phi\sigma_{W_2}^2$$
$$+ \sqrt{\zeta}(g_7 g_{16} + g_6 g_{20})n_0^{3/2}\phi\eta'\sigma_{W_2}^2 + \zeta^2 g_{13}g_{16}\psi\sigma_{W_2}^2 - \zeta\eta g_{13}g_{16}\psi\sigma_{W_2}^2 \tag{S170ae}$$

$$0 = -\gamma g_8 n_0 - \sqrt{\zeta}g_{13}g_{18}\sqrt{n_0}\psi - \sqrt{\zeta}g_8 g_{19}\sqrt{n_0}\psi + \zeta g_{12}g_{13}n_0 + \zeta g_8 g_{16}n_0 + \zeta g_{13}g_{20}n_0 - \eta g_{12}g_{13}n_0$$
$$- \eta g_8 g_{16}n_0 - \eta g_{13}g_{20}n_0 + \zeta g_8 n_0\sigma_{W_2}^2 - \zeta g_8 g_{14}n_0\sigma_{W_2}^2 - \zeta g_{13}g_{17}n_0\sigma_{W_2}^2 - g_8 n_0\eta'\sigma_{W_2}^2 + \zeta^2 g_{13}g_{14}\sigma_{W_2}^4$$
$$+ \zeta^2 g_{13}g_{16}\sigma_{W_2}^2 + g_{13}(\zeta - \eta)\big(\psi(\zeta - \eta) - \zeta\sigma_{W_2}^2\big) + g_5 g_{13}(\zeta - \eta)\big(\zeta\psi - \eta\psi - \zeta\sigma_{W_2}^2\big) - \zeta\eta g_{13}g_{16}\sigma_{W_2}^2 \tag{S170af}$$

$$0 = \gamma\sqrt{\zeta}g_5 g_7 n_0^{3/2}\phi + \gamma\sqrt{\zeta}g_6 g_{11}n_0^{3/2}\phi + \zeta g_5 g_8 n_0\psi + \zeta g_{11}g_{13}n_0\psi - \eta g_5 g_8 n_0\psi - \eta g_{11}g_{13}n_0\psi$$
$$+ g_3 n_0\big(g_{13}\psi(\zeta - \eta) + \phi(\sqrt{\zeta}g_6\sqrt{n_0} - 1)(\gamma + \sigma_{W_2}^2(\eta' - \zeta))\big) + n_0\big(g_8\psi(\zeta - \eta)$$
$$+ \sqrt{\zeta}g_2\sqrt{n_0}\phi(\gamma + \sigma_{W_2}^2(\eta' - \zeta))\big) - \zeta^{3/2}g_5 g_7 n_0^{3/2}\phi\sigma_{W_2}^2 - \zeta^{3/2}g_6 g_{11}n_0^{3/2}\phi\sigma_{W_2}^2 + \sqrt{\zeta}g_5 g_7 n_0^{3/2}\phi\eta'\sigma_{W_2}^2$$
$$+ \sqrt{\zeta}g_6 g_{11}n_0^{3/2}\phi\eta'\sigma_{W_2}^2 + \zeta^2 g_5 g_{13}\psi\sigma_{W_2}^2 - \zeta\eta g_5 g_{13}\psi\sigma_{W_2}^2 \tag{S170ag}$$

$$0 = -\sqrt{\zeta}g_6\sqrt{n_0}\phi\big(\psi(\zeta - \eta) - \zeta\sigma_{W_2}^2\big)\big(\gamma + \sigma_{W_2}^2(\eta' - \zeta)\big) - \sqrt{\zeta}g_5 g_6\sqrt{n_0}\phi\big(\psi(\zeta - \eta) - \zeta\sigma_{W_2}^2\big)\big(\gamma + \sigma_{W_2}^2(\eta' - \zeta)\big)$$
$$+ g_9 n_0\psi\big(\gamma\phi + \sigma_{W_2}^2(\phi(\eta' - \zeta) + \zeta g_{13})\big) + \zeta g_4\psi\big(g_8 n_0\sigma_{W_2}^2 + g_5(\gamma + \sigma_{W_2}^2(\eta' - \zeta)) - \zeta g_{13}\sigma_{W_2}^4\big)$$
$$- \sqrt{\zeta}(g_2 g_{16} + g_6(g_{12} + g_{20}))n_0^{3/2}\phi\big(\gamma + \sigma_{W_2}^2(\eta' - \zeta)\big) + \zeta\psi\sigma_{W_2}^2\big(g_8 n_0 - \zeta g_{13}\sigma_{W_2}^2\big)$$
$$+ \zeta g_4\psi\phi\big(\gamma + \sigma_{W_2}^2(\eta' - \zeta)\big) \tag{S170ah}$$

$$0 = -\gamma\sqrt{\zeta}g_6 g_{12}n_0^{3/2}\phi - \gamma\sqrt{\zeta}g_7 g_{16}n_0^{3/2}\phi - \gamma\sqrt{\zeta}g_6 g_{20}n_0^{3/2}\phi + \gamma g_{17}n_0\psi\phi - \sqrt{\zeta}\phi\big(\gamma + \sigma_{W_2}^2(\eta' - \zeta)\big)$$
$$\big(g_6\sqrt{n_0}(\zeta\psi - \eta\psi - \zeta\sigma_{W_2}^2) - \sqrt{\zeta}g_{14}\psi\big) - \sqrt{\zeta}g_5\phi\big(\gamma + \sigma_{W_2}^2(\eta' - \zeta)\big)\big(g_6\sqrt{n_0}(\zeta\psi - \eta\psi - \zeta\sigma_{W_2}^2) - \sqrt{\zeta}g_{14}\psi\big)$$
$$+ \zeta^{3/2}g_6 g_{12}n_0^{3/2}\phi\sigma_{W_2}^2 + \zeta^{3/2}g_7 g_{16}n_0^{3/2}\phi\sigma_{W_2}^2 + \zeta^{3/2}g_6 g_{20}n_0^{3/2}\phi\sigma_{W_2}^2 - n_0\phi\eta'\sigma_{W_2}^2\big(\sqrt{\zeta}(g_7 g_{16} + g_6(g_{12} + g_{20}))\big)\sqrt{n_0}$$

$$- g_{17}\psi) + \zeta g_8 g_{14} n_0 \psi \sigma_{W_2}^2 + \zeta g_{13} g_{17} n_0 \psi \sigma_{W_2}^2 - \zeta g_{17} n_0 \psi \phi \sigma_{W_2}^2 - \zeta^2 g_{13} g_{14} \psi \sigma_{W_2}^4 \tag{S170ai}$$

$$0 = -\gamma\sqrt{\zeta}g_{12}g_{15}n_0^{3/2}\phi - \gamma\sqrt{\zeta}g_{10}g_{16}n_0^{3/2}\phi - \gamma\sqrt{\zeta}g_{15}g_{20}n_0^{3/2} + \gamma g_{18}n_0\psi\phi - \sqrt{\zeta}\phi\big(\gamma + \sigma_{W_2}^2(\eta' - \zeta)\big)$$

$$\big(g_{15}\sqrt{n_0}(\zeta\psi - \eta\psi - \zeta\sigma_{W_2}^2) - \sqrt{\zeta}g_{19}\psi\big) - \sqrt{\zeta}g_5\phi\big(\gamma + \sigma_{W_2}^2(\eta' - \zeta)\big)\big(g_{15}\sqrt{n_0}(\zeta\psi - \eta\psi - \zeta\sigma_{W_2}^2) - \zeta g_{18}n_0\psi\phi\sigma_{W_2}^2$$

$$- \sqrt{\zeta}g_{19}\psi\big) + \zeta^{3/2}g_{12}g_{15}n_0^{3/2}\phi\sigma_{W_2}^2 + \zeta^{3/2}g_{10}g_{16}n_0^{3/2}\phi\sigma_{W_2}^2 + \zeta^{3/2}g_{15}g_{20}n_0^{3/2}\phi\sigma_{W_2}^2 - \zeta^2 g_{13}g_{19}\psi\sigma_{W_2}^4$$

$$- n_0\phi\eta'\sigma_{W_2}^2\big(\sqrt{\zeta}\big(g_{10}g_{16} + g_{15}(g_{12} + g_{20})\big)\sqrt{n_0} - g_{18}\psi\big) + \zeta g_{13}g_{18}n_0\psi\sigma_{W_2}^2 + \zeta g_8 g_{19}n_0\psi\sigma_{W_2}^2 \tag{S170aj}$$

After some straightforward algebra, one can eliminate all $g_i$ except for $g_6$ and $g_{13}$, which satisfy coupled polynomial equations. Those equations can be shown to be identical to eqn. (S66) by invoking the change of variables,

$$g_6 = -\frac{\sqrt{\zeta}\psi}{\sqrt{n_0}\phi}\tau_2, \quad \text{and} \quad g_{13} = \big(\gamma + \sigma_{W_2}^2(\eta' - \zeta)\big)\tau_1. \tag{S171}$$

In terms of $\tau_1$, $\tau_2$, and their derivatives $\tau_1'$ (S87), $\tau_2'$ (S88), the error $E_{33}$ is given by,

$$E_{33} = \sigma_{W_2}^2\left[\big(\tau_1 + (\sigma_{W_2}^2(\eta' - \zeta) + \gamma)\tau_1' + \sigma_{W_2}^2\zeta\tau_2'\big)/\tau_1^2 - \eta\right] - E_{22}. \tag{S172}$$

## S6    Exact asymptotics for bias and variance terms

Following Sec. S1, for each random variable in question we introduce an iid copy of it denoted by a tilde. Using this simplifying notation and recalling $P = \{W_1, W_2\}$ we have,

$$B = \mathbb{E}_{(\mathbf{x},y)}(y - \mathbb{E}_{(P,X,\varepsilon)}\hat{y}(\mathbf{x}; P, X, \varepsilon))^2 \tag{S173}$$

$$= \mathbb{E}_{(\mathbf{x},y)}\mathbb{E}_{(P,X,\varepsilon)}\mathbb{E}_{(\tilde{P},\tilde{X},\tilde{\varepsilon})}(y - \hat{y}(\mathbf{x}; P, X, \varepsilon))(y - \hat{y}(\mathbf{x}; \tilde{P}, \tilde{X}, \tilde{\varepsilon})) \tag{S174}$$

$$= 1 + E_{21} + H_{000}, \tag{S175}$$

where $E_{21}$ was computed previously and $H_{000}$ and the other $H_{ijk}$(also defined above) are,

$$H_{000} = \mathbb{E}\hat{y}(\mathbf{x}; P, X, \varepsilon)\hat{y}(\mathbf{x}; \tilde{P}, \tilde{X}, \tilde{\varepsilon}) \tag{S176}$$

$$H_{001} = \mathbb{E}\hat{y}(\mathbf{x}; P, X, \varepsilon)\hat{y}(\mathbf{x}; \tilde{P}, \tilde{X}, \varepsilon) \tag{S177}$$

$$H_{010} = \mathbb{E}\hat{y}(\mathbf{x}; P, X, \varepsilon)\hat{y}(\mathbf{x}; \tilde{P}, X, \tilde{\varepsilon}) \tag{S178}$$

$$H_{011} = \mathbb{E}\hat{y}(\mathbf{x}; P, X, \varepsilon)\hat{y}(\mathbf{x}; \tilde{P}, X, \varepsilon) \tag{S179}$$

$$H_{100} = \mathbb{E}\hat{y}(\mathbf{x}; P, X, \varepsilon)\hat{y}(\mathbf{x}; P, \tilde{X}, \tilde{\varepsilon}) \tag{S180}$$

$$H_{101} = \mathbb{E}\hat{y}(\mathbf{x}; P, X, \varepsilon)\hat{y}(\mathbf{x}; P, \tilde{X}, \varepsilon) \tag{S181}$$

$$H_{110} = \mathbb{E}\hat{y}(\mathbf{x}; P, X, \varepsilon)\hat{y}(\mathbf{x}; P, X, \tilde{\varepsilon}) \tag{S182}$$

$$H_{111} = \mathbb{E}\hat{y}(\mathbf{x}; P, X, \varepsilon)\hat{y}(\mathbf{x}; P, X, \varepsilon), \tag{S183}$$

where the expectations are over $\mathbf{x}, P, X, \varepsilon, \tilde{P}, \tilde{X},$ and $\tilde{\varepsilon}$. Recalling the definition of $\hat{y}$,

$$\hat{y}(\mathbf{x}; P, X, \varepsilon) := N_0(\mathbf{x}; P) + (Y(X, \epsilon) - N_0(X; P))K(X, X; P)^{-1}K(X, \mathbf{x}; P) \tag{S184}$$

and the techniques described in the previous section, it is straightforward to analyze each of the above terms, which we do in the following subsections. To aid those calculations, we first note that, similar to above, we can write,

$$\mathbb{E}_{\mathbf{x}}K(X, \mathbf{x}; P)K(\mathbf{x}, \tilde{X}; \tilde{P}) = \frac{\sigma_{W_2}^4\zeta^2}{n_0^2}X^\top\tilde{X} + \frac{\sigma_{W_2}^2\zeta^{3/2}}{n_0^{3/2}n_1}(X^\top W_1^T\tilde{F} + F^\top\tilde{W}_1\tilde{X}) + \frac{\zeta}{n_0 n_1^2}F^\top W_1\tilde{W}_1^\top\tilde{F} \tag{S185}$$

$$= \big(\frac{\sigma_{W_2}^2\zeta}{n_0}X^\top + \frac{\zeta}{\sqrt{n_0}n_1}F^\top W_1\big)\big(\frac{\sigma_{W_2}^2\zeta}{n_0}\tilde{X}^\top + \frac{\sqrt{\zeta}}{\sqrt{n_0}n_1}\tilde{F}^\top\tilde{W}_1\big)^\top \tag{S186}$$

$$\mathbb{E}_{\mathbf{x}} K(X, \mathbf{x}; P) K(\mathbf{x}, X; \tilde{P}) = (\frac{\sigma_{W_2}^2 \zeta}{n_0} X^\top + \frac{\zeta}{\sqrt{n_0 n_1}} F^\top W_1)(\frac{\sigma_{W_2}^2 \zeta}{n_0} X^\top + \frac{\sqrt{\zeta}}{\sqrt{n_0 n_1}} f(\tilde{W}_1 X)^\top \tilde{W}_1)^\top \tag{S187}$$

$$\mathbb{E}_{\mathbf{x}} K(X, \mathbf{x}; P) K(\mathbf{x}, \tilde{X}; P) = (\frac{\sigma_{W_2}^2 \zeta}{n_0} X^\top + \frac{\zeta}{\sqrt{n_0 n_1}} F^\top W_1)(\frac{\sigma_{W_2}^2 \zeta}{n_0} \tilde{X}^\top + \frac{\sqrt{\zeta}}{\sqrt{n_0 n_1}} f(W_1 \tilde{X})^\top W_1)^\top + \frac{\eta - \zeta}{n_1^2} F^\top f(W_1 \tilde{X}) \tag{S188}$$

**S6.1** $H_{000}$

$$H_{000} = \mathbb{E} \hat{y}(\mathbf{x}; P, X, \varepsilon) \hat{y}(\mathbf{x}; \tilde{P}, \tilde{X}, \tilde{\varepsilon}) \tag{S189}$$

$$= \mathbb{E} K(\mathbf{x}, \tilde{X}; \tilde{P}) K(\tilde{X}, \tilde{X}; \tilde{P})^{-1} Y(\tilde{X}, \tilde{\varepsilon})^\top Y(X, \varepsilon) K(X, X; P)^{-1} K(X, \mathbf{x}; P) \tag{S190}$$

$$= \mathbb{E} \operatorname{tr} \left( K(\tilde{X}, \tilde{X}; \tilde{P})^{-1} \tilde{X}^\top X K(X, X; P)^{-1} K(X, \mathbf{x}; P) K(\mathbf{x}, \tilde{X}; \tilde{P}) \right) \tag{S191}$$

$$= \mathbb{E} \operatorname{tr} \left( K(\tilde{X}, \tilde{X}; \tilde{P})^{-1} \tilde{X}^\top X K(X, X; P)^{-1} (\frac{\sigma_{W_2}^2 \zeta}{n_0} X^\top + \frac{\sqrt{\zeta}}{\sqrt{n_0 n_1}} F^\top W_1)(\frac{\sigma_{W_2}^2 \zeta}{n_0} \tilde{X}^\top + \frac{\sqrt{\zeta}}{\sqrt{n_0 n_1}} \tilde{F}^\top \tilde{W}_1)^\top \right) \tag{S192}$$

$$= \operatorname{tr} \left( X K^{-1} (\frac{\sigma_{W_2}^2 \zeta}{n_0} X^\top + \frac{\sqrt{\zeta}}{\sqrt{n_0 n_1}} F^\top W_1) \right)^2 \tag{S193}$$

$$\equiv E_4 \tag{S194}$$

A linear pencil for $E_4$ follows from the representation,

$$E_4 = \operatorname{tr}(U_4^T Q_4^{-1} V_4)^2, \tag{S195}$$

where,

$$U_4^T = \begin{pmatrix} 0 & \frac{\zeta I_{n_0} \sigma_{W_2}^2}{n_0} & 0 & 0 & \frac{\sqrt{\zeta} I_m}{\sqrt{n_0 n_1}} \end{pmatrix}, \quad V_4^T = \begin{pmatrix} 0 & -\frac{n_0 I_{n_0}}{\zeta \sigma_{W_2}^2} & 0 & 0 & \frac{\sqrt{n_0 n_1} I_m}{\sqrt{\zeta}} \end{pmatrix} \tag{S196}$$

and,

$$Q_4 = \begin{pmatrix} I_m \left( \gamma + \sigma_{W_2}^2 (\eta' - \zeta) \right) & \frac{\zeta X^\top \sigma_{W_2}^2}{n_0} & \frac{\sqrt{\eta - \zeta} \Theta_F^\top}{n_1} & \frac{\sqrt{\zeta} X^\top}{\sqrt{n_0 n_1}} & 0 \\ -X & I_{n_0} & 0 & 0 & 0 \\ -\sqrt{\eta - \zeta} \Theta_F & -\frac{\sqrt{\zeta} W_1}{\sqrt{n_0}} & I_{n_1} & 0 & 0 \\ 0 & 0 & -W_1^\top & I_{n_0} & 0 \\ 0 & 0 & 0 & 0 & I_m \end{pmatrix}. \tag{S197}$$

The equations satisfied by the operator-valued Stieltjes transform $G$ of $\bar{Q}_4$ induce the following structure on $G$,

$$G = \begin{pmatrix} 0 & G_{12} \\ G_{12}^\top & 0 \end{pmatrix}, \tag{S198}$$

where,

$$G_{12} = \begin{pmatrix} g_3 & 0 & 0 & 0 & 0 \\ 0 & g_4 & 0 & g_2 & 0 \\ 0 & 0 & g_6 & 0 & 0 \\ 0 & g_7 & 0 & g_5 & 0 \\ 0 & 0 & 0 & 0 & g_1 \end{pmatrix}, \tag{S199}$$

and the independent entry-wise component functions $g_i$ give the error $E_4$ through the relation,

$$E_4 = (g_1 - g_4)^2, \tag{S200}$$

and themselves satisfy the following system of polynomial equations,

$$0 = 1 - g_1 \tag{S201a}$$

$$0 = \sqrt{\zeta}g_4 g_6 \sqrt{n_0} - g_7 \psi \tag{S201b}$$

$$0 = \sqrt{\zeta}g_2 g_6 \sqrt{n_0} - g_5 \psi + \psi \tag{S201c}$$

$$0 = -\sqrt{\zeta}g_3 g_4 \psi - g_2 \sqrt{n_0}\phi\big(\gamma + \sigma_{W_2}^2(\eta' - \zeta)\big) \tag{S201d}$$

$$0 = -\sqrt{\zeta}g_3 g_7 \psi - (g_5 - 1)\sqrt{n_0}\phi\big(\gamma + \sigma_{W_2}^2(\eta' - \zeta)\big) \tag{S201e}$$

$$0 = g_2 \sqrt{n_0}\phi\big(\gamma + \sigma_{W_2}^2(\eta' - \zeta)\big) + g_3\big(\sqrt{\zeta}g_5 \psi + \zeta g_2 \sqrt{n_0}\sigma_{W_2}^2\big) \tag{S201f}$$

$$0 = (g_4 - 1)\sqrt{n_0}\phi\big(\gamma + \sigma_{W_2}^2(\eta' - \zeta)\big) + g_3\big(\sqrt{\zeta}g_7 \psi + \zeta g_4 \sqrt{n_0}\sigma_{W_2}^2\big) \tag{S201g}$$

$$0 = \sqrt{\zeta}g_5 g_6 \sqrt{n_0}\phi\big(\sigma_{W_2}^2(\zeta - \eta') - \gamma\big) + g_7 \psi\big(\gamma\phi + \sigma_{W_2}^2\big(-\zeta\phi + \phi\eta' + \zeta g_3\big)\big) \tag{S201h}$$

$$0 = g_6\big(g_3 \psi(\zeta - \eta) + \phi\big(\sqrt{\zeta}g_2 \sqrt{n_0} - 1\big)\big(\gamma + \sigma_{W_2}^2(\eta' - \zeta)\big)\big) + \phi\big(\gamma + \sigma_{W_2}^2(\eta' - \zeta)\big) \tag{S201i}$$

$$0 = g_3\big(\sqrt{\zeta}g_7 \psi + \sqrt{n_0}\big(\gamma + g_6(\eta - \zeta) + \sigma_{W_2}^2\big(\eta' + \zeta(g_4 - 1)\big)\big)\big) - \sqrt{n_0}\big(\gamma + \sigma_{W_2}^2(\eta' - \zeta)\big) \tag{S201j}$$

$$0 = g_4 \psi\big(\gamma\phi + \sigma_{W_2}^2\big(-\zeta\phi + \phi\eta' + \zeta g_3\big)\big) - \phi\big(\sqrt{\zeta}g_2 g_6 \sqrt{n_0} + \psi\big)\big(\gamma + \sigma_{W_2}^2(\eta' - \zeta)\big) \tag{S201k}$$

After some straightforward algebra, one can eliminate all $g_i$ except for $g_2$ and $g_3$, which satisfy coupled polynomial equations. Those equations can be shown to be identical to eqn. (S66) by invoking the change of variables,

$$g_2 = -\frac{\sqrt{\zeta}\psi}{\sqrt{n_0}\phi}\tau_2, \quad \text{and} \quad g_3 = \big(\gamma + \sigma_{W_2}^2(\eta' - \zeta)\big)\tau_1. \tag{S201l}$$

In terms of the related variables defined in eqn. (S89), the error $E_4$ is given by,

$$E_4 = \tilde{\tau}_2^2. \tag{S201m}$$

## S6.2 $H_{001}$

$$H_{001} = \mathbb{E}\hat{y}(\mathbf{x}; P, X, \varepsilon)\hat{y}(\mathbf{x}; \tilde{P}, \tilde{X}, \varepsilon) \tag{S202}$$

$$= \mathbb{E}K(\mathbf{x}, \tilde{X}; \tilde{P})K(\tilde{X}, \tilde{X}; \tilde{P})^{-1}Y(\tilde{X}, \varepsilon)^\top Y(X, \varepsilon)K(X, X; P)^{-1}K(X, \mathbf{x}; P) \tag{S203}$$

$$= \mathbb{E}\operatorname{tr}\big(K(\tilde{X}, \tilde{X}; \tilde{P})^{-1}\tilde{X}^\top X K(X, X; P)^{-1}K(X, \mathbf{x}; P)K(\mathbf{x}, \tilde{X}; \tilde{P})\big) \tag{S204}$$

$$= H_{000} \tag{S205}$$

## S6.3 $H_{010}$

$$H_{010} = \mathbb{E}\hat{y}(\mathbf{x}; P, X, \varepsilon)\hat{y}(\mathbf{x}; \tilde{P}, X, \tilde{\varepsilon}) \tag{S206}$$

$$= \mathbb{E}K(\mathbf{x}, X; \tilde{P})K(X, X; \tilde{P})^{-1}Y(X, \tilde{\varepsilon})^\top Y(X, \varepsilon)K(X, X; P)^{-1}K(X, \mathbf{x}; P) \tag{S207}$$

$$= \mathbb{E}\operatorname{tr}\big(K(X, X; \tilde{P})^{-1}X^\top X K(X, X; P)^{-1}K(X, \mathbf{x}; P)K(\mathbf{x}, X; \tilde{P})\big) \tag{S208}$$

$$= \mathbb{E}\operatorname{tr}\Big(K(X, X; \tilde{P})^{-1}X^\top X K(X, X; P)^{-1}K(X, \mathbf{x}; P)K(\mathbf{x}, X; \tilde{P}) \tag{S209}$$

$$\times \big(\frac{\sigma_{W_2}^2\zeta}{n_0}X^\top + \frac{\zeta}{\sqrt{n_0 n_1}}F^\top W_1\big)\big(\frac{\sigma_{W_2}^2\zeta}{n_0}X^\top + \frac{\sqrt{\zeta}}{\sqrt{n_0 n_1}}f(\tilde{W}_1 X)^\top \tilde{W}_1)^\top\big)\Big) \tag{S210}$$

$$\equiv E_5. \tag{S211}$$

A linear pencil for $E_5$ follows from the representation,

$$E_5 = \operatorname{tr}(U_5^T Q_5^{-1} V_5), \tag{S212}$$

where,

$$U_5^T = \begin{pmatrix} 0 & \frac{I_{n_0}}{m} & 0 & 0 & 0 & 0 & 0 & 0 & 0 \end{pmatrix}, \quad V_5^T = \begin{pmatrix} 0 & 0 & 0 & 0 & 0 & 0 & 0 & 0 & -\frac{\sqrt{n_0 n_1} I_{n_0}}{\sqrt{\zeta}} \end{pmatrix} \tag{S213}$$

and,

$$Q_5 = \begin{pmatrix}
I_m\left(\gamma+\sigma_{W_2}^2(\eta'-\zeta)\right) & 0 & \frac{\zeta X^\top \sigma_{W_2}^2}{n_0} & \frac{\sqrt{\eta-\zeta}\Theta_F^\top}{n_1} & \frac{\sqrt{\zeta}X^\top}{\sqrt{n_0}n_1} & -\frac{\zeta^2 m X^\top \sigma_{W_2}^4}{n_0^2} & 0 & 0 & 0 \\
-X & I_{n_0} & 0 & 0 & 0 & 0 & 0 & 0 & 0 \\
-X & 0 & I_{n_0} & 0 & 0 & 0 & 0 & \frac{\sqrt{\zeta}m\bar{W}_1^\top}{\sqrt{n_0}n_1} & 0 \\
-\sqrt{\eta-\zeta}\Theta_F & 0 & -\frac{\sqrt{\zeta}W_1}{\sqrt{n_0}} & I_{n_1} & 0 & \frac{\zeta^{3/2}m W_1 \sigma_{W_2}^2}{n_0^{3/2}} & 0 & 0 & 0 \\
0 & 0 & 0 & -W_1^\top & I_{n_0} & 0 & 0 & 0 & 0 \\
0 & 0 & 0 & 0 & 0 & I_{n_0} & -X & 0 & 0 \\
0 & 0 & 0 & 0 & 0 & \frac{\zeta X^\top \sigma_{W_2}^2}{n_0} & I_m\left(\gamma+\sigma_{W_2}^2(\eta'-\zeta)\right) & \frac{\sqrt{\eta-\zeta}\Theta_F^\top}{n_1} & \frac{\sqrt{\zeta}X^\top}{\sqrt{n_0}n_1} \\
0 & 0 & 0 & 0 & 0 & -\frac{\sqrt{\zeta}\bar{W}_1}{\sqrt{n_0}} & -\sqrt{\eta-\zeta}\tilde{\Theta}_F & I_{n_1} & 0 \\
0 & 0 & 0 & 0 & 0 & 0 & 0 & -\tilde{W}_1^\top & I_{n_0}
\end{pmatrix}. \tag{S214}$$

The equations satisfied by the operator-valued Stieltjes transform $G$ of $\bar{Q}_5$ induce the following structure on $G$,

$$G = \begin{pmatrix} 0 & G_{12} \\ G_{12}^\top & 0 \end{pmatrix}, \tag{S215}$$

where,

$$G_{12} = \begin{pmatrix}
g_9 & 0 & 0 & 0 & 0 & 0 & g_6 & 0 & 0 \\
0 & g_1 & g_5 & 0 & g_8 & g_3 & 0 & 0 & g_2 \\
0 & 0 & g_{10} & 0 & g_8 & g_{13} & 0 & 0 & g_7 \\
0 & 0 & 0 & g_{12} & 0 & 0 & 0 & 0 & 0 \\
0 & 0 & g_{15} & 0 & g_{11} & g_{14} & 0 & 0 & g_4 \\
0 & 0 & 0 & 0 & 0 & g_{10} & 0 & 0 & g_8 \\
0 & 0 & 0 & 0 & 0 & 0 & g_9 & 0 & 0 \\
0 & 0 & 0 & 0 & 0 & 0 & 0 & g_{12} & 0 \\
0 & 0 & 0 & 0 & 0 & g_{15} & 0 & 0 & g_{11}
\end{pmatrix}, \tag{S216}$$

and the independent entry-wise component functions $g_i$ give the error $E_5$ through the relation,

$$E_5 = -\frac{g_2\sqrt{n_0}\phi}{\sqrt{\zeta}\psi}, \tag{S217}$$

and themselves satisfy the following system of polynomial equations,

$$0 = 1 - g_1 \tag{S218a}$$

$$0 = \sqrt{\zeta}g_{10}g_{12}\sqrt{n_0} - g_{15}\psi \tag{S218b}$$

$$0 = \sqrt{\zeta}g_8 g_{12}\sqrt{n_0} - g_{11}\psi + \psi \tag{S218c}$$

$$0 = \sqrt{\zeta}g_{12}\sqrt{n_0}\left(g_7\phi - \zeta g_8 \sigma_{W_2}^2\right) - g_4\psi\phi \tag{S218d}$$

$$0 = \sqrt{\zeta}g_{12}\sqrt{n_0}\left(g_{13}\phi - \zeta g_{10}\sigma_{W_2}^2\right) - g_{14}\psi\phi \tag{S218e}$$

$$0 = -\sqrt{\zeta}g_9 g_{10}\psi - g_8\sqrt{n_0}\phi\left(\gamma + \sigma_{W_2}^2(\eta'-\zeta)\right) \tag{S218f}$$

$$0 = -\sqrt{\zeta}g_9 g_{15}\psi - (g_{11}-1)\sqrt{n_0}\phi\left(\gamma + \sigma_{W_2}^2(\eta'-\zeta)\right) \tag{S218g}$$

$$0 = -\sqrt{\zeta}g_1 g_9\psi - \sqrt{\zeta}g_5 g_9\psi - g_8\sqrt{n_0}\phi\left(\gamma + \sigma_{W_2}^2(\eta'-\zeta)\right) \tag{S218h}$$

$$0 = -\sqrt{\zeta}g_6 g_{10}\psi - \sqrt{\zeta}g_9 g_{13}\psi - g_7\sqrt{n_0}\phi\left(\gamma + \sigma_{W_2}^2(\eta'-\zeta)\right) \tag{S218i}$$

$$0 = -\sqrt{\zeta}g_9 g_{14}\psi - \sqrt{\zeta}g_6 g_{15}\psi - g_4\sqrt{n_0}\phi\big(\gamma + \sigma_{W_2}^2(\eta' - \zeta)\big) \tag{S218j}$$

$$0 = g_8\sqrt{n_0}\phi\big(\gamma + \sigma_{W_2}^2(\eta' - \zeta)\big) + g_9\big(\sqrt{\zeta}g_{11}\psi + \zeta g_8\sqrt{n_0}\sigma_{W_2}^2\big) \tag{S218k}$$

$$0 = g_5\sqrt{n_0}\phi\big(\gamma + \sigma_{W_2}^2(\eta' - \zeta)\big) + g_9\big(\sqrt{\zeta}g_{15}\psi + \zeta g_{10}\sqrt{n_0}\sigma_{W_2}^2\big) \tag{S218l}$$

$$0 = (g_{10} - 1)\sqrt{n_0}\phi\big(\gamma + \sigma_{W_2}^2(\eta' - \zeta)\big) + g_9\big(\sqrt{\zeta}g_{15}\psi + \zeta g_{10}\sqrt{n_0}\sigma_{W_2}^2\big) \tag{S218m}$$

$$0 = -\sqrt{\zeta}\big((g_1 + g_5)g_6 + g_3 g_9\big)\psi - \gamma g_2\sqrt{n_0}\phi + \zeta g_2\sqrt{n_0}\phi\sigma_{W_2}^2 + g_2\sqrt{n_0}(-\phi)\eta'\sigma_{W_2}^2 \tag{S218n}$$

$$0 = \sqrt{\zeta}g_{11}g_{12}\sqrt{n_0}\phi\big(\sigma_{W_2}^2(\zeta - \eta') - \gamma\big) + g_{15}\psi\big(\gamma\phi + \sigma_{W_2}^2(-\zeta\phi + \phi\eta' + \zeta g_9)\big) \tag{S218o}$$

$$0 = g_{12}\big(g_9\psi(\zeta - \eta) + \phi\big(\sqrt{\zeta}g_8\sqrt{n_0} - 1\big)\big(\gamma + \sigma_{W_2}^2(\eta' - \zeta)\big)\big) + \phi\big(\gamma + \sigma_{W_2}^2(\eta' - \zeta)\big) \tag{S218p}$$

$$0 = g_{10}\psi\big(\gamma\phi + \sigma_{W_2}^2(-\zeta\phi + \phi\eta' + \zeta g_9)\big) - \phi\big(\sqrt{\zeta}g_8 g_{12}\sqrt{n_0} + \psi\big)\big(\gamma + \sigma_{W_2}^2(\eta' - \zeta)\big) \tag{S218q}$$

$$0 = g_9\big(\sqrt{\zeta}g_{15}\psi + \sqrt{n_0}\big(\gamma + g_{12}(\eta - \zeta) + \sigma_{W_2}^2(\eta' + \zeta(g_{10} - 1))\big)\big) - \sqrt{n_0}\big(\gamma + \sigma_{W_2}^2(\eta' - \zeta)\big) \tag{S218r}$$

$$0 = -\sqrt{\zeta}g_8 g_{12}\sqrt{n_0}\phi\big(\gamma + \sigma_{W_2}^2(\eta' - \zeta)\big) + g_5\psi\big(\gamma\phi + \sigma_{W_2}^2(-\zeta\phi + \phi\eta' + \zeta g_9)\big) + \zeta g_1 g_9\psi\sigma_{W_2}^2 \tag{S218s}$$

$$0 = \sqrt{\zeta}g_4 g_9\psi\phi + \sqrt{n_0}\big(g_2\phi^2\big(\gamma + \sigma_{W_2}^2(\eta' - \zeta)\big) + \zeta g_9\sigma_{W_2}^2\big(g_7\phi - \zeta g_8\sigma_{W_2}^2\big)\big) + g_6\big(\sqrt{\zeta}g_{11}\psi\phi + \zeta g_8\sqrt{n_0}\phi\sigma_{W_2}^2\big) \tag{S218t}$$

$$0 = g_6\phi\big(\sqrt{\zeta}g_{15}\psi + \sqrt{n_0}\big(\gamma + g_{12}(\eta - \zeta) + \sigma_{W_2}^2(\eta' + \zeta(g_{10} - 1))\big)\big)$$
$$+ g_9\big(\sqrt{\zeta}g_{14}\psi\phi + \zeta\sqrt{n_0}\sigma_{W_2}^2\big(g_{13}\phi - \zeta g_{10}\sigma_{W_2}^2\big)\big) \tag{S218u}$$

$$0 = \phi\big(g_3\sqrt{n_0}\phi\big(\gamma + \sigma_{W_2}^2(\eta' - \zeta)\big) + g_6\big(\sqrt{\zeta}g_{15}\psi + \zeta g_{10}\sqrt{n_0}\sigma_{W_2}^2\big) + g_9\big(\sqrt{\zeta}g_{14}\psi\phi + \zeta\sqrt{n_0}\sigma_{W_2}^2\big(g_{13}\phi - \zeta g_{10}\sigma_{W_2}^2\big)\big) \tag{S218v}$$

$$0 = \phi\big(\sqrt{n_0}\big(\zeta g_{10}g_{12} + g_{13}\phi\big)\big(\gamma + \sigma_{W_2}^2(\eta' - \zeta)\big) + g_6\big(\sqrt{\zeta}g_{15}\psi + \zeta g_{10}\sqrt{n_0}\sigma_{W_2}^2\big)\big)$$
$$+ g_9\big(\sqrt{\zeta}g_{14}\psi\phi + \zeta\sqrt{n_0}\sigma_{W_2}^2\big(g_{13}\phi - \zeta g_{10}\sigma_{W_2}^2\big)\big) \tag{S218w}$$

$$0 = \sqrt{\zeta}g_4 g_9\psi\phi + \sqrt{n_0}\big(g_7\phi\big(\gamma\phi + \sigma_{W_2}^2(-\zeta\phi + \phi\eta' + \zeta g_9)\big) + \zeta g_8\big(g_{12}\phi\big(\gamma + \sigma_{W_2}^2(\eta' - \zeta)\big) - \zeta g_9\sigma_{W_2}^4\big)\big)$$
$$+ g_6\big(\sqrt{\zeta}g_{11}\psi\phi + \zeta g_8\sqrt{n_0}\phi\sigma_{W_2}^2\big) \tag{S218x}$$

$$0 = \sqrt{\zeta}g_{12}\phi\big(\gamma + \sigma_{W_2}^2(\eta' - \zeta)\big)\big(\sqrt{\zeta}g_{15}\psi + \sqrt{n_0}\big(\zeta g_{11}\sigma_{W_2}^2 - g_4\phi\big)\big) + g_{14}\psi\phi\big(\gamma\phi + \sigma_{W_2}^2(-\zeta\phi + \phi\eta' + \zeta g_9)\big)$$
$$+ \zeta g_{15}\psi\sigma_{W_2}^2\big(g_6\phi - \zeta g_9\sigma_{W_2}^2\big) \tag{S218y}$$

$$0 = \phi\big(g_{13}\psi\big(\gamma\phi + \sigma_{W_2}^2(-\zeta\phi + \phi\eta' + \zeta g_9)\big) - \sqrt{\zeta}g_{12}\sqrt{n_0}\big(g_7 - \zeta g_8\sigma_{W_2}^2\big)\big(\gamma + \sigma_{W_2}^2(\eta' - \zeta)\big)\big)$$
$$+ \zeta g_{10}\psi\big(g_{12}\phi\big(\gamma + \sigma_{W_2}^2(\eta' - \zeta)\big) - \zeta g_9\sigma_{W_2}^4 + g_6\phi\sigma_{W_2}^2\big) \tag{S218z}$$

$$0 = -\sqrt{\zeta}g_2 g_{12}\sqrt{n_0}\phi^2\big(\gamma + \sigma_{W_2}^2(\eta' - \zeta)\big) + \zeta\sigma_{W_2}^2\big(\sqrt{\zeta}g_8 g_{12}\sqrt{n_0}\phi\big(\gamma + \sigma_{W_2}^2(\eta' - \zeta)\big) + g_1\psi\big(g_6\phi - \zeta g_9\sigma_{W_2}^2\big)\big)$$
$$+ g_3\psi\phi\big(\gamma\phi + \sigma_{W_2}^2\big(\phi(\eta' - \zeta) + \zeta g_9\big)\big) + \zeta g_5\psi\big(g_{12}\phi\big(\gamma + \sigma_{W_2}^2(\eta' - \zeta)\big) - \zeta g_9\sigma_{W_2}^4 + g_6\phi\sigma_{W_2}^2\big) \tag{S218aa}$$

After some straightforward algebra, one can eliminate all $g_i$ except for $g_8$ and $g_9$, which satisfy coupled polynomial equations. Those equations can be shown to be identical to eqn. (S66) by invoking the change of variables,

$$g_8 = -\frac{\sqrt{\zeta}\psi}{\sqrt{n_0}\phi}\tau_2, \quad \text{and} \quad g_9 = \big(\gamma + \sigma_{W_2}^2(\eta' - \zeta)\big)\tau_1. \tag{S219}$$

In terms of the related variables defined in eqn. (S89), the error $E_4$ is given by,

$$E_5 = \tilde{\tau}_2^2(1 + \phi + 2\tilde{\tau}_2\phi)/(1 - \tilde{\tau}_2^2\phi). \tag{S220}$$

**S6.4** $H_{011}$

$$H_{011} = \mathbb{E}\hat{y}(\mathbf{x}; P, X, \varepsilon)\hat{y}(\mathbf{x}; \tilde{P}, X, \tilde{\varepsilon}) \tag{S221}$$
$$= \mathbb{E}K(\mathbf{x}, X; \tilde{P})K(X, X; \tilde{P})^{-1}Y(X, \tilde{\varepsilon})^{\top}Y(X, \varepsilon)K(X, X; P)^{-1}K(X, \mathbf{x}; P) \tag{S222}$$

$$= \mathbb{E}\,\mathrm{tr}\left(K(X,X;\tilde{P})^{-1}(X^\top X + \sigma_\varepsilon^2 n_1 I_m)K(X,X;P)^{-1}K(X,\mathbf{x};P)K(\mathbf{x},X;\tilde{P})\right) \tag{S223}$$

$$= \mathbb{E}\,\mathrm{tr}\left(K(X,X;\tilde{P})^{-1}(X^\top X + \sigma_\varepsilon^2 n_1 I_m)K(X,X;P)^{-1}K(X,\mathbf{x};P)K(\mathbf{x},X;\tilde{P})\right. \tag{S224}$$

$$\left.\times\left(\frac{\sigma_{W_2}^2\zeta}{n_0}X^\top + \frac{\zeta}{\sqrt{n_0 n_1}}F^\top W_1\right)\left(\frac{\sigma_{W_2}^2\zeta}{n_0}X^\top + \frac{\sqrt{\zeta}}{\sqrt{n_0 n_1}}f(\tilde{W}_1 X)^\top \tilde{W}_1)^\top\right)\right) \tag{S225}$$

$$\equiv H_{010} + E_6\,, \tag{S226}$$

where,

$$E_6 = \sigma_\varepsilon^2 n_1 \mathbb{E}\,\mathrm{tr}\left(K(X,X;\tilde{P})^{-1}K(X,X;P)^{-1}\right. \tag{S227}$$

$$\left.\times\left(\frac{\sigma_{W_2}^2\zeta}{n_0}X^\top + \frac{\zeta}{\sqrt{n_0 n_1}}F^\top W_1\right)\left(\frac{\sigma_{W_2}^2\zeta}{n_0}X^\top + \frac{\sqrt{\zeta}}{\sqrt{n_0 n_1}}f(\tilde{W}_1 X)^\top \tilde{W}_1)^\top\right)\right). \tag{S228}$$

A linear pencil for $E_6$ follows from the representation,

$$E_6 = \mathrm{tr}(U_6^T Q_6^{-1} V_6)\,, \tag{S229}$$

where,

$$U_6^T = \begin{pmatrix} \sigma_\varepsilon^2 I_m & 0 & 0 & 0 & 0 & 0 & 0 & 0 \end{pmatrix}, \quad V_6^T = \begin{pmatrix} 0 & 0 & 0 & 0 & 0 & \frac{I_m}{\gamma + \sigma_{W_2}^2(\eta'-\zeta)} & 0 & 0 \end{pmatrix} \tag{S230}$$

and,

$$Q_6 = \begin{pmatrix} I_m(\gamma + \sigma_{W_2}^2(\eta'-\zeta)) & \frac{\zeta X^\top \sigma_{W_2}^2}{n_0} & \frac{\sqrt{\eta-\zeta}\Theta_F^\top}{n_1} & \frac{\sqrt{\zeta}X^\top}{\sqrt{n_0 n_1}} & -\frac{\zeta^2 m X^\top \sigma_{W_2}^4}{n_0^2} & 0 & -\frac{\zeta^{3/2}m X^\top \sigma_{W_2}^2}{n_0^{3/2}n_1} & 0 \\ -X & I_{n_0} & 0 & 0 & 0 & 0 & 0 & 0 \\ -\sqrt{\eta-\zeta}\Theta_F & -\frac{\sqrt{\zeta}W_1}{\sqrt{n_0}} & I_{n_1} & 0 & \frac{\zeta^{3/2}m W_1 \sigma_{W_2}^2}{n_0^{3/2}} & 0 & \frac{\zeta m W_1}{n_0 n_1} & 0 \\ 0 & 0 & -W_1^\top & I_{n_0} & 0 & 0 & 0 & 0 \\ 0 & 0 & 0 & 0 & I_{n_0} & -X & 0 & 0 \\ 0 & 0 & 0 & 0 & \frac{\zeta X^\top \sigma_{W_2}^2}{n_0} & I_m(\gamma + \sigma_{W_2}^2(\eta'-\zeta)) & \frac{\sqrt{\zeta}X^\top}{\sqrt{n_0 n_1}} & \frac{\sqrt{\eta-\zeta}\tilde{\Theta}_F^\top}{n_1} \\ 0 & 0 & 0 & 0 & 0 & 0 & I_{n_0} & -\tilde{W}_1^\top \\ 0 & 0 & 0 & 0 & -\frac{\sqrt{\zeta}\tilde{W}_1}{\sqrt{n_0}} & -\sqrt{\eta-\zeta}\tilde{\Theta}_F & 0 & I_{n_1} \end{pmatrix}. \tag{S231}$$

The equations satisfied by the operator-valued Stieltjes transform $G$ of $\bar{Q}_6$ induce the following structure on $G$,

$$G = \begin{pmatrix} 0 & G_{12} \\ G_{12}^\top & 0 \end{pmatrix}, \tag{S232}$$

where,

$$G_{12} = \begin{pmatrix} g_5 & 0 & 0 & 0 & 0 & g_2 & 0 & 0 \\ 0 & g_6 & 0 & g_3 & g_1 & 0 & g_4 & 0 \\ 0 & 0 & g_8 & 0 & 0 & 0 & 0 & 0 \\ 0 & g_{11} & 0 & g_7 & g_{10} & 0 & g_9 & 0 \\ 0 & 0 & 0 & 0 & g_6 & 0 & g_3 & 0 \\ 0 & 0 & 0 & 0 & 0 & g_5 & 0 & 0 \\ 0 & 0 & 0 & 0 & g_{11} & 0 & g_7 & 0 \\ 0 & 0 & 0 & 0 & 0 & 0 & 0 & g_8 \end{pmatrix}, \tag{S233}$$

and the independent entry-wise component functions $g_i$ give the error $E_6$ through the relation,

$$E_6 = \frac{g_2 \sigma_\varepsilon^2}{\left(\gamma + \sigma_{W_2}^2(\eta'-\zeta)\right)}\,, \tag{S234}$$

and themselves satisfy the following system of polynomial equations,

$$0 = \sqrt{\zeta} g_6 g_8 \sqrt{n_0} - g_{11}\psi \tag{S235a}$$

$$0 = \sqrt{\zeta} g_3 g_8 \sqrt{n_0} - g_7\psi + \psi \tag{S235b}$$

$$0 = -\sqrt{\zeta} g_5 g_6 \psi - g_3\sqrt{n_0}\phi\big(\gamma + \sigma_{W_2}^2(\eta' - \zeta)\big) \tag{S235c}$$

$$0 = -\zeta g_7 g_8 \psi + \sqrt{\zeta} g_8\sqrt{n_0}\big(g_4\phi - \zeta g_3\sigma_{W_2}^2\big) - g_9\psi\phi \tag{S235d}$$

$$0 = -\sqrt{\zeta} g_5 g_{11}\psi - (g_7 - 1)\sqrt{n_0}\phi\big(\gamma + \sigma_{W_2}^2(\eta' - \zeta)\big) \tag{S235e}$$

$$0 = -\zeta g_8 g_{11}\psi + \sqrt{\zeta} g_8\sqrt{n_0}\big(g_1\phi - \zeta g_6\sigma_{W_2}^2\big) + g_{10}\psi(-\phi) \tag{S235f}$$

$$0 = g_3\sqrt{n_0}\phi\big(\gamma + \sigma_{W_2}^2(\eta' - \zeta)\big) + g_5\big(\sqrt{\zeta} g_7\psi + \zeta g_3\sqrt{n_0}\sigma_{W_2}^2\big) \tag{S235g}$$

$$0 = (g_6 - 1)\sqrt{n_0}\phi\big(\gamma + \sigma_{W_2}^2(\eta' - \zeta)\big) + g_5\big(\sqrt{\zeta} g_{11}\psi + \zeta g_6\sqrt{n_0}\sigma_{W_2}^2\big) \tag{S235h}$$

$$0 = \sqrt{\zeta} g_7 g_8\sqrt{n_0}\phi\big(\sigma_{W_2}^2(\zeta - \eta') - \gamma\big) + g_{11}\psi\big(\gamma\phi + \sigma_{W_2}^2(-\zeta\phi + \phi\eta' + \zeta g_5)\big) \tag{S235i}$$

$$0 = g_8\big(g_5\psi(\zeta - \eta) + \phi\big(\sqrt{\zeta} g_3\sqrt{n_0} - 1\big)\big(\gamma + \sigma_{W_2}^2(\eta' - \zeta)\big)\big) + \phi\big(\gamma + \sigma_{W_2}^2(\eta' - \zeta)\big) \tag{S235j}$$

$$0 = g_6\psi\big(\gamma\phi + \sigma_{W_2}^2(-\zeta\phi + \phi\eta' + \zeta g_5)\big) - \phi\big(\sqrt{\zeta} g_3 g_8\sqrt{n_0} + \psi\big)\big(\gamma + \sigma_{W_2}^2(\eta' - \zeta)\big) \tag{S235k}$$

$$0 = g_5\big(\sqrt{\zeta} g_{11}\psi + \sqrt{n_0}\big(\gamma + g_8(\eta - \zeta) + \sigma_{W_2}^2(\eta' + \zeta(g_6 - 1))\big)\big) - \sqrt{n_0}\big(\gamma + \sigma_{W_2}^2(\eta' - \zeta)\big) \tag{S235l}$$

$$0 = \sqrt{\zeta} g_2 g_{11}\psi\phi + \sqrt{n_0}\phi\big(\zeta g_7 g_8 + g_9\phi\big)\big(\gamma + \sigma_{W_2}^2(\eta' - \zeta)\big) + \sqrt{\zeta} g_5\psi\big(g_{10}\phi - \zeta g_{11}\sigma_{W_2}^2\big) \tag{S235m}$$

$$0 = g_2\big(\sqrt{\zeta} g_{11}\psi + \sqrt{n_0}\big(\gamma + g_8(\eta - \zeta) + \sigma_{W_2}^2(\eta' + \zeta(g_6 - 1))\big)\big)$$
$$\quad + g_5\big(\sqrt{\zeta} g_{10}\psi\phi - \zeta\sigma_{W_2}^2\big(\sqrt{\zeta} g_{11}\psi + \sqrt{n_0}(\zeta g_6\sigma_{W_2}^2 - g_1\phi)\big)\big) \tag{S235n}$$

$$0 = g_4\sqrt{n_0}\phi^2\big(\gamma + \sigma_{W_2}^2(\eta' - \zeta)\big) + g_2\big(\sqrt{\zeta} g_7\psi\phi + \zeta g_3\sqrt{n_0}\phi\sigma_{W_2}^2\big) + g_5\big(\sqrt{\zeta} g_9\psi\phi$$
$$\quad - \zeta\sigma_{W_2}^2\big(\sqrt{\zeta} g_7\psi + \sqrt{n_0}(\zeta g_3\sigma_{W_2}^2 - g_4\phi)\big)\big) \tag{S235o}$$

$$0 = -\sqrt{\zeta} g_8\sqrt{n_0}\phi\big(g_9\phi - \zeta g_7\sigma_{W_2}^2\big)\big(\gamma + \sigma_{W_2}^2(\eta' - \zeta)\big) + g_{10}\psi\phi\big(\gamma\phi + \sigma_{W_2}^2(-\zeta\phi + \phi\eta' + \zeta g_5)\big)$$
$$\quad + \zeta g_{11}\psi\sigma_{W_2}^2\big(g_2\phi - \zeta g_5\sigma_{W_2}^2\big) \tag{S235p}$$

$$0 = \phi\big(g_1\sqrt{n_0}\phi\big(\gamma + \sigma_{W_2}^2(\eta' - \zeta)\big) + g_2\big(\sqrt{\zeta} g_{11}\psi + \zeta g_6\sqrt{n_0}\sigma_{W_2}^2\big)\big) + g_5\big(\sqrt{\zeta} g_{10}\psi\phi$$
$$\quad - \zeta\sigma_{W_2}^2\big(\sqrt{\zeta} g_{11}\psi + \sqrt{n_0}(\zeta g_6\sigma_{W_2}^2 - g_1\phi)\big)\big) \tag{S235q}$$

$$0 = \sqrt{\zeta} g_1 g_5\psi\phi + \sqrt{\zeta} g_2 g_6\psi\phi + \gamma\zeta g_3 g_8\sqrt{n_0}\phi + \gamma g_4\sqrt{n_0}\phi^2 - \zeta^2 g_3 g_8\sqrt{n_0}\phi\sigma_{W_2}^2 + \sqrt{n_0}\phi\eta'\sigma_{W_2}^2\big(\zeta g_3 g_8 + g_4\phi\big)$$
$$\quad - \zeta g_4\sqrt{n_0}\phi^2\sigma_{W_2}^2 - \zeta^{3/2} g_5 g_6\psi\sigma_{W_2}^2 \tag{S235r}$$

$$0 = -\sqrt{\zeta} g_4 g_8\sqrt{n_0}\phi^2\big(\gamma + \sigma_{W_2}^2(\eta' - \zeta)\big) + \zeta\sigma_{W_2}^2\big(\sqrt{\zeta} g_3 g_8\sqrt{n_0}\phi\big(\gamma + \sigma_{W_2}^2(\eta' - \zeta)\big) - \zeta g_5 g_6\psi\sigma_{W_2}^2 + g_2 g_6\psi\phi\big)$$
$$\quad + g_1\psi\phi\big(\gamma\phi + \sigma_{W_2}^2\big(\phi(\eta' - \zeta) + \zeta g_5\big)\big) \tag{S235s}$$

$$\tag{S235t}$$

After some straightforward algebra, one can eliminate all $g_i$ except for $g_3$ and $g_5$, which satisfy coupled polynomial equations. Those equations can be shown to be identical to eqn. (S66) by invoking the change of variables,

$$g_3 = -\frac{\sqrt{\zeta}\psi}{\sqrt{n_0}\phi}\tau_2, \quad \text{and} \quad g_5 = \big(\gamma + \sigma_{W_2}^2(\eta' - \zeta)\big)\tau_1. \tag{S236}$$

In terms of the related variables defined in eqn. (S89), the error $E_6$ is given by,

$$E_6 = \sigma_\varepsilon^2\phi\tilde{\tau}_2^2/(1 - \tilde{\tau}_2^2\phi) \tag{S237}$$

## S6.5  $H_{100}$

$$H_{100} = \mathbb{E}\hat{y}(\mathbf{x}; P, X, \varepsilon)\hat{y}(\mathbf{x}; P, \tilde{X}, \tilde{\varepsilon}) \tag{S238}$$

$$= \mathbb{E}\Big[ N_0(\mathbf{x};P)N_0(\mathbf{x};P)^\top + K(\mathbf{x},\tilde{X};P)K(\tilde{X},\tilde{X};P)^{-1}Y(\tilde{X},\tilde{\varepsilon})^\top Y(X,\varepsilon)K(X,X;P)^{-1}K(X,\mathbf{x};P)$$

$$+ K(\mathbf{x},\tilde{X};\mathbb{P})K(\tilde{X},\tilde{X};P)^{-1}N_0(\tilde{X})^\top N_0(X)K(X,X;P)^{-1}K(X,\mathbf{x};P)$$

$$- N_0(\mathbf{x};P)N_0(X;P)K(X,X;P)^{-1}K(X,\mathbf{x};P) - N_0(\mathbf{x};P)N_0(\tilde{X};P)K(\tilde{X},\tilde{X};P)^{-1}K(\tilde{X},\mathbf{x};P)\Big] \quad \text{(S239)}$$

$$= \nu\sigma_{W_2}^2\eta + \nu E_{22} + \mathbb{E}\operatorname{tr}\Big(K(\tilde{X},\tilde{X};P)^{-1}(\tilde{X}^\top X + \nu\frac{\sigma_{W_2}^2}{n_1}f(W_1\tilde{X})^T F)K(X,X;P)^{-1}K(X,\mathbf{x};P)K(\mathbf{x},\tilde{X};P)\Big) \quad \text{(S240)}$$

$$= \nu\sigma_{W_2}^2\eta + \nu E_{22} + \mathbb{E}\operatorname{tr}\Big(K(\tilde{X},\tilde{X};P)^{-1}(\tilde{X}^\top X + \nu\frac{\sigma_{W_2}^2}{n_1}f(W_1\tilde{X})^T F)K(X,X;P)^{-1} \quad \text{(S241)}$$

$$\times (\frac{\sigma_{W_2}^2\zeta}{n_0}X^\top + \frac{\zeta}{\sqrt{n_0 n_1}}F^\top W_1)(\frac{\sigma_{W_2}^2\zeta}{n_0}\tilde{X}^\top + \frac{\sqrt{\zeta}}{\sqrt{n_0 n_1}}f(W_1\tilde{X})^\top W_1)^\top\Big) \quad \text{(S242)}$$

$$\equiv \nu\sigma_{W_2}^2\eta + \nu E_{22} + E_{71} + \nu E_{72}, \quad \text{(S243)}$$

where, $E_{22}$ is given above and,

$$E_{71} = \mathbb{E}\operatorname{tr}\Big(K(\tilde{X},\tilde{X};P)^{-1}\tilde{X}^\top X K(X,X;P)^{-1}\big(\frac{\eta-\zeta}{n_1^2}f(W_1 X)^\top f(W_1\tilde{X}) \quad \text{(S244)}$$

$$+ (\frac{\sigma_{W_2}^2\zeta}{n_0}X^\top + \frac{\zeta}{\sqrt{n_0 n_1}}F^\top W_1)(\frac{\sigma_{W_2}^2\zeta}{n_0}\tilde{X}^\top + \frac{\sqrt{\zeta}}{\sqrt{n_0 n_1}}f(W_1\tilde{X})^\top W_1)^\top\big)\Big) \quad \text{(S245)}$$

$$E_{72} = \frac{\sigma_{W_2}^2}{n_1}\mathbb{E}\operatorname{tr}\Big(K(\tilde{X},\tilde{X};P)^{-1}f(W_1\tilde{X})^T F K(X,X;P)^{-1}\big(\frac{\eta-\zeta}{n_1^2}f(W_1 X)^\top f(W_1\tilde{X}) \quad \text{(S246)}$$

$$+ (\frac{\sigma_{W_2}^2\zeta}{n_0}X^\top + \frac{\zeta}{\sqrt{n_0 n_1}}F^\top W_1)(\frac{\sigma_{W_2}^2\zeta}{n_0}\tilde{X}^\top + \frac{\sqrt{\zeta}}{\sqrt{n_0 n_1}}f(W_1\tilde{X})^\top W_1)^\top\big)\Big). \quad \text{(S247)}$$

### S6.5.1  $E_{71}$

A linear pencil for $E_{71}$ follows from the representation,

$$E_{71} = \operatorname{tr}(U_{71}^T Q_{71}^{-1} V_{71}), \quad \text{(S248)}$$

where,

$$U_{71}^T = \begin{pmatrix} 0 & \frac{I_{n_0}}{m} & 0 & 0 & 0 & 0 & 0 & 0 & 0 \end{pmatrix} \quad \text{(S249)}$$

$$V_{71}^T = \begin{pmatrix} 0 & 0 & 0 & 0 & 0 & 0 & 0 & 0 & -\frac{\sqrt{n_0 n_1}I_{n_0}}{\sqrt{\zeta}} \end{pmatrix} \quad \text{(S250)}$$

and, for $\beta = \big(n_0(\zeta-\eta) - \zeta n_1\sigma_{W_2}^2\big)$,

$$Q_{71} = \begin{pmatrix}
I_m\big(\gamma+\sigma_{W_2}^2(\eta'-\zeta)\big) & 0 & \frac{\zeta X^\top\sigma_{W_2}^2}{n_0} & \frac{\sqrt{\eta-\zeta}\Theta_F^\top}{n_1} & \frac{\sqrt{\zeta}X^\top}{\sqrt{n_0}n_1} & -\frac{\zeta^2 m X^\top\sigma_{W_2}^4}{n_0^2} & 0 & 0 & 0 \\
-X & I_{n_0} & 0 & 0 & 0 & 0 & 0 & 0 & 0 \\
-X & 0 & I_{n_0} & 0 & 0 & 0 & 0 & \frac{\sqrt{\zeta}m W_1^\top}{\sqrt{n_0}n_1} & 0 \\
-\sqrt{\eta-\zeta}\Theta_F & 0 & -\frac{\sqrt{\zeta}W_1}{\sqrt{n_0}} & I_{n_1} & 0 & -\frac{\sqrt{\zeta}m W_1\beta}{n_0^{3/2}n_1} & \frac{m(\eta-\zeta)^{3/2}\tilde{\Theta}_F}{n_1} & 0 & 0 \\
0 & 0 & 0 & -W_1^\top & I_{n_0} & 0 & 0 & 0 & 0 \\
0 & 0 & 0 & 0 & 0 & I_{n_0} & -\tilde{X} & 0 & 0 \\
0 & 0 & 0 & 0 & 0 & \frac{\zeta\sigma_{W_2}^2\tilde{X}^\top}{n_0} & I_m\big(\gamma+\sigma_{W_2}^2(\eta'-\zeta)\big) & \frac{\sqrt{\eta-\zeta}\tilde{\Theta}_F^\top}{n_1} & \frac{\sqrt{\zeta}\tilde{X}^\top}{\sqrt{n_0}n_1} \\
0 & 0 & 0 & 0 & 0 & -\frac{\sqrt{\zeta}W_1}{\sqrt{n_0}} & -\sqrt{\eta-\zeta}\tilde{\Theta}_F & I_{n_1} & 0 \\
0 & 0 & 0 & 0 & 0 & 0 & 0 & -W_1^\top & I_{n_0}
\end{pmatrix}. \quad \text{(S251)}$$

The equations satisfied by the operator-valued Stieltjes transform $G$ of $\bar{Q}_{71}$ induce the following structure on $G$,

$$G = \begin{pmatrix} 0 & G_{12} \\ G_{12}^\top & 0 \end{pmatrix}, \tag{S252}$$

where,

$$G_{12} = \begin{pmatrix} g_8 & 0 & 0 & 0 & 0 & 0 & 0 & 0 & 0 \\ 0 & g_1 & g_3 & 0 & g_5 & g_6 & 0 & 0 & g_2 \\ 0 & 0 & g_9 & 0 & g_5 & g_{12} & 0 & 0 & g_4 \\ 0 & 0 & 0 & g_{11} & 0 & 0 & 0 & g_{15} & 0 \\ 0 & 0 & g_{14} & 0 & g_{10} & g_{13} & 0 & 0 & g_7 \\ 0 & 0 & 0 & 0 & 0 & g_9 & 0 & 0 & g_5 \\ 0 & 0 & 0 & 0 & 0 & 0 & g_8 & 0 & 0 \\ 0 & 0 & 0 & 0 & 0 & 0 & 0 & g_{11} & 0 \\ 0 & 0 & 0 & 0 & 0 & g_{14} & 0 & 0 & g_{10} \end{pmatrix}, \tag{S253}$$

and the independent entry-wise component functions $g_i$ give the error $E_{71}$ through the relation,

$$E_{71} = -\frac{g_2\sqrt{n_0}\phi}{\sqrt{\zeta}\psi}, \tag{S254}$$

and themselves satisfy the following system of polynomial equations,

$$0 = 1 - g_1 \tag{S255a}$$

$$0 = \sqrt{\zeta}g_9 g_{11}\sqrt{n_0} - g_{14}\psi \tag{S255b}$$

$$0 = \sqrt{\zeta}g_5 g_{11}\sqrt{n_0} - g_{10}\psi + \psi \tag{S255c}$$

$$0 = -\sqrt{\zeta}g_6 g_8 \psi - g_2\sqrt{n_0}\phi\left(\gamma + \sigma_{W_2}^2\left(\eta' - \zeta\right)\right) \tag{S255d}$$

$$0 = -\sqrt{\zeta}g_8 g_9 \psi - g_5\sqrt{n_0}\phi\left(\gamma + \sigma_{W_2}^2\left(\eta' - \zeta\right)\right) \tag{S255e}$$

$$0 = -\sqrt{\zeta}g_8 g_{12}\psi - g_4\sqrt{n_0}\phi\left(\gamma + \sigma_{W_2}^2\left(\eta' - \zeta\right)\right) \tag{S255f}$$

$$0 = -\sqrt{\zeta}g_8 g_{13}\psi - g_7\sqrt{n_0}\phi\left(\gamma + \sigma_{W_2}^2\left(\eta' - \zeta\right)\right) \tag{S255g}$$

$$0 = -\sqrt{\zeta}g_8 g_{14}\psi - (g_{10} - 1)\sqrt{n_0}\phi\left(\gamma + \sigma_{W_2}^2\left(\eta' - \zeta\right)\right) \tag{S255h}$$

$$0 = -\sqrt{\zeta}g_1 g_8 \psi - \sqrt{\zeta}g_3 g_8 \psi - g_5\sqrt{n_0}\phi\left(\gamma + \sigma_{W_2}^2\left(\eta' - \zeta\right)\right) \tag{S255i}$$

$$0 = g_3\sqrt{n_0}\phi\left(\gamma + \sigma_{W_2}^2\left(\eta' - \zeta\right)\right) + g_8\left(\sqrt{\zeta}g_{14}\psi + \zeta g_9\sqrt{n_0}\sigma_{W_2}^2\right) \tag{S255j}$$

$$0 = g_5\sqrt{n_0}\phi\left(\gamma + \sigma_{W_2}^2\left(\eta' - \zeta\right)\right) + g_8\left(\sqrt{\zeta}g_{10}\psi + \zeta g_5\sqrt{n_0}\sigma_{W_2}^2\right) \tag{S255k}$$

$$0 = \sqrt{\zeta}\sqrt{n_0}\left(g_5\left(g_{11}\left(\zeta\psi - \eta\psi - \zeta\sigma_{W_2}^2\right) + g_{15}\phi\right) + g_4 g_{11}\phi\right) - g_7\psi\phi \tag{S255l}$$

$$0 = \sqrt{\zeta}\sqrt{n_0}\left(g_9\left(g_{11}\left(\zeta\psi - \eta\psi - \zeta\sigma_{W_2}^2\right) + g_{15}\phi\right) + g_{11}g_{12}\phi\right) - g_{13}\psi\phi \tag{S255m}$$

$$0 = (g_9 - 1)\sqrt{n_0}\phi\left(\gamma + \sigma_{W_2}^2\left(\eta' - \zeta\right)\right) + g_8\left(\sqrt{\zeta}g_{14}\psi + \zeta g_9\sqrt{n_0}\sigma_{W_2}^2\right) \tag{S255n}$$

$$0 = \sqrt{\zeta}g_7 g_8 \psi\phi + \sqrt{n_0}\left(g_2\phi^2\left(\gamma + \sigma_{W_2}^2\left(\eta' - \zeta\right)\right) + \zeta g_8 \sigma_{W_2}^2\left(g_4\phi - \zeta g_5 \sigma_{W_2}^2\right)\right) \tag{S255o}$$

$$0 = \sqrt{\zeta}g_{10}g_{11}\sqrt{n_0}\phi\left(\sigma_{W_2}^2\left(\zeta - \eta'\right) - \gamma\right) + g_{14}\psi\left(\gamma\phi + \sigma_{W_2}^2\left(-\zeta\phi + \phi\eta' + \zeta g_8\right)\right) \tag{S255p}$$

$$0 = g_{11}\left(g_8\psi(\zeta - \eta) + \phi\left(\sqrt{\zeta}g_5\sqrt{n_0} - 1\right)\left(\gamma + \sigma_{W_2}^2\left(\eta' - \zeta\right)\right)\right) + \phi\left(\gamma + \sigma_{W_2}^2\left(\eta' - \zeta\right)\right) \tag{S255q}$$

$$0 = g_6\sqrt{n_0}\phi^2\left(\gamma + \sigma_{W_2}^2\left(\eta' - \zeta\right)\right) + g_8\left(\sqrt{\zeta}g_{13}\psi\phi + \zeta\sqrt{n_0}\sigma_{W_2}^2\left(g_{12}\phi - \zeta g_9 \sigma_{W_2}^2\right)\right) \tag{S255r}$$

$$0 = g_9\psi\left(\gamma\phi + \sigma_{W_2}^2\left(-\zeta\phi + \phi\eta' + \zeta g_8\right)\right) - \phi\left(\sqrt{\zeta}g_5 g_{11}\sqrt{n_0} + \psi\right)\left(\gamma + \sigma_{W_2}^2\left(\eta' - \zeta\right)\right) \tag{S255s}$$

$$0 = g_8\left(\sqrt{\zeta}g_{14}\psi + \sqrt{n_0}\left(\gamma + g_{11}(\eta - \zeta) + \sigma_{W_2}^2\left(\eta' + \zeta(g_9 - 1)\right)\right)\right) - \sqrt{n_0}\left(\gamma + \sigma_{W_2}^2\left(\eta' - \zeta\right)\right) \tag{S255t}$$

$$0 = -\sqrt{\zeta}g_5 g_{11}\sqrt{n_0}\phi\left(\gamma + \sigma_{W_2}^2\left(\eta' - \zeta\right)\right) + g_3\psi\left(\gamma\phi + \sigma_{W_2}^2\left(-\zeta\phi + \phi\eta' + \zeta g_8\right)\right) + \zeta g_1 g_8 \psi\phi\sigma_{W_2}^2 \tag{S255u}$$

$$0 = \sqrt{n_0}\phi\left(\zeta g_9 g_{11} + g_{12}\phi\right)\left(\gamma + \sigma_{W_2}^2\left(\eta' - \zeta\right)\right) + g_8\left(\sqrt{\zeta}g_{13}\psi\phi + \zeta\sqrt{n_0}\sigma_{W_2}^2\left(g_{12}\phi - \zeta g_9\sigma_{W_2}^2\right)\right) \tag{S255v}$$

$$0 = \sqrt{\zeta}g_7 g_8\psi\phi + \sqrt{n_0}\left(g_4\phi\left(\gamma\phi + \sigma_{W_2}^2\left(-\zeta\phi + \phi\eta' + \zeta g_8\right)\right) + \zeta g_5\left(g_{11}\phi\left(\gamma + \sigma_{W_2}^2\left(\eta' - \zeta\right)\right) - \zeta g_8\sigma_{W_2}^4\right)\right) \tag{S255w}$$

$$0 = g_8\psi\left(-(\zeta - \eta)\right)\left(g_{11}\psi(\zeta - \eta) + g_{15}\phi\right) - \phi\left(\gamma + \sigma_{W_2}^2\left(\eta' - \zeta\right)\right)$$
$$\left(-\zeta g_9 g_{11}\psi + \sqrt{\zeta}\sqrt{n_0}\left(g_5\left(g_{11}\left(\zeta\psi - \eta\psi - \zeta\sigma_{W_2}^2\right) + g_{15}\phi\right) + g_4 g_{11}\phi\right) - g_{15}\phi\right) \tag{S255x}$$

$$0 = \phi\left(g_{12}\psi\left(\gamma\phi + \sigma_{W_2}^2\left(-\zeta\phi + \phi\eta' + \zeta g_8\right)\right) - \sqrt{\zeta}\sqrt{n_0}\left(\gamma + \sigma_{W_2}^2\left(\eta' - \zeta\right)\right)\right.$$
$$\left.\left(g_5\left(g_{11}\left(\zeta\psi - \eta\psi - \zeta\sigma_{W_2}^2\right) + g_{15}\phi\right) + g_4 g_{11}\phi\right)\right) + \zeta g_9\psi\left(g_{11}\phi\left(\gamma + \sigma_{W_2}^2\left(\eta' - \zeta\right)\right) - \zeta g_8\sigma_{W_2}^4\right) \tag{S255y}$$

$$0 = \sqrt{\zeta}\left(-g_{10}g_{15}\sqrt{n_0}\phi^2\left(\gamma + \sigma_{W_2}^2\left(\eta' - \zeta\right)\right) - g_{11}\phi\left(\gamma + \sigma_{W_2}^2\left(\eta' - \zeta\right)\right)\left(\sqrt{n_0}\left(g_{10}\left(\zeta\psi - \eta\psi - \zeta\sigma_{W_2}^2\right) + g_7\phi\right)\right.\right.$$
$$\left.\left. - \sqrt{\zeta}g_{14}\psi\right) - \zeta^{3/2}g_8 g_{14}\psi\sigma_{W_2}^4\right) + g_{13}\psi\phi\left(\gamma\phi + \sigma_{W_2}^2\left(\phi\left(\eta' - \zeta\right) + \zeta g_8\right)\right) \tag{S255z}$$

$$0 = \sqrt{\zeta}\left(-\gamma\zeta g_5 g_{11}\sqrt{n_0}\psi\phi + \gamma\eta g_5 g_{11}\sqrt{n_0}\psi\phi - \gamma g_5 g_{15}\sqrt{n_0}\phi^2 - g_2 g_{11}\sqrt{n_0}\phi^2\left(\gamma + \sigma_{W_2}^2\left(\eta' - \zeta\right)\right)\right.$$
$$+ \gamma\zeta g_5 g_{11}\sqrt{n_0}\phi\sigma_{W_2}^2 + \zeta^2 g_5 g_{11}\sqrt{n_0}\psi\phi\sigma_{W_2}^2 - \zeta^2 g_5 g_{11}\sqrt{n_0}\phi\sigma_{W_2}^4$$
$$+ g_5\sqrt{n_0}\phi\eta'\sigma_{W_2}^2\left(g_{11}\left(-\zeta\psi + \eta\psi + \zeta\sigma_{W_2}^2\right) - g_{15}\phi\right) - \zeta\eta g_5 g_{11}\sqrt{n_0}\psi\phi\sigma_{W_2}^2 + \zeta g_5 g_{15}\sqrt{n_0}\phi^2\sigma_{W_2}^2$$
$$+ \sqrt{\zeta}g_3\psi\left(g_{11}\phi\left(\gamma + \sigma_{W_2}^2\left(\eta' - \zeta\right)\right) - \zeta g_8\sigma_{W_2}^4\right) - \zeta^{3/2}g_1 g_8\psi\sigma_{W_2}^4\right)$$
$$+ g_6\psi\phi\left(\gamma\phi + \sigma_{W_2}^2\left(\phi\left(\eta' - \zeta\right) + \zeta g_8\right)\right) \tag{S255aa}$$

After some straightforward algebra, one can eliminate all $g_i$ except for $g_5$ and $g_8$, which satisfy coupled polynomial equations. Those equations can be shown to be identical to eqn. (S66) by invoking the change of variables,

$$g_5 = -\frac{\sqrt{\zeta}\psi}{\sqrt{n_0}\phi}\tau_2, \quad \text{and} \quad g_8 = \left(\gamma + \sigma_{W_2}^2\left(\eta' - \zeta\right)\right)\tau_1. \tag{S256}$$

In terms of the related variables defined in eqn. (S89), the error $E_{71}$ is given by,

$$E_{71} = \frac{\psi\tilde{\tau}_1^2\left(2\zeta\tilde{\tau}_2 + \eta\right) + \zeta\phi^2\tilde{\tau}_2^2}{\zeta\left(\phi^2 - \psi\tilde{\tau}_1^2\right)} \tag{S257}$$

$$= \tau_2'/\tau_1' - 2\tau_2/\tau_1 + 1. \tag{S258}$$

### S6.5.2 $E_{72}$

A linear pencil for $E_{72}$ follows from the representation,

$$E_{72} = \text{tr}(U_{72}^T Q_{72}^{-1} V_{72}), \tag{S259}$$

where,

$$U_{72}^T = \left(\begin{array}{ccccccccccc} 0 & \frac{I_{n_1}}{m} & 0 & 0 & 0 & 0 & 0 & 0 & 0 & 0 & 0 \end{array}\right) \tag{S260}$$

$$V_{72}^T = \left(\begin{array}{ccccccccccc} 0 & 0 & 0 & 0 & 0 & 0 & 0 & 0 & 0 & -n_1 I_{n_1} & 0 \end{array}\right) \tag{S261}$$

and, for $\beta = \big(n_0(\zeta - \eta) - \zeta n_1 \sigma_{W_2}^2\big)$,

$$Q_{72} = \begin{pmatrix}
I_m\big(\gamma + \sigma_{W_2}^2(\eta' - \zeta)\big) & 0 & 0 & \frac{\zeta X^\top \sigma_{W_2}^2}{n_0} & \frac{\sqrt{\eta-\zeta}\Theta_F^\top}{n_1} & \frac{\sqrt{\zeta}X^\top}{\sqrt{n_0}n_1} & -\frac{\zeta^2 m X^\top \sigma_{W_2}^4}{n_0^2} & 0 & 0 & 0 & 0 \\
-\sqrt{\eta-\zeta}\Theta_F & I_{n_1} & -\frac{\sqrt{\zeta}W_1}{\sqrt{n_0}} & 0 & 0 & 0 & 0 & 0 & 0 & 0 & 0 \\
-X & 0 & I_{n_0} & 0 & 0 & 0 & 0 & 0 & 0 & 0 & 0 \\
-X & 0 & 0 & I_{n_0} & 0 & 0 & 0 & 0 & \frac{\sqrt{\zeta}m W_1^\top}{\sqrt{n_0}n_1} & 0 & 0 \\
-\sqrt{\eta-\zeta}\Theta_F & 0 & 0 & -\frac{\sqrt{\zeta}W_1}{\sqrt{n_0}} & I_{n_1} & 0 & -\frac{\sqrt{\zeta}m W_1 \beta}{n_0^{3/2}n_1} & \frac{m(\eta-\zeta)^{3/2}\tilde\Theta_F}{n_1} & 0 & 0 & 0 \\
0 & 0 & 0 & 0 & -W_1^\top & I_{n_0} & 0 & 0 & 0 & 0 & 0 \\
0 & 0 & 0 & 0 & 0 & 0 & I_{n_0} & -\tilde X & 0 & 0 & 0 \\
0 & 0 & 0 & 0 & 0 & 0 & \frac{\zeta \sigma_{W_2}^2 \tilde X^\top}{n_0} & I_m\big(\gamma + \sigma_{W_2}^2(\eta' - \zeta)\big) & 0 & \frac{\sqrt{\eta-\zeta}\tilde\Theta_F^\top}{n_1} & \frac{\sqrt{\zeta}\tilde X^\top}{\sqrt{n_0}n_1} \\
0 & 0 & 0 & 0 & 0 & 0 & -\frac{\sqrt{\zeta}W_1}{\sqrt{n_0}} & -\sqrt{\eta-\zeta}\tilde\Theta_F & I_{n_1} & 0 & 0 \\
0 & 0 & 0 & 0 & 0 & 0 & -\frac{\sqrt{\zeta}W_1}{\sqrt{n_0}} & -\sqrt{\eta-\zeta}\tilde\Theta_F & 0 & I_{n_1} & 0 \\
0 & 0 & 0 & 0 & 0 & 0 & 0 & 0 & 0 & -W_1^\top & I_{n_0}
\end{pmatrix}.$$
(S262)

The equations satisfied by the operator-valued Stieltjes transform $G$ of $\bar Q_{72}$ induce the following structure on $G$,

$$G = \begin{pmatrix} 0 & G_{12} \\ G_{12}^\top & 0 \end{pmatrix},$$
(S263)

where,

$$G_{12} = \begin{pmatrix}
g_{12} & 0 & 0 & 0 & 0 & 0 & 0 & 0 & 0 & 0 & 0 \\
0 & g_1 & 0 & 0 & g_6 & 0 & 0 & 0 & g_{11} & g_3 & 0 \\
0 & 0 & g_1 & g_4 & 0 & g_7 & g_8 & 0 & 0 & 0 & g_2 \\
0 & 0 & 0 & g_{13} & 0 & g_7 & g_{16} & 0 & 0 & 0 & g_5 \\
0 & 0 & 0 & 0 & g_{15} & 0 & 0 & 0 & g_{19} & g_{10} & 0 \\
0 & 0 & 0 & g_{18} & 0 & g_{14} & g_{17} & 0 & 0 & 0 & g_9 \\
0 & 0 & 0 & 0 & 0 & 0 & g_{13} & 0 & 0 & 0 & g_7 \\
0 & 0 & 0 & 0 & 0 & 0 & 0 & g_{12} & 0 & 0 & 0 \\
0 & 0 & 0 & 0 & 0 & 0 & 0 & 0 & g_1 & g_6 & 0 \\
0 & 0 & 0 & 0 & 0 & 0 & 0 & 0 & 0 & g_{15} & 0 \\
0 & 0 & 0 & 0 & 0 & 0 & g_{18} & 0 & 0 & 0 & g_{14}
\end{pmatrix},$$
(S264)

and the independent entry-wise component functions $g_i$ give the error $E_{72}$ through the relation,

$$E_{72} = -\frac{\phi g_3}{\psi},$$
(S265)

and themselves satisfy the following system of polynomial equations,

$$0 = 1 - g_1 \tag{S266a}$$
$$0 = -\zeta g_{13}g_{15}\psi - g_{19}\phi \tag{S266b}$$
$$0 = \sqrt{\zeta}g_{13}g_{15}\sqrt{n_0} - g_{18}\psi \tag{S266c}$$
$$0 = \sqrt{\zeta}g_7 g_{15}\sqrt{n_0} - g_{14}\psi + \psi \tag{S266d}$$
$$0 = g_{11}(-\phi) - \zeta\big(g_1 g_4 + g_6 g_{13}\big)\psi \tag{S266e}$$
$$0 = -\sqrt{\zeta}g_8 g_{12}\psi - g_2\sqrt{n_0}\phi\big(\gamma + \sigma_{W_2}^2(\eta' - \zeta)\big) \tag{S266f}$$
$$0 = -\sqrt{\zeta}g_{12}g_{13}\psi - g_7\sqrt{n_0}\phi\big(\gamma + \sigma_{W_2}^2(\eta' - \zeta)\big) \tag{S266g}$$
$$0 = -\sqrt{\zeta}g_{12}g_{16}\psi - g_5\sqrt{n_0}\phi\big(\gamma + \sigma_{W_2}^2(\eta' - \zeta)\big) \tag{S266h}$$
$$0 = -\sqrt{\zeta}g_{12}g_{17}\psi - g_9\sqrt{n_0}\phi\big(\gamma + \sigma_{W_2}^2(\eta' - \zeta)\big) \tag{S266i}$$

$$0 = -\sqrt{\zeta}g_{12}g_{18}\psi - (g_{14}-1)\sqrt{n_0}\phi\big(\gamma + \sigma_{W_2}^2(\eta'-\zeta)\big) \tag{S266j}$$

$$0 = -\sqrt{\zeta}g_1g_{12}\psi - \sqrt{\zeta}g_4g_{12}\psi - g_7\sqrt{n_0}\phi\big(\gamma + \sigma_{W_2}^2(\eta'-\zeta)\big) \tag{S266k}$$

$$0 = g_{12}g_{15}\psi(\zeta-\eta) - \phi\big(g_6 - \sqrt{\zeta}g_7g_{15}\sqrt{n_0}\big)\big(\gamma + \sigma_{W_2}^2(\eta'-\zeta)\big) \tag{S266l}$$

$$0 = g_7\sqrt{n_0}\phi\big(\gamma + \sigma_{W_2}^2(\eta'-\zeta)\big) + g_{12}\big(\sqrt{\zeta}g_{14}\psi + \zeta g_7\sqrt{n_0}\sigma_{W_2}^2\big) \tag{S266m}$$

$$0 = g_4\sqrt{n_0}\phi\big(\gamma + \sigma_{W_2}^2(\eta'-\zeta)\big) + g_{12}\big(\sqrt{\zeta}g_{18}\psi + \zeta g_{13}\sqrt{n_0}\sigma_{W_2}^2\big) \tag{S266n}$$

$$0 = \sqrt{\zeta}\sqrt{n_0}\big(g_7\big(g_{15}(\zeta\psi - \eta\psi - \zeta\sigma_{W_2}^2) + (g_{10}+g_{19})\phi\big) + g_5g_{15}\phi\big) - g_9\psi\phi \tag{S266o}$$

$$0 = (g_{13}-1)\sqrt{n_0}\phi\big(\gamma + \sigma_{W_2}^2(\eta'-\zeta)\big) + g_{12}\big(\sqrt{\zeta}g_{18}\psi + \zeta g_{13}\sqrt{n_0}\sigma_{W_2}^2\big) \tag{S266p}$$

$$0 = \sqrt{\zeta}\sqrt{n_0}\big(g_{13}\big(g_{15}(\zeta\psi - \eta\psi - \zeta\sigma_{W_2}^2) + g_{19}\phi\big) + g_{10}g_{13}\phi + g_{15}g_{16}\phi\big) - g_{17}\psi\phi \tag{S266q}$$

$$0 = \sqrt{\zeta}g_9g_{12}\psi\phi + \sqrt{n_0}\big(g_2\phi^2\big(\gamma + \sigma_{W_2}^2(\eta'-\zeta)\big) + \zeta g_{12}\sigma_{W_2}^2\big(g_5\phi - \zeta g_7\sigma_{W_2}^2\big)\big) \tag{S266r}$$

$$0 = \sqrt{\zeta}g_{14}g_{15}\sqrt{n_0}\phi\big(\sigma_{W_2}^2(\zeta-\eta') - \gamma\big) + g_{18}\psi\big(\gamma\phi + \sigma_{W_2}^2\big(-\zeta\phi + \phi\eta' + \zeta g_{12}\big)\big) \tag{S266s}$$

$$0 = g_{15}\big(g_{12}\psi(\zeta-\eta) + \phi\big(\sqrt{\zeta}g_7\sqrt{n_0} - 1\big)\big(\gamma + \sigma_{W_2}^2(\eta'-\zeta)\big)\big) + \phi\big(\gamma + \sigma_{W_2}^2(\eta'-\zeta)\big) \tag{S266t}$$

$$0 = g_8\sqrt{n_0}\phi^2\big(\gamma + \sigma_{W_2}^2(\eta'-\zeta)\big) + g_{12}\big(\sqrt{\zeta}g_{17}\psi\phi + \zeta\sqrt{n_0}\sigma_{W_2}^2\big(g_{16}\phi - \zeta g_{13}\sigma_{W_2}^2\big)\big) \tag{S266u}$$

$$0 = g_{13}\psi\big(\gamma\phi + \sigma_{W_2}^2\big(-\zeta\phi + \phi\eta' + \zeta g_{12}\big)\big) - \phi\big(\sqrt{\zeta}g_7g_{15}\sqrt{n_0} + \psi\big)\big(\gamma + \sigma_{W_2}^2(\eta'-\zeta)\big) \tag{S266v}$$

$$0 = g_{12}\big(\sqrt{\zeta}g_{18}\psi + \sqrt{n_0}\big(\gamma + g_{15}(\eta-\zeta) + \sigma_{W_2}^2\big(\eta' + \zeta(g_{13}-1)\big)\big)\big) - \sqrt{n_0}\big(\gamma + \sigma_{W_2}^2(\eta'-\zeta)\big) \tag{S266w}$$

$$0 = g_{12}g_{19}\psi(\zeta-\eta) - \phi\big(g_{11} - \sqrt{\zeta}g_7g_{19}\sqrt{n_0}\big)\big(\gamma + \sigma_{W_2}^2(\eta'-\zeta)\big) + \zeta g_1g_4\psi\big(\sigma_{W_2}^2(\zeta-\eta') - \gamma\big) \tag{S266x}$$

$$0 = g_{19}\big(g_{12}\psi(\zeta-\eta) + \phi\big(\sqrt{\zeta}g_7\sqrt{n_0} - 1\big)\big(\gamma + \sigma_{W_2}^2(\eta'-\zeta)\big)\big) - \zeta g_1g_{13}\psi\big(\gamma + \sigma_{W_2}^2(\eta'-\zeta)\big) \tag{S266y}$$

$$0 = -\sqrt{\zeta}g_7g_{15}\sqrt{n_0}\phi\big(\gamma + \sigma_{W_2}^2(\eta'-\zeta)\big) + g_4\psi\big(\gamma\phi + \sigma_{W_2}^2\big(-\zeta\phi + \phi\eta' + \zeta g_{12}\big)\big) + \zeta g_1g_{12}\psi\sigma_{W_2}^2 \tag{S266z}$$

$$0 = \sqrt{n_0}\phi\big(\zeta(g_1+g_6)g_{13} + g_{16}\phi\big)\big(\gamma + \sigma_{W_2}^2(\eta'-\zeta)\big) + g_{12}\big(\sqrt{\zeta}g_{17}\psi\phi + \zeta\sqrt{n_0}\sigma_{W_2}^2\big(g_{16}\phi - \zeta g_{13}\sigma_{W_2}^2\big)\big) \tag{S266aa}$$

$$0 = g_6\big(g_{12}\psi(\zeta-\eta) + \phi\big(\sqrt{\zeta}g_7\sqrt{n_0} - 1\big)\big(\gamma + \sigma_{W_2}^2(\eta'-\zeta)\big)\big) + g_1\big(g_{12}\psi(\zeta-\eta)$$
$$+ \sqrt{\zeta}g_7\sqrt{n_0}\phi\big(\gamma + \sigma_{W_2}^2(\eta'-\zeta)\big)\big) \tag{S266ab}$$

$$0 = \sqrt{\zeta}g_9g_{12}\psi\phi + \sqrt{n_0}\big(g_5\phi\big(\gamma\phi + \sigma_{W_2}^2\big(\phi(\eta'-\zeta) + \zeta g_{12}\big)\big) + \zeta g_7\big(g_1\phi\big(\gamma + \sigma_{W_2}^2(\eta'-\zeta)\big)$$
$$+ g_6\phi\big(\gamma + \sigma_{W_2}^2(\eta'-\zeta)\big) - \zeta g_{12}\sigma_{W_2}^4\big)\big) \tag{S266ac}$$

$$0 = g_{12}\psi(-(\zeta-\eta))\big(g_{15}\psi(\zeta-\eta) + g_{19}\phi\big) - \sqrt{\zeta}\sqrt{n_0}\phi\big(\gamma + \sigma_{W_2}^2(\eta'-\zeta)\big)$$
$$\big(g_7\big(g_{15}(\zeta\psi - \eta\psi - \zeta\sigma_{W_2}^2) + g_{19}\phi\big) + g_5g_{15}\phi\big) + g_{10}\phi\big(g_{12}\psi(\eta-\zeta)$$
$$- \phi\big(\sqrt{\zeta}g_7\sqrt{n_0} - 1\big)\big(\gamma + \sigma_{W_2}^2(\eta'-\zeta)\big)\big) \tag{S266ad}$$

$$0 = g_{15}\big(g_{12}\psi^2\big(-(\zeta-\eta)^2\big) - \sqrt{\zeta}\sqrt{n_0}\phi\big(\gamma + \sigma_{W_2}^2(\eta'-\zeta)\big)\big(g_7(\zeta\psi - \eta\psi - \zeta\sigma_{W_2}^2) + g_5\phi\big)\big)$$
$$+ g_{10}\phi\big(g_{12}\psi(\eta-\zeta) - \phi\big(\sqrt{\zeta}g_7\sqrt{n_0} - 1\big)\big(\gamma + \sigma_{W_2}^2(\eta'-\zeta)\big)\big) + \zeta g_6g_{13}\psi\phi\big(\gamma + \sigma_{W_2}^2(\eta'-\zeta)\big) \tag{S266ae}$$

$$0 = \zeta g_{10}g_{12}\psi - \eta g_{10}g_{12}\psi + \gamma\sqrt{\zeta}g_7g_{10}\sqrt{n_0}\phi + \gamma\sqrt{\zeta}g_2g_{15}\sqrt{n_0}\phi - \zeta^{3/2}g_7g_{10}\sqrt{n_0}\phi\sigma_{W_2}^2$$
$$- \zeta^{3/2}g_2g_{15}\sqrt{n_0}\phi\sigma_{W_2}^2 + \sqrt{\zeta}\big(g_7g_{10} + g_2g_{15}\big)\sqrt{n_0}\phi\eta'\sigma_{W_2}^2 + \zeta g_4g_6\psi\big(\sigma_{W_2}^2(\zeta-\eta') - \gamma\big)$$
$$- g_3\phi\big(\gamma + \sigma_{W_2}^2(\eta'-\zeta)\big) \tag{S266af}$$

$$0 = \phi\big(g_8\psi\big(\gamma\phi + \sigma_{W_2}^2\big(-\zeta\phi + \phi\eta' + \zeta g_{12}\big)\big) - \sqrt{\zeta}\sqrt{n_0}\big(\gamma + \sigma_{W_2}^2(\eta'-\zeta)\big)\big(g_7\big(g_{15}(\zeta\psi - \eta\psi - \zeta\sigma_{W_2}^2)$$
$$+ (g_{10}+g_{19})\phi\big) + g_2g_{15}\phi\big)\big) + \zeta g_1\psi\big(g_4\phi\big(\gamma + \sigma_{W_2}^2(\eta'-\zeta)\big) - \zeta g_{12}\sigma_{W_2}^4\big)$$
$$+ \zeta g_4\psi\big(g_6\phi\big(\gamma + \sigma_{W_2}^2(\eta'-\zeta)\big) - \zeta g_{12}\sigma_{W_2}^4\big) \tag{S266ag}$$

$$0 = g_6\big(g_{12}\psi^2\big(-(\zeta-\eta)^2\big) - \sqrt{\zeta}\sqrt{n_0}\phi\big(\gamma + \sigma_{W_2}^2(\eta'-\zeta)\big)\big(g_7(\zeta\psi - \eta\psi - \zeta\sigma_{W_2}^2) + g_5\phi\big)\big)$$
$$+ \phi\big(g_{11}\big(g_{12}\psi(\eta-\zeta) + \sqrt{\zeta}g_7\sqrt{n_0}\phi\big(\sigma_{W_2}^2(\zeta-\eta') - \gamma\big)\big) + \sqrt{\zeta}g_1g_2\sqrt{n_0}\phi\big(\sigma_{W_2}^2(\zeta-\eta') - \gamma\big)\big)$$

$$+ g_3\phi\big(g_{12}\psi(\eta - \zeta) - \phi\big(\sqrt{\zeta}g_7\sqrt{n_0} - 1\big)\big(\gamma + \sigma_{W_2}^2\big(\eta' - \zeta\big)\big)\big) \tag{S266ah}$$

$$\begin{aligned}
0 = \sqrt{\zeta}\Big( &- \gamma\zeta g_{14}g_{15}\sqrt{n_0}\psi\phi + \gamma\eta g_{14}g_{15}\sqrt{n_0}\psi\phi - \gamma g_{10}g_{14}\sqrt{n_0}\phi^2 - \gamma g_9 g_{15}\sqrt{n_0}\phi^2 - \gamma g_{14}g_{19}\sqrt{n_0}\phi^2 \\
&+ \zeta g_{14}g_{15}\sqrt{n_0}\phi\sigma_{W_2}^2 + \zeta^2 g_{14}g_{15}\sqrt{n_0}\psi\phi\sigma_{W_2}^2 - \zeta^2 g_{14}g_{15}\sqrt{n_0}\psi\phi\sigma_{W_2}^4 \\
&- \sqrt{n_0}\phi\eta'\sigma_{W_2}^2\big(g_{14}\big(g_{15}\big(\zeta\psi - \eta\psi - \zeta\sigma_{W_2}^2\big) + g_{19}\phi\big) + g_{10}g_{14}\phi + g_9 g_{15}\phi\big) - \zeta\eta g_{14}g_{15}\sqrt{n_0}\psi\phi\sigma_{W_2}^2 \\
&+ \zeta g_{10}g_{14}\sqrt{n_0}\phi^2\sigma_{W_2}^2 + \zeta g_9 g_{15}\sqrt{n_0}\phi^2\sigma_{W_2}^2 + \zeta g_{14}g_{19}\sqrt{n_0}\phi^2\sigma_{W_2}^2 + \sqrt{\zeta}g_1 g_{18}\psi\phi\big(\gamma + \sigma_{W_2}^2\big(\eta' - \zeta\big)\big) \\
&+ \sqrt{\zeta}g_6 g_{18}\psi\phi\big(\gamma + \sigma_{W_2}^2\big(\eta' - \zeta\big)\big) - \zeta^{3/2}g_{12}g_{18}\psi\sigma_{W_2}^4\Big) + g_{17}\psi\phi\big(\gamma\phi + \sigma_{W_2}^2\big(\phi\big(\eta' - \zeta\big) + \zeta g_{12}\big)\big) \tag{S266ai}
\end{aligned}$$

$$\begin{aligned}
0 = \; &\gamma g_{16}\psi\phi^2 - \gamma\zeta^{3/2}g_7 g_{15}\sqrt{n_0}\psi\phi + \gamma\sqrt{\zeta}\eta g_7 g_{15}\sqrt{n_0}\psi\phi - \gamma\sqrt{\zeta}g_7 g_{10}\sqrt{n_0}\phi^2 - \gamma\sqrt{\zeta}g_5 g_{15}\sqrt{n_0}\phi^2 \\
&- \gamma\sqrt{\zeta}g_7 g_{19}\sqrt{n_0}\phi^2 + \gamma\zeta^{3/2}g_7 g_{15}\sqrt{n_0}\phi\sigma_{W_2}^2 - \zeta^{3/2}\eta g_7 g_{15}\sqrt{n_0}\psi\phi\sigma_{W_2}^2 + \zeta^{3/2}g_7 g_{10}\sqrt{n_0}\phi^2\sigma_{W_2}^2 \\
&+ \zeta^{3/2}g_5 g_{15}\sqrt{n_0}\phi^2\sigma_{W_2}^2 + \zeta^{3/2}g_7 g_{19}\sqrt{n_0}\phi^2\sigma_{W_2}^2 + \zeta^{5/2}g_7 g_{15}\sqrt{n_0}\psi\phi\sigma_{W_2}^2 - \zeta^{5/2}g_7 g_{15}\sqrt{n_0}\phi\sigma_{W_2}^4 \\
&+ \phi\eta'\sigma_{W_2}^2\big(g_{16}\psi\phi - \sqrt{\zeta}\sqrt{n_0}\big(g_7\big(g_{15}\big(\zeta\psi - \eta\psi - \zeta\sigma_{W_2}^2\big) + \big(g_{10} + g_{19}\big)\phi\big) + g_5 g_{15}\phi\big)\big) \\
&+ \zeta g_1 g_{13}\psi\phi\big(\gamma + \sigma_{W_2}^2\big(\eta' - \zeta\big)\big) + \zeta g_6 g_{13}\psi\phi\big(\gamma + \sigma_{W_2}^2\big(\eta' - \zeta\big)\big) - \zeta^2 g_{12}g_{13}\psi\sigma_{W_2}^4 - \zeta g_{16}\psi\phi^2\sigma_{W_2}^2 \\
&+ \zeta g_{12}g_{16}\psi\phi\sigma_{W_2}^2 \tag{S266aj}
\end{aligned}$$

After some straightforward algebra, one can eliminate all $g_i$ except for $g_7$ and $g_{12}$, which satisfy coupled polynomial equations. Those equations can be shown to be identical to eqn. (S66) by invoking the change of variables,

$$g_7 = -\frac{\sqrt{\zeta}\psi}{\sqrt{n_0}\phi}\tau_2, \quad \text{and} \quad g_{12} = \big(\gamma + \sigma_{W_2}^2\big(\eta' - \zeta\big)\big)\tau_1. \tag{S267}$$

In terms of the related variables defined in eqn. (S89), the error $E_{72}$ is given by,

$$\begin{aligned}
E_{72} = &-\frac{\psi\tilde{\tau}_1^2\big(2\zeta\psi\tilde{\tau}_1(\zeta - 2\eta) + \phi\big(\zeta^2\psi - \zeta\eta(\psi + 1) - \eta^2\psi\big)\big)}{2\zeta\phi\big(\phi^2 - \psi\tilde{\tau}_1^2\big)} + \frac{\psi^2\tilde{\tau}_1^2(\zeta - \eta)^3}{2\zeta^2\big(\tilde{\tau}_2 + 1\big)^2\big(\psi\tilde{\tau}_1^2 - \phi^2\big)} \\
&+ \frac{\psi\tilde{\tau}_1\tilde{\tau}_2\big(\psi\tilde{\tau}_1 + \phi\big)\big(\zeta\tilde{\tau}_1 + \eta\phi\big)}{\phi^3 - \psi\phi\tilde{\tau}_1^2} + \frac{\psi^2\tilde{\tau}_1^2(\zeta - \eta)^2\big(\tilde{\tau}_1 + \phi\big)}{\zeta\phi\big(\tilde{\tau}_2 + 1\big)\big(\phi^2 - \psi\tilde{\tau}_1^2\big)} + \frac{\zeta\tilde{\tau}_2^2\big(\psi\tilde{\tau}_1 + \phi\big)^2}{2\big(\phi^2 - \psi\tilde{\tau}_1^2\big)} \tag{S268} \\
= &-T_2/\tau_1' - E_{22} - \eta\sigma_{W_2}^2, \tag{S269}
\end{aligned}$$

where $T_2$ is given in eqn. (S65).

## S6.6   $H_{101}$

$$\begin{aligned}
H_{101} &= \mathbb{E}\hat{y}(\mathbf{x}; P, X, \varepsilon)\hat{y}(\mathbf{x}; P, \tilde{X}, \varepsilon) \tag{S270} \\
&= \mathbb{E}\Big[N_0(\mathbf{x}; P)N_0(\mathbf{x}; P)^\top + K(\mathbf{x}, \tilde{X}; \mathbb{P})K(\tilde{X}, \tilde{X}; P)^{-1}Y(\tilde{X}, \varepsilon)^\top Y(X, \varepsilon)K(X, X; P)^{-1}K(X, \mathbf{x}; P) \\
&\quad + K(\mathbf{x}, \tilde{X}; \mathbb{P})K(\tilde{X}, \tilde{X}; P)^{-1}N_0(\tilde{X})^\top N_0(X)K(X, X; P)^{-1}K(X, \mathbf{x}; P) \\
&\quad - N_0(\mathbf{x}; P)N_0(X; P)K(X, X; P)^{-1}K(X, \mathbf{x}; P) - N_0(\mathbf{x}; P)N_0(\tilde{X}; P)K(\tilde{X}, \tilde{X}; P)^{-1}K(\tilde{X}, \mathbf{x}; P)\Big] \tag{S271} \\
&= \nu\sigma_{W_2}^2\eta + \nu E_{22} + \mathbb{E}\,\mathrm{tr}\,\big(K(\tilde{X}, \tilde{X}; P)^{-1}(\tilde{X}^\top X + \nu\frac{\sigma_{W_2}^2}{n_1}f(W_1\tilde{X})^T F)K(X, X; P)^{-1}K(X, \mathbf{x}; P)K(\mathbf{x}, \tilde{X}; P)\big) \tag{S272} \\
&= H_{100}, \tag{S273}
\end{aligned}$$

## S6.7 $H_{110}$

$$H_{110} = \mathbb{E}\hat{y}(\mathbf{x}; P, X, \varepsilon)\hat{y}(\mathbf{x}; P, X, \tilde{\varepsilon}) \tag{S274}$$

$$= \mathbb{E}\Big[N_0(\mathbf{x}; P)N_0(\mathbf{x}; P)^\top + K(\mathbf{x}, ; \mathbb{P})K(X, X; P)^{-1}Y(X, \tilde{\varepsilon})^\top Y(X, \varepsilon)K(X, X; P)^{-1}K(X, \mathbf{x}; P)$$

$$+ K(\mathbf{x}, X; \mathbb{P})K(X, X; P)^{-1}N_0(X)^\top N_0(X)K(X, X; P)^{-1}K(X, \mathbf{x}; P)$$

$$- 2N_0(\mathbf{x}; P)N_0(X; P)K(X, X; P)^{-1}K(X, \mathbf{x}; P)\Big] \tag{S275}$$

$$= \mathbb{E}\Big[N_0(\mathbf{x}; P)N_0(\mathbf{x}; P)^\top + K(\mathbf{x}, ; \mathbb{P})K(X, X; P)^{-1}X^\top X K(X, X; P)^{-1}K(X, \mathbf{x}; P)$$

$$+ K(\mathbf{x}, X; \mathbb{P})K(X, X; P)^{-1}N_0(X)^\top N_0(X)K(X, X; P)^{-1}K(X, \mathbf{x}; P)$$

$$- 2N_0(\mathbf{x}; P)N_0(X; P)K(X, X; P)^{-1}K(X, \mathbf{x}; P)\Big] \tag{S276}$$

$$= \nu\sigma_{W_2}^2\eta + \nu E_{22} + E_{32} + \nu E_{33} \tag{S277}$$

## S6.8 $H_{111}$

$$H_{111} = \mathbb{E}\hat{y}(\mathbf{x}; P, X, \varepsilon)\hat{y}(\mathbf{x}; P, X, \varepsilon) \tag{S278}$$

$$= \mathbb{E}\Big[N_0(\mathbf{x}; P)N_0(\mathbf{x}; P)^\top + K(\mathbf{x}, ; \mathbb{P})K(X, X; P)^{-1}Y(X, \varepsilon)^\top Y(X, \varepsilon)K(X, X; P)^{-1}K(X, \mathbf{x}; P)$$

$$+ K(\mathbf{x}, X; \mathbb{P})K(X, X; P)^{-1}N_0(X)^\top N_0(X)K(X, X; P)^{-1}K(X, \mathbf{x}; P)$$

$$- 2N_0(\mathbf{x}; P)N_0(X; P)K(X, X; P)^{-1}K(X, \mathbf{x}; P)\Big] \tag{S279}$$

$$= \mathbb{E}\Big[N_0(\mathbf{x}; P)N_0(\mathbf{x}; P)^\top + K(\mathbf{x}, ; \mathbb{P})K(X, X; P)^{-1}(X^\top X + \sigma_\varepsilon^2 n_1 I_m)K(X, X; P)^{-1}K(X, \mathbf{x}; P)$$

$$+ K(\mathbf{x}, X; \mathbb{P})K(X, X; P)^{-1}N_0(X)^\top N_0(X)K(X, X; P)^{-1}K(X, \mathbf{x}; P)$$

$$- 2N_0(\mathbf{x}; P)N_0(X; P)K(X, X; P)^{-1}K(X, \mathbf{x}; P)\Big] \tag{S280}$$

$$= \nu\sigma_{W_2}^2\eta + \nu E_{22} + E_{31} + E_{32} + \nu E_{33} \tag{S281}$$

## S6.9 Combining results: asymptotic variance terms

Summarizing the above result, we have,

$$H_{000} = E_4$$
$$H_{001} = E_4$$
$$H_{010} = E_5$$
$$H_{011} = E_5 + E_6$$
$$H_{100} = \nu\sigma_{W_2}^2\eta + \nu E_{22} + E_{71} + \nu E_{72}$$
$$H_{101} = \nu\sigma_{W_2}^2\eta + \nu E_{22} + E_{71} + \nu E_{72}$$
$$H_{110} = \nu\sigma_{W_2}^2\eta + \nu E_{22} + E_{32} + \nu E_{33}$$
$$H_{111} = \nu\sigma_{W_2}^2\eta + \nu E_{22} + E_{31} + E_{32} + \nu E_{33},$$

which using eqn. (S18) gives,

$$B = 1 + E_{21} + E_4$$

$$= \tau_2^2/\tau_1^2$$

$$V_P = H_{100} - H_{000}$$
$$= \nu\sigma_{W_2}^2\eta + \nu E_{22} + E_{71} + \nu E_{72} - E_4$$
$$= E_{71} - E_4 - \nu T_2/\tau_1'$$
$$= \tau_2'/\tau_1' + 2\tau_2/\tau_1 - 1 - (\tau_2/\tau_1 - 1)^2 - \nu T_2/\tau_1'$$
$$= \tau_2'/\tau_1' - B - \nu T_2/\tau_1'$$

$$V_X = H_{010} - H_{000}$$
$$= E_5 - E_4$$
$$= \phi\tilde{\tau}_2^2(\tilde{\tau}_2 + 1)^2/(1 - \phi\tilde{\tau}_2^2)$$
$$= \phi B(\tau_1 - \tau_2)^2/(\tau_1^2 - \phi(\tau_1 - \tau_2)^2)$$

$$V_\varepsilon = H_{001} - H_{000}$$
$$= 0$$

$$V_{PX} = H_{110} - H_{010} - H_{100} + H_{000}$$
$$= \nu\sigma_{W_2}^2\eta + \nu E_{22} + E_{32} + \nu E_{33} - E_5 - (\nu\sigma_{W_2}^2\eta + \nu E_{22} + E_{71} + \nu E_{72}) + E_4$$
$$= E_{32} - E_{71} - E_5 + E_4 + \nu(E_{33} - E_{72})$$
$$= 1 - 2\tau_2/\tau_1 - \tau_2'/\tau_1^2 - (\tau_2'/\tau_1' - 2\tau_2/\tau_1 + 1) - V_X$$
$$\quad + \nu\left[\sigma_{W_2}^2\left[(\tau_1 + (\sigma_{W_2}^2(\eta' - \zeta) + \gamma)\tau_1' + \sigma_{W_2}^2\zeta\tau_2')/\tau_1^2 - \eta\right] - E_{22} - (T_2 - E_{22} - \eta\sigma_{W_2}^2))\right]$$
$$= -\tau_2'/\tau_1^2 - \tau_2'/\tau_1' - V_X + \nu\left[\sigma_{W_2}^2\left[(\tau_1 + (\sigma_{W_2}^2(\eta' - \zeta) + \gamma)\tau_1' + \sigma_{W_2}^2\zeta\tau_2')/\tau_1^2\right] - T_2)\right]$$
$$= -\tau_2'/\tau_1^2 - B - V_P - V_X + \nu T_2/(\gamma\tau_1)^2$$

$$V_{P\varepsilon} = H_{101} - H_{001} - H_{100} + H_{000}$$
$$= \nu\sigma_{W_2}^2\eta + \nu E_{22} + E_{71} + \nu E_{72} - E_4 - (\nu\sigma_{W_2}^2\eta + \nu E_{22} + E_{71} + \nu E_{72}) + E_4$$
$$= 0$$

$$V_{X\varepsilon} = H_{011} - H_{001} - H_{010} + H_{000}$$
$$= E_5 + E_6 - E_4 - E_5 + E_4$$
$$= E_6$$
$$= \sigma_\varepsilon^2\phi\tilde{\tau}_2^2/(1 - \tilde{\tau}_2^2\phi)$$
$$= \sigma_\varepsilon^2 V_X/B$$

$$V_{PX\varepsilon} = \nu\sigma_{W_2}^2\eta + \nu E_{22} + E_{31} + E_{32} + \nu E_{33} - (E_5 + E_6) - (\nu\sigma_{W_2}^2\eta + \nu E_{22} + E_{71} + \nu E_{72})$$
$$\quad - (\nu\sigma_{W_2}^2\eta + \nu E_{22} + E_{32} + \nu E_{33}) + E_4 + E_5 + \nu\sigma_{W_2}^2\eta + \nu E_{22} + E_{71} + \nu E_{72} - E_4$$
$$= E_{31} - E_6$$
$$= \sigma_\varepsilon^2\left(-\tau_1'/\tau_1^2 - 1\right) - V_{X\varepsilon}.$$

Therefore, we have established the main result, Theorem S3.

## S7 Proof of Corollary 1

### S7.1 Bias is non-increasing

In terms of the auxiliary variables $\tilde\tau_1$ and $\tilde\tau_2$ defined in eqn. (S89), the coupled equations defining $\tau_1$ and $\tau_2$, eqn. (S42), simplify to

$$0 = \gamma\phi\tilde\tau_2 - \gamma\tilde\tau_1 + \sigma_{W_2}^2\left(\tilde\tau_2\left(\zeta\phi\tilde\tau_2 + \zeta + \phi\eta'\right) + \tilde\tau_1\left(\eta - \eta'\right) + \zeta\right) \tag{S282}$$

$$0 = \left(\tilde\tau_1 - \phi\tilde\tau_2\right)\left(\psi\tilde\tau_1\left(\zeta\tilde\tau_2 + \eta\right) + \zeta\phi\left(\tilde\tau_2 + 1\right)\right) + \zeta\phi\tilde\tau_1\left(\tilde\tau_2 + 1\right)\sigma_{W_2}^2 . \tag{S283}$$

Eliminating $\tilde\tau_1$ from these equations gives,

$$\zeta\phi\left(\tilde\tau_2 + 1\right)\left(\left(\eta - \eta'\right)\sigma_{W_2}^2 - \gamma\right)\left(\tilde\tau_2\left(\zeta\phi\tilde\tau_2 + \phi(\gamma + \eta) + \zeta\right) + \sigma_{W_2}^2\left(\tilde\tau_2\left(\zeta\phi\tilde\tau_2 + \zeta + \phi\eta'\right) + \zeta\right) + \zeta\right)$$
$$= \psi\left(\zeta\tilde\tau_2 + \eta\right)\left(\tilde\tau_2\left(\zeta\phi\tilde\tau_2 + \zeta + \eta\phi\right) + \zeta\right)\left(\gamma\phi\tilde\tau_2 + \sigma_{W_2}^2\left(\tilde\tau_2\left(\zeta\phi\tilde\tau_2 + \zeta + \phi\eta'\right) + \zeta\right)\right) \tag{S284}$$

Specializing to the random feature kernel ($\sigma_{W_2} = 0$), the equation becomes,

$$\left(\tilde\tau_2\left(\zeta\psi\tilde\tau_2 + \zeta + \eta\psi\right) + \zeta\right)\left(\tilde\tau_2\left(\zeta\phi\tilde\tau_2 + \zeta + \eta\phi\right) + \zeta\right) = -\gamma\zeta\phi\tilde\tau_2\left(\tilde\tau_2 + 1\right) . \tag{S285}$$

In the ridgeless limit, $\gamma = 0$, and the quartic equation factorizes into the product of two quadratic polynomials. The root of these equations that respects the conditions of Lemma 1 is given by

$$\tilde\tau_2 = \frac{-\zeta - \eta\omega + \sqrt{(\zeta + \eta\omega)^2 - 4\zeta^2\omega}}{2\zeta\omega} , \tag{S286}$$

where $\omega = \max\{\phi, \psi\}$. Next, recall from Theorem 1 that $B = \tau_2^2/\tau_1^2 = (1 + \tilde\tau_2)^2$ so that

$$\frac{\partial B}{\partial n_1} = -\frac{\psi^2}{n_0}\frac{\partial B}{\partial\psi}(1 + \tilde\tau_2)^2 = -2\frac{\psi^2}{n_0}(1 + \tilde\tau_2)\frac{\partial\tilde\tau_2}{\partial\psi} . \tag{S287}$$

To show that $\partial B/\partial n_1 \le 0$, we show that $(1 + \tilde\tau_2) \ge 0$ and that $\partial\tilde\tau_2/\partial\psi \ge 0$. First of all,

$$1 + \tilde\tau_2 = 1 + \frac{-\zeta - \eta\omega + \sqrt{(\zeta + \eta\omega)^2 - 4\zeta^2\omega}}{2\zeta\omega} \tag{S288}$$

$$= \frac{-(-2\zeta\omega + \zeta + \eta\omega) + \sqrt{(\zeta + \eta\omega)^2 - 4\zeta^2\omega}}{2\zeta\omega} \tag{S289}$$

$$\ge \frac{-\sqrt{(-2\zeta\omega + \zeta + \eta\omega)^2} + \sqrt{(\zeta + \eta\omega)^2 - 4\zeta^2\omega}}{2\zeta\omega} \tag{S290}$$

$$= \frac{-\sqrt{(\zeta + \eta\omega)^2 - 4\zeta^2\omega - 4\zeta(\eta - \zeta)\omega^2} + \sqrt{(\zeta + \eta\omega)^2 - 4\zeta^2\omega}}{2\zeta\omega} \ge 0 , \tag{S291}$$

where we used the relation $\eta \ge \zeta$ which was proved in [26]. As for the derivative, note that $\partial\tilde\tau_2/\partial\psi = 0$ if $\psi < \phi$ and otherwise,

$$\frac{\partial\tilde\tau_2}{\partial\psi} = \frac{-(-2\zeta\omega + \zeta + \eta\omega) + \sqrt{(\zeta + \eta\omega)^2 - 4\zeta^2\omega}}{2\omega^2\sqrt{(\zeta + \eta\omega)^2 - 4\zeta^2\omega}} \tag{S292}$$

$$\ge \frac{-\sqrt{(-2\zeta\omega + \zeta + \eta\omega)^2} + \sqrt{(\zeta + \eta\omega)^2 - 4\zeta^2\omega}}{2\omega^2\sqrt{(\zeta + \eta\omega)^2 - 4\zeta^2\omega}} \tag{S293}$$

$$= \frac{-\sqrt{(\zeta + \eta\omega)^2 - 4\zeta^2\omega - 4\zeta(\eta - \zeta)\omega^2} + \sqrt{(\zeta + \eta\omega)^2 - 4\zeta^2\omega}}{2\omega^2\sqrt{(\zeta + \eta\omega)^2 - 4\zeta^2\omega}} \tag{S294}$$

$$\ge 0 . \tag{S295}$$

Therefore we have shown that

$$\frac{\partial B}{\partial n_1} \le 0 , \tag{S296}$$

i.e. the bias $B$ is monotonically decreasing.

## S7.2 Behavior near the interpolation boundary

From inspection of the expressions in Theorem 1, the bias and variance terms depend on $\tau_1$ and $\tau_2$ through four ratios, $\tau_2/\tau_1$, $\tau_2'/\tau_1'$, $\tau_2'/\tau_1^2$ and $\tau_1'/\tau_1^2$. In the ridgeless ($\gamma = 0$) limit, these ratios can all be expressed in terms of $\tilde{\tau}_2$ by using eqns. (S87), (S88), (S89) and (S285).

We examine the behavior near the interpolation threshold $\phi = \psi$ by taking the limit from both directions. It is straightforward algebraic substitution to show that for $\phi < \psi$,

$$\frac{\tau_2}{\tau_1} = \frac{\tau_2'}{\tau_1'} = 1 + \tilde{\tau}_2 \,, \quad \frac{\tau_1'}{\tau_1^2} = \frac{\zeta\,(\tilde{\tau}_2 + 1)}{(\phi - \psi)\tilde{\tau}_2\,(\zeta\tilde{\tau}_2 + \eta)} \,, \quad \frac{\tau_2'}{\tau_1^2} = \frac{\zeta\,(\tilde{\tau}_2 + 1)^{\,2}}{(\phi - \psi)\tilde{\tau}_2\,(\zeta\tilde{\tau}_2 + \eta)} \,. \tag{S297}$$

From eqn. (S286), we see that $\tilde{\tau}_2$ is finite when $\phi = \psi$ so that the terms in eqn. (S297) obey,

$$\frac{\tau_2}{\tau_1} = \frac{\tau_2'}{\tau_1'} = \mathcal{O}(1) \,, \quad \frac{\tau_1'}{\tau_1^2} = \mathcal{O}\left(\frac{1}{\phi - \psi}\right) \,, \quad \frac{\tau_2'}{\tau_1^2} = \mathcal{O}\left(\frac{1}{\phi - \psi}\right) \,. \tag{S298}$$

Turning now to the case of $\phi > \psi$, similar algebraic substitutions yield,

$$\frac{\tau_2}{\tau_1} = \tilde{\tau}_2 + 1 \tag{S299}$$

$$\frac{\tau_2'}{\tau_1'} = \frac{\zeta^2\tilde{\tau}_2\left(\tilde{\tau}_2^2(-\psi + \phi + 1) + 3\tilde{\tau}_2 + 2\right) + \zeta\eta\left(\tilde{\tau}_2^3(\psi - \phi) + \tilde{\tau}_2^2(\phi - \psi) + \tilde{\tau}_2 + 1\right) + \eta^2\tilde{\tau}_2^2(\psi - \phi)}{\zeta\left(\zeta\tilde{\tau}_2\left(\tilde{\tau}_2^2(\phi - \psi) + \tilde{\tau}_2 + 2\right) + \eta\tilde{\tau}_2^2(\phi - \psi) + \eta\right)} \tag{S300}$$

$$\frac{\tau_1'}{\tau_1^2} = \frac{\zeta\,(\tilde{\tau}_2 + 1)}{\tilde{\tau}_2(\psi - \phi)\,(\zeta\tilde{\tau}_2 + \eta)} - \frac{\zeta\tilde{\tau}_2\,(\tilde{\tau}_2 + 1)}{\zeta\tilde{\tau}_2\,(\tilde{\tau}_2 + 2) + \eta} \tag{S301}$$

$$\frac{\tau_2'}{\tau_1^2} = \frac{\zeta\,(\tilde{\tau}_2 + 1)^{\,2}}{\tilde{\tau}_2(\psi - \phi)\,(\zeta\tilde{\tau}_2 + \eta)} + \frac{\tilde{\tau}_2\,(\tilde{\tau}_2 + 1)\,(\eta - \zeta)}{\zeta\tilde{\tau}_2\,(\tilde{\tau}_2 + 2) + \eta} \,. \tag{S302}$$

We can isolate the pole at $\phi = \psi$ by examining the relevant functions of $\tilde{\tau}_2$. In particular, substituting the solution (S286) gives,

$$\frac{\tau_2'}{\tau_1'}\big|_{\psi=\phi} = \frac{\sqrt{(\zeta + \eta\phi)^2 - 4\zeta^2\phi} + 2\zeta\phi - \zeta - \eta\phi}{2\zeta\phi} \tag{S303}$$

which is evidently finite when $\phi = \psi$ and,

$$\frac{\tilde{\tau}_2\,(\tilde{\tau}_2 + 1)}{\zeta\tilde{\tau}_2\,(\tilde{\tau}_2 + 2) + \eta} = \frac{2\zeta\phi}{(\zeta + \eta\phi)\sqrt{(\zeta + \eta\phi)^2 - 4\zeta^2\phi} + (\zeta + \eta\phi)^2 - 4\zeta^2\phi} \tag{S304}$$

whose denominator is a sum of non-negative terms that only vanishes if $\phi = 1$ and $\eta = \zeta$, i.e. the activation function is linear. Therefore, we find for $\phi > \psi$ the same behavior as for $\phi < \psi$, namely,

$$\frac{\tau_2}{\tau_1} = \frac{\tau_2'}{\tau_1'} = \mathcal{O}(1) \,, \quad \frac{\tau_1'}{\tau_1^2} = \mathcal{O}\left(\frac{1}{\phi - \psi}\right) \,, \quad \frac{\tau_2'}{\tau_1^2} = \mathcal{O}\left(\frac{1}{\phi - \psi}\right) \,. \tag{S305}$$

Altogether, we conclude that as $\phi \to \psi$,

$$B = \mathcal{O}(1) \,, \quad V_P = \mathcal{O}(1) \,, \quad V_X = \mathcal{O}(1) \,, \quad V_{X\varepsilon} = \mathcal{O}(1) \,, \quad V_{PX} = \mathcal{O}\left(\frac{1}{\phi - \psi}\right) \,, \quad V_{PX\varepsilon} = \mathcal{O}\left(\frac{1}{\phi - \psi}\right) \,, \tag{S306}$$

i.e. the only divergent terms are $V_{PX}$ and $V_{PX\varepsilon}$.

Figure S3: The multivariate variance decomposition of [35]. Following the setup of Fig. 1, panel (a) depicts the decomposition with a Venn diagram and panel (b) shows plots of the individual terms as functions of the overparameterization ratio $n_1/m$. The total variance is partitioned into three terms in a sequential manner, breaking the symmetry of the random variables and failing to account for their interactions. Since it is those interactions that cause the divergences (see Corollary 1), it is not possible to unambiguously attribute the divergences to a univariate source of variance, despite the the observed spikes in $\mathcal{E}_{\text{Noise}}$ and $\mathcal{E}_{\text{Init}}$.

## S8    The Bias-Variance Decomposition of d'Ascoli *et al.* [35]

While finalizing this manuscript, we became aware of a related work [35] that similarly proposes and calculates a multivariate variance decomposition in order to examine the origins of double descent. Their approach is sequential in nature, first defining $\mathcal{E}_{\text{Noise}}$ to be the (expected) variance conditional on $P$ and $X$, then $\mathcal{E}_{\text{Init}}$ to be the remaining variance conditional on $X$, and finally $\mathcal{E}_{\text{Samp}}$ to be the remaining variance. In terms of our fine-grained decomposition, their expressions read,

$$\mathcal{E}_{\text{Bias}} = B\,, \quad \mathcal{E}_{\text{Init}} = V_P + V_{PX}\,, \quad \mathcal{E}_{\text{Samp}} = V_X\,, \quad \mathcal{E}_{\text{Noise}} = V_{PX\varepsilon} + V_{X_\varepsilon} + V_{P\varepsilon} + V_\varepsilon\,. \tag{S307}$$

Fig. S3(a) illustrates their decomposition in terms of a Venn diagram and Fig. S3(b) shows how the components of their decomposition behave as the number of random features varies, similarly to Figs. 1 and S2. Note that their total bias and total variance agree with ours, and that their decomposition also resolves the two separate divergent terms at the interpolation threshold (since $\mathcal{E}_{\text{Noise}}$ contains $V_{PX\varepsilon}$ and $\mathcal{E}_{\text{Init}}$ contains $V_{PX}$). However, because their decomposition is not fully multivariate, the resulting areas do not necessarily possess the interpretations one might expect from the names "noise variance," "initialization variance," and "sampling variance." For example, the divergence in $\mathcal{E}_{\text{Noise}}$ ultimately comes from the contribution of $V_{PX\varepsilon}$, which vanishes when you ensemble over initial parameters, for example. This strong dependence on the parameters does not seem like a desirable property of a quantity designed to measure the variance due to noise. Similarly, the divergence of $\mathcal{E}_{\text{Init}}$ can be eliminated by ensembling (bagging) over different training samples, which also seems like a undesirable property of "initialization variance." The underlying reason for these inconsistent interpretations is that the divergences ultimately arise from the *interaction* terms $V_{PX}$ and $V_{PX\varepsilon}$, but these interactions are not captured in their decomposition.

## Footnotes

[5]We assume the width $n_{\mathrm{T}} \to \infty$, but the rate is not important.

[6]Any non-zero $\sigma_{W_1}^2$ can be absorbed into a redefinition of $\sigma$.

[7]If the width is not asymptotically larger than the dataset size, the kernel system may not accurately describe the late-time predictions of the neural network.