[Reviews · NeurIPS 2020]

Review 1

Summary and Contributions: This paper studies the double descent phenomenon in random features models. To disentangle the variance with respect to the random training set, the random features and the noise, a notion of symmetric decomposition of the variance is introduced. The asymptotic behavior of the resulting 7 types of variance and of the bias is then recovered, allowing a precise understanding of which 'type' of variance leads to the explosion of variance at the transition from the under- to over-parametrized regimes, characterizing the double-descent curve.

Strengths: The article is very clear and the proposed decomposition of the variance is well motivated. Though I haven't verified the proofs in details, I haven't observed any issues. The experiments, though not very extensive show a very good match between the theoretical predictions and numerical experiments.

Weaknesses: As mentioned, the data distribution is a bit simplistic (i.i.d. Gaussian inputs and ouptuts depending linearly on the inputs), the reader is referred to the appendix for a motivation for this simple choice, however no exact section of the appendix is given and I could not find it (or the section I found which seems to discuss something related doesn't really address this aspect).

Correctness: The proofs appear to be correct, though I haven't checked them in details.

Clarity: The paper is very well-written.

Relation to Prior Work: There has been a lot of recent results which are very closely related and should be mentioned and if possible discussed: (Implicit Regularization of Random Feature Models, Jacot et al. ICML 2020) introduces a notion of explicit ridge which explains the evolution of the bias as a function of the number of datapoints. (Double Trouble in Double Descent : Bias and Variance(s) in the Lazy Regime, d'Ascoli et al. 2020) also decomposes the loss of random features into the different types of variances. Finally, the NTK is mentioned multiple times, but no citation is given. I don't think that the NTK is so well-known that it needs no introduction and even then credit should be properly given, so please cite at least (Neural Tangent Kernel: Convergence and Generalization in Neural Networks, Jacot et al. NeurIPS 2018).

Reproducibility: Yes

Additional Feedback:


Review 2

Summary and Contributions: Post-rebuttal update: I am happy to read the authors' clear responses on related work. I am increasing my score and would be happy to recommend acceptance. ========= The authors improve upon the work by Neal et al, 2018 with the goal of disentangling the sources of variance contributing to the irregular bias-variance behavior of certain ML methods, like neural networks. The variance decomposition used in Neal et al is not unique: it critically relies on the order of the conditioning in the law of total variance. Hence, the interpretation of the decomposition results comes with some ambiguity. The main contribution of the present paper is a variance decomposition that is symmetric and unique. Said uniqueness comes with the promise of an unambiguous attribution of interesting variance phenomena to specific variance sources (or combinations of variance sources interacting). The authors analyze the various bias and variance components for random feature kernel regression. They attribute the typical ‘spike’ in variance to the variance term that is related to the interaction between the randomness in the parameters and randomness in the data. An interesting conclusion, is that If we eliminate either the variance from parameters or the variance from the data, then the spike can be attenuated or eliminated. The authors provide some results on ensembles and bagging to corroborate this conclusion. I am happy to see this kind of work plugging an obvious hole in the methodology of past work and I think that with minor improvements it should be accepted. My main concern are about the discussion and citation of important prior work and its contributions. Secondarily, I would have loved to see more intuition about variance terms that involve interaction between sources. Details below.

Strengths: - This paper plugs an important methodological hole of previous work. It does a better job at disentangling the sources of variance that contribute to prediction variance in NNs (and other similar models). - The analysis of the random feature model is a nice way to get example curves for the terms of the variance decomposition. - An important piece of insight from the above analysis is corroborated in the context of ensembling and bagging. I believe that the presented methodology has the capacity to yield very useful actionable insights that will drive further research.

Weaknesses: The most important weakness has to do with the citation and discussion around prior work. Please see details in the relevant section below. With this issue addressed, I will be happy to propose acceptance. A secondary points (that has no bearing on the acceptance decision) has to do with my wish for more clarity insight around some of the quantities (cf. Clarity section below).

Correctness: As far as I can tell the methodology and results are correct.

Clarity: The paper is relatively well written. My main complaint on this front is that intuition around quantities like V_{PD} is largely missing. I think I understand that this term has to do with variance that ought to be attributed to some kind of interaction between parameter randomness and data randomness but beyond this generality I don't know what this means exactly. The discussion around ensembling and bagging helps. To be clear, I will not be taking this particular point as a 'negative' that will influence my recommendation for acceptance or rejection. For all I know, fully getting to the bottom of this question requires many more research papers.

Relation to Prior Work: I think that the way the paper deals with existing work is unfortunate. First the citation and discussion around Neal et al. 2018 is a bit sloppy. Then, there are two important missing pieces of work that ought to be cited and discussed. Details below. - Neal et al., 2018 revisit the direct study of bias and variance on NNs via measurement for the first time since Geman et al., 1992 as far as I know. The paper was first to show, before the ‘double descent’ paper, that prediction variance in the overparametrized regime decreases as the NN width increases (something that is in Geman et al.’s plots, but they actually dismissed a training/numerical problem). They also propose a variance decomposition to identify the various sources. Given that the main focus of paper under review is to improve upon the methodology of Neal et al. on that last contribution it is unfortunate that Neal et al is not properly cited for its contributions in the introduction. - Yang et al., 2020 (below) is very relevant because it experiments with the effect that a few different sources of randomness can have on the spike. - d’Ascoli et al., 2020 (below) covers some of the same topics. The authors provide a variance decomposition and they study the effect of ensembling. It is quite recent work, but according to the NeurIPS reviewing guidelines it is not strictly speaking contemporaneous as it appeared >two months before the full paper deadline. I invite the authors to consider this literature in their rebuttal discussion and in their revision. EXTRA REFERENCES Yang, Z., Yu, Y., You, C., Steinhardt, J., & Ma, Y. (2020). Rethinking bias-variance trade-off for generalization of neural networks. arXiv preprint arXiv:2002.11328. d'Ascoli, S., Refinetti, M., Biroli, G., & Krzakala, F. (2020). Double trouble in double descent: Bias and variance (s) in the lazy regime. arXiv preprint arXiv:2003.01054.

Reproducibility: Yes

Additional Feedback:


Review 3

Summary and Contributions: ====== AFTER FEEDBACK ======== I have read the author's response, and I am generally satisfied with it except for two points: I disagree with the comment that linear models can not be studied with varying d--there is some confusion in the literature about it but it can be done see section 2 of the recent paper https://arxiv.org/abs/2006.10189 I also insist on clarifying that prior works did not make a mistake in section 4.3 Nevertheless, I appreciate the good response that the authors prepared and I have accordingly increased my score to 7 (and I reiterate that the authors take into account the above two points to whatever extent possible in the revision) ============================== This work provides a finer decomposition of the bias-variance terms to better understand the tradeoff between different terms. In particular, they decompose the variance into several conditional (like) variances trying to correctly attribute the variation of test error to change in parameters, change in design, noise in observations.

Strengths: The fine-grained decomposition of the variance into various terms, P, D, epsilon seems like a good contribution of the work which can help tease apart the effects of different sources of randomness on test error in various ML settings. The work is a good starting point with some clean theoretical results for random feature kernel regression.

Weaknesses: While the finer decomposition of variance in the paper is a good step, several issues remain. Some notation in the paper is confusing. The choice of the models and fitted models is not really well explained (e.g., What happens if the fitted model is linear too?). The notation in Prop 1 is confusing and the meaning of P and D is not easily understood beyond the context of the model considered in the paper (which in fact is presented later in the paper). E.g., What would P mean for linear models? Decision Trees? DNNs? Also, some clarifications on the decomposition in example 2---Is that decomposition optimal / enough in some sense?--would be useful.

Correctness: The appendix is too long to rigorously check for correctness. While some results make sense intuitively, some results are not explained/discussed much (other than explaining the trend) making it a bit puzzling at first. See other comments for more discussion. Empirical section has few results which look correct to me. -- I appreciate the fact that the bias-variance discussion was restricted to a setting where data is drawn from a fixed generative model and only the fitted model was varied (something often confused in recent works).

Clarity: In general, yes. But the paper can significantly benefit from -- a better discussion of their choice of models (and what happens if the setting was even simpler), -- a better discussion of their results in Section 3.1 (and why they make sense intuitively) -- clarification of their notation in more general settings (and likely implications of their results)

Relation to Prior Work: In general yes. But two clarifications can be made: -- I am not sure, I fully understand the authors claim that their decomposition mitigates some confusion in the prior literature in Sec 4.2. To me, it appears that prior works have not made an inaccurate explanation of what parts blow up in the variance term. Only when one considers an alternate definition as in the paper, does a confusion arise. It would be good to clarify if indeed there was any inaccurate assessment in prior works that is being resolved via the results in this paper. Or whether the confusion may have arisen and this work apriori puts that confusion to rest. -- Putting results from this ICML work in context which also discusses a closer look at bias-variance terms: Rethinking Bias-Variance Trade-off for Generalization of Neural Networks https://arxiv.org/pdf/2002.11328.pdf

Reproducibility: No

Additional Feedback: I have some additonal comments: -- A discussion on the various terms in Theorem 1 is useful. Things that blow up, or are zero. What is the intuition behind them blowing up or being zero? -- May want to clarify that tau' denotes the derivative of tau with respect to gamma (the regularization parameter). -- I wonder to why was a linear generative model + NTK fitted model chosen. It would be easy to discuss the various terms if even the fitted model was linear. -- What can be said for tuned ridge NTK? At least some experiments could be performed? -- Section 5 seems to distinguish between the two variants of bagging (Referring to one as ensemble learning). I am not sure that is a correct way to summarize it )I understand that the two variants are slightly different). While it is good to know how the double-averaging enters different terms and affects the peak, it would be good to clarify and comment if this dual averaging is of any real impact in other more common regimes (n<<d and n>>d); by commenting on when are the problematic terms V_PX and V_PXeps are actually dominant (from Figure 1, they appear to dominate in a narrow region around d==n).


Review 4

Summary and Contributions: This paper analyzes "double descent" phenomenon, which is when the generalization error of a model peaks at the interpolation threshold (as a function either of model complexity or of sample size). The authors develop a fine-grained bias-variance decomposition which decomposes the risk into the bias and several different variance terms. They apply this decomposition to the random features regression model and show which of these terms lead to divergence.

Strengths: This paper addresses an important issue that has lately been focus of much research. It suggests "fine-grained" bias-variance decomposition that allows to clarify several subtle effects. The paper also shows how several previous recent works had misleading/incomplete results based on a bias-variance decomposition that was not sufficiently fine-grained. Figure 1 is a great example of a novel explanation that clarifies prior literature in a unified framework.

Weaknesses: The biggest problem with this paper is that a very similar work has been released as a preprint in March https://arxiv.org/abs/2003.01054 and was presented at ICML in July. The similarity is uncanny: that paper (d'Ascoli et al) also suggested fine-grained variance decomposition; it also applied to random features regression; and it also discussed applications of this theory to model averaging. The d'Ascoli paper is not cited in the current submission. At the same time, the technical apparatus in the d'Ascoli paper and here seems to be very different: this paper derives the results for the random features regression from the random matrix theory, while d'Asoli paper derived the results from the "replica trick" and mean-field theory of statistical physics. Disclaimer: I did not go into details of either derivation. But it seems they are different. So I am willing to believe that the two papers independently arrived at very similar results. The authors absolutely MUST clarify this in the author response. I think the acceptance decision will have to crucially depend on this response. They authors also MUST include into the revised version an upfront discussion of the d'Ascoli paper and how it compares to their paper; if indeed this was an entirely parallel work then the authors should state so in the manuscript. Apart from this critical issue, the paper is very interesting and would be a clear accept, subject to improving the related work section and claryfing some of the exposition (see below).

Correctness: yes (probably - I did not check the math in the appendix)

Clarity: yes (some improvements suggested below)

Relation to Prior Work: yes (some further suggestions see below) apart from not mentioning d'Ascoli et al. paper that could though be simultaneous parallel development...

Reproducibility: Yes

Additional Feedback: This paper analyzes "double descent" phenomenon, which is when the generalization error of a model peaks at the interpolation threshold (as a function either of model complexity or of sample size). The authors develop a fine-grained bias-variance decomposition which decomposes the risk into the bias and several different variance terms. They apply this decomposition to the random features regression model and show which of these terms lead to divergence. This paper addresses an important issue that has lately been focus of much research. It suggests "fine-grained" bias-variance decomposition that allows to clarify several subtle effects. The paper also shows how several previous recent works had misleading/incomplete results based on a bias-variance decomposition that was not sufficiently fine-grained. Figure 1 is a great example of a novel explanation that clarifies prior literature in a unified framework. The biggest problem with this paper is that a very similar work has been released as a preprint in March https://arxiv.org/abs/2003.01054 and was presented at ICML in July. The similarity is uncanny: that paper (d'Ascoli et al) also suggested fine-grained variance decomposition; it also applied to random features regression; and it also discussed applications of this theory to model averaging. The d'Ascoli paper is not cited in the current submission. At the same time, the technical apparatus in the d'Ascoli paper and here seems to be very different: this paper derives the results for the random features regression from the random matrix theory, while d'Asoli paper derived the results from the "replica trick" and mean-field theory of statistical physics. Disclaimer: I did not go into details of either derivation. But it seems they are different. So I am willing to believe that the two papers independently arrived at very similar results. The authors absolutely MUST clarify this in the author response. I think the acceptance decision will have to crucially depend on this response. They authors also MUST include into the revised version an upfront discussion of the d'Ascoli paper and how it compares to their paper; if indeed this was an entirely parallel work then the authors should state so in the manuscript. Apart from this critical issue, the paper is very interesting and would be a clear accept, subject to improving the related work section and claryfing some of the exposition (see below). Disclaimer: I did not make an attempt to follow the math in the supplementary materials, where the amount of algebra is CRAZY. Major issues * See above Medium issues * Related work section could be more complete: a) line 65: Ref [13,14] were as early as [5]. Actually [13] preprint predates [5] preprint. Please reformulate such that it does not give all credit to [5]. b) line 66: You might want to add more references to the work on linear regression: https://arxiv.org/abs/1805.10939 (currently in press in JMLR), and also two very recent preprints might be relevant https://arxiv.org/abs/2006.05800, https://arxiv.org/abs/2006.06386. c) line 67: There is other recent work on double descent in random features model, but I am not sure how relevant it is. Consider including it is relevant: https://arxiv.org/abs/2006.05013, https://arxiv.org/abs/2002.08404. c) Cite and discuss other recent work on bias-variance in neural networks: https://arxiv.org/abs/2002.11328 d) Obviously discuss d'Ascoli paper mentioned above. * line 81: isn't item (3) shown in Mei & Montanari paper that you cite? * The presentation in section 2 is confusing in places possibly due to notation that is either not clear enough, or explained not clear enough. Here are some places where I was confused: a) Eq 1: if this is for a given test point, then what is the expectation over? Unclear. b) Eq 2: E_epsilon corresponds to the sampling noise. What exactly is E_x? Is it the expectation over test points? I assume so but please write it explicitly. c) line 96: Here you say that sampling noise is assumed to come entirely from epsilon. But later on the next page you have a lot of variance terms associated with capital X (e.g. Eq 9) which is sampling noise separate from the label noise epsilon. This is very confusing. Why do you say in line 96 that you "adopt" the setup where sampling noise is only due to epsilon? d) line 114 and Eq 4: please do not drop explicit references to theta and epsilon for the sake of clarity. Why not write E_{theta, epsilon} etc.? e) the decomposition in Section 2.2.2 / Proposition 1 reminds me of ANOVA (analysis of variance) that also decomposes the variance into additive chunks. The standard terms there are "main effects", "two-way interactions", "three-way interactions" etc., which seems to correspond exactly to what is suggested here e.g. in Example 2. Is it correct? Perhaps mention this analogy somewhere? f) Proposition 1 has variance and expectation without indices. I assume here all random variables are defined over the same space and so expectation/variances are over this space. Okay (but could be made explicit). But then in Example 1 there is suddently E_x appearing in addition to V and E. What is E_x? Why did it appear? This is not explained. V and E are still without indices which starts to be confusing here. The same in Example 2. g) In Example 2 you introduce terms indexed by X but above you said there is no samplin noise associated with X (see point c above). * Section 3 before 3.1: Maybe insert an explicit explanation of what plays the role of theta (line 102) in the random features model? I assume it's the W_1 weights but it's never written explicitly. * line 163: does the equivalence to kernel ridge regression only hold asymptotically (when n_1->inf) or always? My impression was that it only holds asymptotically but this is never written explicitly. * Figure 1: maybe show the sum of all terms (entire squared error), at least in panel f? * Figure 1: labels in the bottom row are unreadably small. The same in Figure 2. * Figure 1 and surrounding text: why is bias not going to 0 when n_1/m->inf? It seems that the bias has an asymptote and there is some irreducible bias, as if the model is misspecified. Why does it happen? Maybe insert a comment about that into the text. * Eq 24: comparing to Eq 3, one sees the same problems that I complained above: is P the same as theta? not explained. Why there is X here but not in Eq 3? Not explained. * Eq 31: there are 5 variance terms here. But there are only 3 variance terms in the d'Ascoli paper. How do they correspond to each other? Is there a correspondence? My feeling is that d'Ascoli terms are "sequential" and not symmetric, but they might correspond to some grouping of your terms. Do they? * Figure 2 caption: what is the gamma value here? * line 274: "verified them experimentally" -- where?? All presented results are analytic (when m,n->inf). * line 276: "while is not required... it does indeed cause it" -- very unclear formulation. Consider reformulating. * Conclusions. Can this theory be applied to a linear model, e.g. when sigma() is a linear activation function? What would happen then? Minor issues * line 29: "until the point ... after this point" -- as a function of what? clarify * line 178: maybe clarify that "ridgeless" means gamma->0 ---------- POST-REBUTTAL: I thank the authors for clarifying the issue regarding the d'Ascoli paper. I am leaving my score of 7 which means that I still think it's a good paper that should be accepted. I hope the revised paper will contain some discussion of the d'Ascoli results. In their response, the authors wrote that their results are more general than the d'Ascoli et al results. I think this should be elaborated upon in the revised version. Also, some comments on how the mathematical methods employed there compare to the methods employed here (pros/cons) would be interesting.

[Author Response · NeurIPS 2020]

We thank the reviewers for their careful reviews during these trying times. We also appreciate their additional efforts
considering this rebuttal, which we believe addresses all of their major concerns.

**Prior work.** The reviewers highlighted a few relevant related works that we were unaware of at the time of submission.
We have added the below figure and a discussion of these papers to an updated version of the paper.

*d'Ascoli et al. (2020).* It was surprising to see a paper with such a similar problem setup
and motivation at ICML (unfortunately, we missed the arXiv version). Our efforts were
in fact completely independent and, as R4 points out, use entirely different techniques.
These calculations are exceedingly technical — taking well over a year to complete —
so please bear this in mind when considering if the two works are contemporaneous.
We emphasize that we significantly advance the understanding put forward in d'Ascoli
in at least four important ways. 1) We give a unique, symmetric decomposition of the
variance that applies to *any* model with multiple sources of randomness. The growing
zoo of published decompositions for RF regression can all be understood as special cases
of our decomposition (including d'Ascoli; see figure to the right, which renders their
decomposition in the setup of Fig. 1 of our paper). 2) Our decomposition resolves implicit
ambiguities in the d'Ascoli approach — if they had investigated bagging, they would have
found that it actually removes the divergence, which would seem to contradict their main
conclusion that it is caused by label noise and parameter initialization. 3) Their analysis

was restricted to unstructured RF models, whereas ours extends to the NTK (see Sec. S2). 4) Our techniques from RMT
are significantly more generalizable; we believe the replica method cannot be simply extended to handle non-Gaussian
or correlated matrices, whereas our free probability approach opens the door for such investigations in future work.

*Yang et al. (2020).* This paper defines the total bias and variance similarly to Neal *et al.* (2018), but they do not
decompose the variance. Several of their observations, *e.g.* that label noise increases the total variance and can lead to
double descent, are made precise through the decomposition in our paper.

*Jacot et al. (2020).* This paper studies the relationship between Gaussian RF models and Kernel Ridge Regression.
Unlike our analysis, their bias-variance decomposition is conditional on the dataset $X$. The variance decomposition
itself utilizes the law of total variance, and so can again be viewed as a special case of our fine-grained decomposition.

**Simpler models.** We believe the simplest tractable model that captures all the phenomena of double descent is the
RF model we study here — for linear regression, the number of parameters cannot be varied without simultaneously
changing the training data. Our results do apply to the RF model with $\sigma(x) = x$, but it is rank constrained by
$\min\{n_0, n_1\}$ and does not properly model the over-parameterized regime. We have added a discussion of these points
to the text.

**R1.**   We chose the data distribution to be defined by a linear function for three reasons, which we have now clarified in
the text: 1) it is the setup analyzed in the prior work that we compare to; 2) it reproduces the rich phenomena of double
descent; and 3) the more general case of a nonlinear NN teacher can be reduced to this case, see Sec. S2.

The missing citation to Jacot *et al.* (2018) was an oversight and has been added to an updated version.

**R2.**   We have not found any straightforward intuition for all of the variance terms in our decomposition. However, we
feel that Example 1 is useful. Ultimately, if you ask how to break up a variance into disjoint components that can be
summed together in a meaningful way, this is in some sense the "correct" way to do so. We hope our example of the
variance reducing effects of ensemble and bagging methods demonstrate this. Finally, we have found that considering
the variance of multivariate polynomials in iid Gaussian random variables can be illuminating.

We have expanded our discussion of Neal *et al.* to properly credit its many contributions to double descent and the
related body of work. To be clear, that paper is an important contribution and was a strong motivation for our study.

**R3.**   The many random variables involved make the notation of the variance terms tricky. The decomposition can
be applied to classical models: $D$ would be defined as in our paper as the training data and $P$ would be any random
parameters on which the algorithm depends. For linear regression, for example, $P$ would be the initial weights if it is fit
with GD, or $P = \emptyset$ if one instead utilized the closed-form expression for the predictor. Similarly, for decision trees, the
definition of $P$ depends on how the tree is constructed and whether any randomization is used.

We do not claim that previous works have any mathematical inaccuracies, merely that the interpretation of their results
can lead to an ambiguous or incomplete picture. Our goal is to unify previous decompositions, which is especially
important given the growing number of different approaches, which we discuss in the Prior work section.

[Meta-Review · NeurIPS 2020]

This paper studies the double descent phenomenon in random features models. The authors disentangle the sources of variance contributing to the irregular bias-variance behavior of certain ML methods. All reviewers had a favorable assessment of the paper. The reviewers raised various technical concerns in their reviews including relationship with prior work but thought that the authors’ response adequately addressed these concerns and multiple reviewers raised their score. I concur with this assessment and recommend acceptance.